# VACT: A Video Automatic Causal Testing System and a Benchmark

## Abstract

With the rapid advancement of text-conditioned Video Generation Models (VGMs), the quality of generated videos has significantly improved, bringing these models closer to functioning as "*world simulators*" and making real-world-level video generation more accessible and cost-effective. However, the generated videos often contain factual inaccuracies and lack understanding of fundamental physical laws. While some previous studies have highlighted this issue in limited domains through manual analysis, a comprehensive solution has not yet been established, primarily due to the absence of a generalized, automated approach for modeling and assessing the causal reasoning of these models across diverse scenarios. To address this gap, we propose VACT: an *automated* framework for modeling, evaluating, and measuring the causal understanding of VGMs in real-world scenarios. By combining causal analysis techniques with a carefully designed large language model assistant, our system can assess the causal behavior of models in various contexts without human annotation, which offers strong generalization and scalability. Additionally, we introduce multi-level causal evaluation metrics to provide a detailed analysis of the causal performance of VGMs. As a demonstration, we use our framework to benchmark several prevailing VGMs, offering insight into their causal reasoning capabilities. Our work lays the foundation for systematically addressing the causal understanding deficiencies in VGMs and contributes to advancing their reliability and real-world applicability.

## 1 Introduction

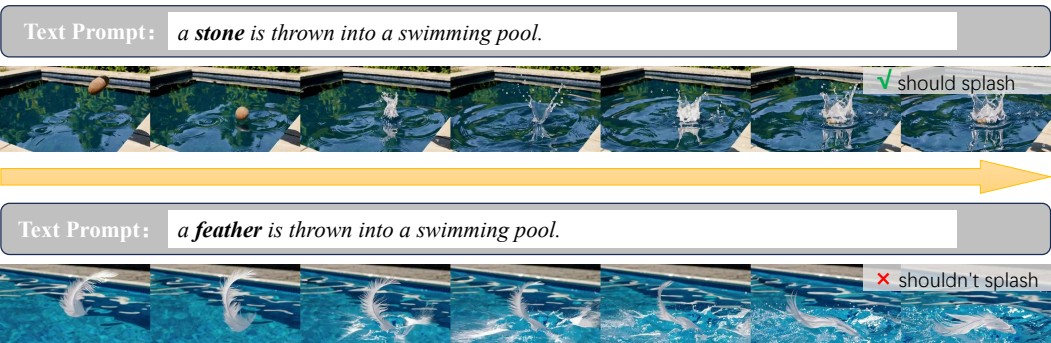

Figure 1: Videos generated by OpenAI Sora, shown as frames. The text prompt of the **above** is: *"a stone is thrown into a swimming pool"*; **Below** is: *"a feather is thrown into a swimming pool"*. Both the generations show *noticeable splashes*, which is correct for the above (stone) but **incorrect** for the **below** (feather).

With the rapid development of Video Generation Models (VGMs), generated videos are becoming increasingly indistinguishable from real recordings. VGMs, particularly text-to-video (T2V) mod-

els[1], are expected to serve as "world models" or "world simulators", allowing users to generate scenes from text descriptions of real-world events or environments. This approach is cheaper, faster, and more scalable than arranging and recording real-world scenes and is expected to benefit fields like robotics, autonomous driving, and video understanding.

However, the "*hallucination*" problem hinders the progress, which refers to a generation that seems correct but contains factual errors or fabrications. While VGMs have made significant strides in video quality—such as clarity, dynamic range, and continuity—they still struggle with issues like cause-and-effect confusion, detail errors, and incorrect object relationships, making the videos appear misleading upon closer inspection.

In Figure 1, OpenAI Sora (OpenAI, 2024) is required to generate videos for two scenarios: "*a stone is thrown into a swimming pool*" and "*a feather is thrown into a swimming pool*". In both cases, an obvious splash and ripples occur around the object. While in the stone scenario the splash is accurate, the feather scenario fails to follow the correct physics principles, as the feather is too light to create a noticeable splash or ripples in reality. Here, the model seems to learn a **spurious correlation** (or "**shortcut**") between "*object hitting water*" and "*splash*", without understanding the actual causal factors, such as *mass* and *velocity*, making it difficult for the VGMs to generalize to less typical situations or to be utilized as "world simulator". We provide similar results with other VGMs and more analysis in Appendix A.

Although some works have acknowledged the hallucination problem in VGMs and proposed preliminary benchmarks to identify commonsense violations (Bansal et al., 2024; Meng et al., 2024), most of them rely on manual design of physical rules and test cases and focus on limited fields. However, real-world causal relationships are highly complex, with different scenarios involving different physical laws. Furthermore, even a simple scenario can involve various causal relationships. For instance, in the case of "*two objects collision*", dynamics might focus on mass, velocity, and elasticity to determine the object motion after collision, while material properties like hardness or brittleness might determine whether the objects deform or break. More complex relationships, like sparks from a flint or splashes from wet objects, further highlight this complexity, making it difficult to systematically address the hallucination problem through manual labeling.

To address this challenge, we propose an **automatic** method for identifying causal rules in specific scenarios and evaluating models' *causal understanding*. Our process, utilizing an LLM assistant, identifies possibly involved causal factors for a given scenario and rules between the factors, and then describes the rules as a *causal system*. The automatic generation proves effective by comparing them with human annotations. *Intervention experiments* (Pearl, 2009), one of the effective methods in causal inference, are then used to assess causal behaviors in VGMs by varying the text prompts with different factor values. For example, as shown in Figure 1, replacing a heavy stone with a light feather reveals that the VGM had not correctly learned the causal rules related to density.

To analyze causal learning in VGMs, we define three levels of consistency: *text consistency* (to follow explicit conditions), *generation consistency* (to maintain consistent generation under the same conditions) and *rule consistency* (to learn correct causal rules). Each of these three metrics forms the prerequisite for the next, creating a multi-level evaluation for the "world simulator" with progressively increasing difficulty.

In summary, we introduce the **V**ideo **A**utomatic **C**ausal **T**esting (**VACT**) system, which requires no human annotation, scoring, or intervention. To our knowledge, this is the first approach to automatically apply causal analysis tools for testing causal understanding in VGMs. It is scalable, generalizable, and can be applied across various fields without additional manual effort to propose and write the rules in test cases, while also providing a detailed causal analysis of model behavior. To validate its effectiveness and generalizability, we conducted crowd experiments, where 19 different scenarios generated by our system (involving various scenarios such as motion, force, light, heat, fluid, material, etc.) are compared to those human written, showing that automatic annotations achieve comparable (and even better) performance with human annotation.

---

[1]In this paper, the term VGM refers specifically to T2V models. Text-conditioned generation is the most versatile and user-friendly method in world simulation, whereas image-conditioned models can enable T2V generation by combining with a text-to-image model.

We also construct a benchmark that covers these 19 scenario spanning 99 causal factors and 49 causal rules. For each video generation model, we generate 718 evaluation videos under controlled interventions and systematically score text, generation, and rule consistency. Using this benchmark, we assess current models and provide their rankings across multiple dimensions. Our results reveal that existing video generation models still exhibit significant limitations in understanding causal rules. The benchmark also offers a practical basis for mitigating hallucinations through dataset supplementation and reinforcement-learning–based alignment.

## 2 RELATED WORK

**Text-to-Video (T2V) generation models**    T2V models generate videos from textual descriptions. Early methods using generative adversarial networks (GANs) (Wang et al., 2020) and variational autoencoders (VAEs) (Li et al., 2018; Pan et al., 2017) faced limitations like low resolution and diversity. Starting with Video Diffusion Models (Ho et al., 2022), recent advances in diffusion models have significantly improved T2V generation. CogVideo (Hong et al., 2022) combines a pre-trained text-to-image model with a T2V framework, facilitating effective learning. LaVie (Wang et al., 2024) enhances video quality with interpolation and super-resolution techniques. VideoCrafter2 (Chen et al., 2024) leverages Diffusion Transformers(DiT) (Peebles & Xie, 2023) to synthesize high-quality videos by refining generated sequences with high-resolution images. Models like Gen-3 Alpha (Esser et al., 2023), HunyuanVideo (Tencent, 2025), Haoluo (MiniMax, 2024), pika (pika, 2024), Kling (Kuaishou, 2024), and Sora (OpenAI, 2024) further push the boundaries with advanced architectures and techniques. Comprehensive reviews on the developments are available in Xing et al. (2024) and Sun et al. (2024).

**Evaluation for video generation models**    The rapid advancement of VGMs has underscored the need for accurate quality evaluation. Traditional metrics like IS (Salimans et al., 2016), FVD (Unterthiner et al., 2019), and CLIP (Hessel et al., 2022; Liu et al., 2023) assess only limited aspects like frame quality, and often fail to align with human judgment. To address this, benchmarks like V-Bench (Huang et al., 2024) and EvalCrafter (Liu et al., 2024) provide more comprehensive evaluations, considering factors like subject consistency, spatial relationships, dynamic degree, and action continuity. However, these metrics still focus on visual quality while overlooking the logical coherence of events and scenes in videos.

**Evaluation for world simulators**    As video quality further improves and the concept of a "*world simulator*" becomes an expectation, the focus has shifted from *aesthetics* to *authenticity* — ensuring generated content follows real-world physics rules. Recent benchmarks including VideoPhy (Bansal et al., 2024) and PhyGenBench (Meng et al., 2024) have made initial attempts to address this. VideoPhy uses human collection and evaluation to verify commonsense violations within three classes of physics scene: solid-solid, solid-fluid and fluid-fluid. Their benchmark heavily depends on human efforts and is hard to generalize to new field. Their attempts to fine-tune a vision-text model for automatic ranking have yet to align well with human assessments, limiting its scalability. PhyGenBench (Meng et al., 2024) tests on 27 *human-designed* physics laws, using LLM-generated questions to check rule fidelity in videos by a video language model. However, it remains human-dependent in rule design and does not distinguish between true causal understanding and shortcut-based pattern recognition. For instance, a model generating "stone splashing water" doesn't prove it understands the physics of the splash, as testing with a feather (density) or other setting is necessary.

As concurrent works, Motamed et al. (2025) evaluate whether VGMs learn physical principles by predicting video continuations. They first film 396 videos, each divided into two segments: a conditioning segment and a subsequent ground truth segment. Predictions generated by the models based on the conditioning segment are compared against the ground truth to assess accuracy. Li et al. (2025) collect image and text conditions about 5 common physics laws such as Newton's first law and fluid mechanism. They finetune a VLLM using extensive human annotations to identify the potential errors in videos.

Our work further expands this series of work in two aspects: 1) **Full automation**: our approach eliminates manual rule design, allowing physical rules to be automatically inferred from a short textual descriptions, enhancing scalability. 2) **Causal evaluation**: We introduce *intervention exper-*

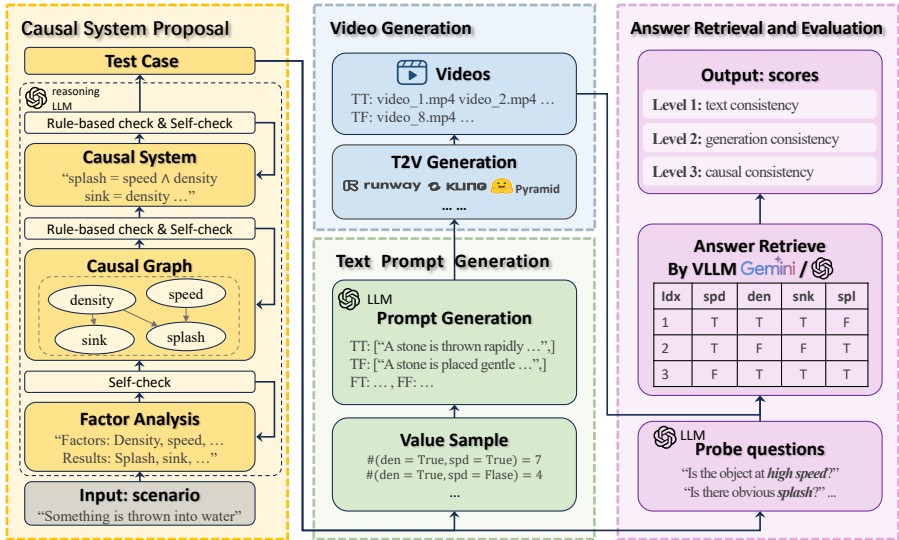

Figure 3: Pipeline of VACT. The pipeline mainly consists of four parts: causal system (i.e. test case) proposal (yellow), text prompt generation, (green), video generation (blue), answer retrieval and evaluation (pink). The pipeline receive a sentence describe a scenario as input and automatically evaluate video generation models *without* any human supervision or annotation.

*iments* to test whether models truly understand physics rather than relying on shortcuts, ensuring a more robust assessment.

Additionally, other works like (Kang et al., 2024) explore 2D physics simulation in VGMs, while WorldSimBench (Qin et al., 2024) assesses world simulators from an embodied perspective. These works, along with ours, collectively contribute to a multi-faceted understanding of world simulators,

# 3 PIPELINE OF AUTOMATIC CAUSAL RULE TESTING

## 3.1 SCENARIO-BASED CAUSAL RULE TESTING

Our tests begin with **scenario**s, short text descriptions of an event, such as "something is thrown into a swimming pool" (in Figure 1). Each scenario involves variables representing object or event properties, linked by causal relationships modeled as a causal graph and a causal system.

**Definition 1** (Causal graph and system (Pearl, 2009)). A deterministic **causal system** over a set of variables $\mathbf{V}$ is a directed acyclic graph $G$ with the node set $\mathbf{V}$ and edge set $\mathbf{E}$, and a series of structural equations $V_j = f_j(pa(V_j))$ for every $V_j \in \mathbf{V}$, where $pa(V_j) = \{V_k \in \mathbf{V} : V_k \to V_j \in \mathbf{E}\}$ are the parents of the node $V_j$. The graph $G$ is called the **causal graph**. Furthermore, let $\mathbf{X} = \{V_j \in \mathbf{V} : pa(V_j) = \emptyset\}$ be the **root** (**cause**) variables and $\mathbf{Y} = \mathbf{V} \setminus \mathbf{X}$ be the **non-root** (**outcome**) variables.

We illustrate this with an example in Figure 2, where the system captures commonsense physical knowledge—for instance, density determines whether an object will sink, while speed, size, and density collectively influence the splash. The directed edges in the graph represent causal relationships between variables, such as the edge "high density" → "object sink" indicating causation, while there is no causal effect between "large size" and "object sink", since a dense object will sink regardless of its size. Root variables (blue) are basic conditions of a scenario that can be directly manipulated. Non-root variables (orange) are outcomes of other variables that cannot be directly adjusted but may influence other factors. For example, the non-root variable "splash" causes "ripples", which in turn causes "sparkling" (Yarin, 2006). The basic unit of our VACT

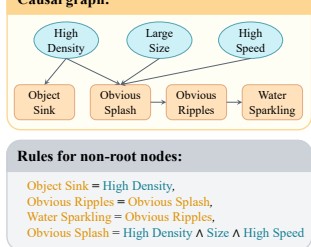

Figure 2: An example causal graph and system: "*throwing something into a swimming pool*". Blue denotes root nodes and orange denotes non-root nodes.

is a causal system, consisting of these rules (i.e. the equations). A scenario can yield different test cases depending on selected factors.

For the sake of clarity, we define each variable in descriptive levels, mapping continuous attributes that are not reliably measurable from video (e.g., speed, mass, elasticity) to stable, interpretable categories (e.g., slow/fast; low/high) that are directly observable in footage, robust to noise and suited to controlled interventions, which aligns with how humans naturally judge physical events and supports consistent, rater-reliable assessment of generated videos. Additionally, the variables must be visually discernible[2] (*visibility*) to ensure suitability for video evaluation, and all root nodes can be set *independently*. Since these values are later retrieved by a VLM, we further exclude scenarios where the relevant factors cannot be reliably perceived from video—such as those involving barely visible objects, motions beyond typical temporal resolution, non-observable physical processes, subtle visual effects, or material properties without clear visual signatures—so that every variable remains consistently identifiable during evaluation. If a video generation model learns the physical laws of a scenario, it should have learned the rule $f$. Thus, by analyzing variable states in generated videos under different conditions, we can assess whether the model understands the underlying law.

## 3.2 LLM-AIDED AUTOMATIC GENERATION OF TEST CASES

As discussed in Section 1, extracting key causal rules from scenarios is challenging due to their inherent complexity and diversity. This task requires both creativity (to imagine alternative scenarios) and commonsense reasoning (to recognize common causal patterns). Fortunately, the advanced commonsense reasoning capabilities of Large Language Models (LLMs) enable automation of this process. We designed a multi-step annotation method that leverages LLMs, incorporating task splitting, rule checking, and self-correction to ensure accuracy and reliability.

As illustrated in Figure 3 (yellow part), our system accepts a scenario (a brief textual description) and prompts the LLM to sequentially perform the following steps: 1) identify key causal factors and outcomes within the scenario, 2) discover relationships between these elements to construct a causal graph, and 3) derive expressions representing the discovered relationships. This step-wise splitting helps LLMs focus on sub-targets and provides more thinking space, ensuring that the output satisfies our requirements and making it possible for an LLM to perform such a challenging task—automatic test case generation.

After each step, we implement *rule-checking* and *self-checking* mechanisms. If errors are identified, such as formatting issues, unusual relationships, or isolated nodes in the causal graph, we provide explicit error feedback to the LLM, prompting it to self-correct. If no errors are found, the model is prompted to perform an additional self-check. The step-splitting and self-correction strategy ensure that errors are detected and corrected, significantly improving the overall quality of generated outputs (Appendix B.2).

Another key strategy is to prompt the LLM not only to produce an answer, but also to explicitly explain factors and rules they proposed. Although this increases the number of output tokens by about 10%, it has three benefits: (1) the explanations encourage more careful reasoning and thus improve reliability; (2) early-stage explanations can be reused in later steps—for example, factor explanations directly specify how a VLM should extract their values; and (3) the additional interpretability helps us, as designers, refine the prompts and the overall pipeline (Appendix B.4).

The final test case comprises both the rule expression (as well as the explanation) and the original scenario text. Detailed generation requirements, inspection indicators, the comprehensive description and some design principle of the process can be found in Appendix B.

**Crowd experiments**  We evaluate the effectiveness of the automatic generation by crowd experiments. We collected 19 diverse scenarios and generated three causal systems for each, 57 in total. For comparison, three undergraduates manually annotated the same scenarios using identical instructions given to the LLM, resulting in another 57 *human-annotated* causal systems. Then, another five undergraduates blindly scored both human and LLM annotations based on three criteria: *requirement* (adherence to visibility, binarity and root independence), *rationality* (reasonableness

---

[2]The visualization here is a relative requirement. For example, although the density is essentially invisible, we can infer the density of an object through its visible material.

of factor selection), and *soundness* (accuracy of causal rules). For each criteria, the scores range from 1 which means there are essential error to 4 which means the annotation perfectly meets the requirement.

The average score from 5 scorer and 57 samples are reported in Table 1. LLM-generated annotations surprisingly **outperformed** those from human, demonstrating the effectiveness of the LLM-driven process and its strong alignment with human reasoning. Specifically, we found that there are occasional controversies in human written cases, involving whether nodes are strictly independent, whether an attribute is "visually discernible", and the strength of some causal relationships may be ambiguous. However, nodes generated by LLMs are generally more reliable, and therefore more in line with requirements. We provide the details of the crowd experiments, the score distribution of each scorer and further analysis in Appendix D.

Table 1: Average scores from crowd experiments

| Source | Requirement | Rationality | Soundness | Average |
|--------|-------------|-------------|-----------|---------|
| **LLM** | $3.91_{\pm 0.02}$ | $3.49_{\pm 0.04}$ | $3.78_{\pm 0.03}$ | $3.73_{\pm 0.02}$ |
| **Human** | $3.80_{\pm 0.03}$ | $3.51_{\pm 0.04}$ | $3.63_{\pm 0.04}$ | $3.65_{\pm 0.03}$ |

### 3.3 AUTOMATIC INTERVENTION EXPERIMENT PIPELINE

Given a causal system, our testing as intervention experiments contains five parts: sampling, prompt generation, video generation, answer retrieval, and evaluation, as shown in Figure 3. Details and the prompt used in these steps are in Appendix E.

**Sampling**. In our intervention experiments, observations are videos characterized by varying condition values $\mathbf{X}$. We sample diverse combinations of root values $\mathbf{X}$ for these intervention. The sample size for each $\mathbf{X}$ value combination is determined by metrics discussed in Section 4. Given the high cost of video generation, it is crucial to minimize the number of videos needed while ensuring accurate and stable metrics measurement. Therefore, we (1) conduct preliminary experiments to determine the minimum number of samples that can ensure the relative stability and validity of each metric and (2) strategically share generated videos across different metrics whenever feasible. Finally, the number of sample for each $\mathbf{X}$ value combination is set to the maximum requirement across the three metrics. We provide the details of samples in Appendix F.4 and the preliminary experiments used to determine the minimum number in Appendix H.4. Through our experiments, we conduct that approximately 30 to 50 videos per causal system are sufficient to achieve stable evaluations.

**Prompt generation**. Given $\mathbf{X}$ values, we use an LLM to generate sentences to constrain restrict the variables in the scenario as meeting the given values. For example, in the scenario "throwing something into a swimming pool", the prompt "*a large rock was thrown quickly into the pool*" sets three root variables: high density, large size, and high speed to true, while another prompt "*a tiny stone is gently placed on the water*" alert large size and high speed to false. These sentences serve as text prompts for video generation.

**Video generation**. The prompts generated are provided to the tested VGM. These models are treated as black boxes, requiring no constraints on their structure.

**Answer retrieval**. Each generated video serves as an observation of the intervention experiments for the causal system. We check (1) whether it follows the text description of variable values $\mathbf{X}$ and (2) whether the generated values $\mathbf{Y}$ align with the causal rules. Following Meng et al. (2024), we use a vision-LLM (VLLM) to retrieve the observed values $\hat{\mathbf{X}}$ and $\hat{\mathbf{Y}}$ in each video, by prompting the VLLM with the video and some "*yes-no*" questions (i.e. *probes*). These probes are also generated by an LLM given the variable list of a causal system. Notice that our variable has been required to be binary when generating test cases, so the generation of these questions is a very simple tasks.

To make sure the values of factors are correctly retrieved from the videos, we test three VLLMs (GPT-5, Qwen3-VL-Plus, and Gemini 2.5 Flash) and compare each model's answers with the majority-vote labels. As shown in Table 2, all three VLLMs achieve high consistency rates with

Table 2: Consistency rate of each VLLM with respect to the majority-vote labels.

|  | GPT-5 | Qwen3-VL-Plus | Gemini 2.5 Flash |
|---|---|---|---|
| Consistency rate | .93 | .94 | .93 |

respect to the majority vote, indicating that in most cases their predictions are consistent. Additionally, in the benchmark evaluation, we use the majority voting to make sure the better accuracy.

We adopt an LLM and a VLLM to automate the steps *prompt generation*, *probe generation*, and *answer retrieval*. To assess the reliability of this pipeline, we quantitatively checked a substantial subset of each component across multiple scenarios. For *prompt generation*, we randomly inspected 106 prompts from 4 scenarios and found only 2 inaccuracies. For *probe generation*, we examined 85 factor–question pairs from 18 scenarios , all of which were correct. For *answer retrieval*, we evaluated 168 question–answer pairs (42 per VGM for 4 models across 3 scenarios) and observed only 6 mismatches with human judgments. We report the sampled accuracy and the 95% confidence interval of the estimated overall accuracy in Table 3. The table shows that with our pipeline and prompt design, the accuracy of each automatic accuracy is satisfying. We provide detailed breakdowns and examples in Appendix G.

| Component | Prompt generation | Probe generation | Answer retrieval |
|---|---|---|---|
| Sample accuracy | 0.98 | 1.00 | 0.96 |
| 95% confidence interval (CI) | [0.96, 1.00] | [0.96, 1.00] | [0.94, 0.99] |

Table 3: Manual check results for different LLM/VLLM components of the pipeline. For *probe generation*, all checked pairs were correct; the 95% confidence interval [0.96, 1.00] follows the standard rule-of-three approximation ($1 - 3/n$ with $n = 85$) for zero observed errors.

## 4 THREE LEVELS OF CAUSAL ABILITY AND THE CORRESPONDING METRICS

To assess the deviation of the model's understanding of the objective world, we propose a three-level framework of causal capabilities with corresponding evaluation metrics. The mathematical definitions are provided in Appendix F. Here, we focus on an intuitive description.

**Text consistency** The first level assesses whether the model accurately reflects the state of every variable described in the prompt. By generating a video from a detailed prompt specifying variable values, the resulting video should correctly reflect those values. It is a fundamental requirement for not only the general usage of video generation models but also our intervention experiments because we need to control video variables through text. The similar metrics have proposed in some previous benchmarks, such as video-text consistency in VBench (Huang et al., 2024), but it is still different. In our test, a video generation requires multiple important but complex attributes simultaneously, and there are some uncommon combinations. Therefore, our test is more difficult than similar evaluations.

We use two types of prompts: the standard prompts (described in Section 3.3 and also used in the next two metrics) specifies *all root* variables $\mathbf{X}$. In addition, we generate prompts constraining all variable values (including roots $\mathbf{X}$ and non-roots $\mathbf{Y}$), describing not only the condition but also expected results in the scenario. These two types correspond to the "*root*" score and "*all*" score, respectively. The first type aligns with common real-world application, where the results are not stated before generating. The second serves as a complementary test about whether explicitly stating expected outcomes helps the model generate correct physical behaviors. For each setting, we measure text consistency using the *average accuracy* of whether the observed values match the described ones.

**Generation consistency** The second level evaluates whether the model stably produces the stable and predictable outcomes given identical causes $\mathbf{X}$, or if its outcomes vary arbitrarily due to unrelated factors like random seed or wording differences. To measure this, we group the samples by

identical $\mathbf{X}$ values, and calculate the *mean variance* of outcomes $Y_j$ within each group. Since grouping is based on $\mathbf{X}$, errors arise if text consistency (level one) is imperfect. To address this, we use two scoring criteria: "*truth*" score groups samples according to expected $\mathbf{X}$ in text prompt and evaluates end-to-end consistency in practical usage, while "*observe*" score groups samples according to actually observed $\hat{\mathbf{X}}$ which is retrieved from videos and ignores condition generation errors. Notice that as text consistency improves, both scores should converge to the same. The variance ranges is $[0, 0.25]$ so we report the consistency score as $1 - 4\operatorname{Var}(Y_i)$, make it in $[0, 1]$ and larger-is-better.

**Rule consistency** The third level, our main and long-term goal, tests the model's ability to learn and apply causal rules consistent with the real world. For one test case, we first calculate the accuracy for each rule (i.e. for each outcome $Y_j \in \mathbf{Y}$). For each outcome $Y_j = f(pa(Y_j))$, we sample prompts to make sure the groundtruth value of $Y_j$ is 50% True and 50% False. Having sampled videos $\mathbf{S} = \{s^{(1)}, \ldots, s^{(n)}\}$, the rule consistency is calculated as (1) the average accuracy $\sum_{i=1}^{n} \mathbb{1}(Y_j^{(i)} = \hat{Y}_j^{(i)})$, or (2) a threshold-based 0-1 score $\mathbb{1}\{\operatorname{mean}[\mathbb{1}(Y_j = \hat{Y}_j)] \geq t\}$. Then the score of the test case take the average value among variables $\mathbf{Y}$. Here, we also distinguish two scores, "*truth*" score using the groundtruth $pa(Y_j)$ and "*observe*" score using the observed $\hat{pa}(Y_j)$ to get the expected $Y_j = f_j(pa(Y_j))$ where the latter isolates the direct cause-effect relationship, excluding errors from unexpected causes.

**Scores on samples** Though our experiments need comparison and evaluation on the scenario level where tens of videos could be involved, these metrics can be also applied to individual videos by redistributing the scores of each video according to its proportion in the overall score. It convenient to identify specific instances where the model's performance deviates, provide insights into its learning mechanisms or provide reward in a reinforcement learning stage. See Appendix F for detailed definitions and some example analysis in Appendix I.2.

## 5 A BENCHMARK OF CAUSAL RULE TESTING

Based on our collected scenarios and test cases generated by our VACT system in crowd experiments, we choose those with highest human score as well as some post-filtering to create a benchmark called "VACT Bench". Our benchmark covers 19 scenarios, comprising 49 causal rules, 99 causal factors, and 718 evaluation videos.

We evaluate prevailing VGMs using our benchmark. Table 4 shows our benchmarking results on some prevailing models. Generation Consistency is computed according to the formula provided in Appendix F.2, with ; therefore, we report the values in the table after multiplying them by 4.

These models occasionally generate videos that are off-topic or with missing subjects and confusing logic. To avoid the score being affected too severely by the low-quality generation, we allowed the VLLM to answer "N/A" (in addition to yes/no) during answer retrieval, filtering out all observations marked as "N/A" across all metrics and reporting the "N/A" ratio in results.

Here, rule consistency is calculated as the average accuracy score. For details on the tested models, testing costs, the impact of N/A, and sample efficiency, see Appendix H.1 to H.4.

From the results in Table 4, there are notable differences in generation quality across models. In terms of text consistency, Hailuo performs relatively poorly, while Veo3-Fast achieves the best performance. In terms of rule consistency, Veo3-Fast and Pika perform best. Current video generation models exhibit a certain degree of causal understanding, but their overall performance remains unsatisfactory. This indicates that our benchmark can serve as a long-term objective for future research in this field.

To verify that the low scores are mainly caused by unfaithful or inaccurate video generation rather than errors introduced by our VLLM/LLM-based evaluation pipeline, we additionally record real-world videos for 8 scenarios in the benchmark where filming is feasible, and report the results in Table 4. These metrics can be regarded as an empirical upper bound that already accounts for the noise of our LLM/VLLM evaluators. The scores are substantially higher than those of all T2V models, indicating that the main limitation lies in the causal understanding of current T2V models.

Table 4: VACT benchmark on prevailing VGMs. The rule consistency is calculated by the "average accuracy". Rule consistency is calculated by the average accuracy, and generation consistency values are reported as the actual scores multiplied by 4. *: Sampled test cases.

| Model Names | N/A ratio | Text Consistency ↑ | | Generation Consistency ↓ | | Rule Consistency ↑ | |
|---|---|---|---|---|---|---|---|
| | | all | root | truth | observe | truth | observe |
| Recorded* | .00 | .83 | .88 | .10 | .07 | .95 | .93 |
| CogVideo | .21 | .61±.02 | .68±.02 | .33±.03 | .25±.06 | .62±.03 | .82±.04 |
| Hailuo | .22 | .58±.03 | .71±.03 | .25±.03 | .27±.04 | .62±.03 | .82±.05 |
| Kling | .19 | .62±.02 | .70±.03 | .20±.04 | .24±.05 | .65±.03 | .76±.04 |
| Veo3-Fast | .13 | .70±.02 | .75±.02 | .19±.02 | .25±.04 | .67±.03 | .76±.04 |
| Pika | .20 | .69±.02 | .74±.02 | .21±.03 | .29±.04 | .68±.03 | .79±.04 |

Based on these observations, we provide a comprehensive analysis and evaluation of the models and their scores. The results are explained as follows:

**Text consistency**   The observed text accuracy ranged from 58% to 70%. Veo3-Fast achieved 70% text consistency, but as a fundamental metric, this result remains unsatisfactory. This indicates that existing models struggle to accurately generate variables from the provided text, whether they are causal or outcome variables. Although text fidelity has recently improved (Sun et al., 2024), our scenarios each contain multiple variables, and we specify multiple variable values simultaneously. Thus, our benchmark specifically requires models to handle multiple variables at once, including those corresponding to less common scenarios. The results highlight the models' difficulty in dealing with complex properties and rare situations. Quantitatively, Pearson correlations between causal-graph size and most accuracy-style metrics are negative (see App. H.5). This suggests that current models are still constrained to common scenarios and lack the generalization capability needed to effectively combine independent variables in broader contexts—a necessary capacity for building world simulators.

**Generation consistency**   The generation consistency scores of all models are generally around 0.05 (note that the results reported in the figure are the actual values multiplied by four).This indicates that models have learned relatively stable "rule", consistently producing similar outcomes for identical input variables $\mathbf{X}$. However, this stability is not necessarily indicative of positive performance. To better understand this, we consider text consistency and generation consistency results together. Although around 30% of root variables being generated incorrectly (corresponding to around 70% text consistency), the "truth" score and "observation" score in generation consistency remain closely aligned. This alignment occurs despite the fact that these scores categorize the same set of samples according to different criteria: the ground truth inputs $\mathbf{X}$ and the observed generated inputs $\hat{\mathbf{X}}$, respectively and there is a significant difference between these two values.

Such a pattern suggests that models may be **producing fixed outcomes $\mathbf{Y}$ regardless of variations in input variables $\mathbf{X}$**. This type of stability reflects what we call a "*degenerative*" rule — akin to a constant function. An illustrative example is shown in Figure 1, where any object entering water invariably generates a splash. This behavior was further validated through manual inspection, as detailed in Appendix I.1.

**Rule consistency**   Finally, we directly evaluate the correctness of the rules learned by the models. The average "truth" rule accuracy is around 65%, with the best-performing model reaching 68%; the average "observe" rule accuracy is about 79%, with the best model achieving 82%. Comparing the recorded videos about 93% and 95%, this suggests that the models have captured certain causal rules from the real world, but their understanding remains limited. It is important to note that the truth score is computed by comparing the root variable values specified in the prompt with the non-root variable values generated in the video, whereas the observe score is computed solely from the variable values observed in the generated video. The fact that the observe score is higher than the truth score indicates that the generated videos exhibit stronger internal consistency—that is, the videos themselves align better with causal rules—but this does not necessarily mean that the models have truly understood real-world causal mechanisms.

When presented with different values of the same scenario variables, video models tend to generate outcomes corresponding to the more **common** values. In these common cases, the generated videos achieve relatively high rule consistency. However, In less common cases, the models may revert to producing the common outcomes, which either leads to a mismatch with the variable settings in the prompt—thereby lowering text consistency—or results in videos that violate causal rules. This demonstrates that video generation models still lack sufficient understanding of real-world causal relationships. Overall, the low scores of the models mainly stem from two factors: (1) misunderstanding of causal rules, and (2) difficulty in accurately grounding text-specified variables into the generated video scenes. To further demonstrate the *"collapse to common"* behavior, we conduct some error analysis for representative scenarios. In the table 6, even the powerful Veo3-Fast model exhibits a clear tendency to ignore the true causal relationship and always generate a *"more common"* outcome. We provide more analysis in Appendix H.6).

Table 5: Metrics for rule consistency by applying threshold for each rule.

| Name | Truth | | | | Observe | | | |
|------|------|------|------|------|---------|------|------|------|
| **Threshold** | 0.65 | 0.75 | 0.85 | 0.95 | 0.65 | 0.75 | 0.85 | 0.95 |
| Hailuo | $.33_{\pm.08}$ | $.20_{\pm.06}$ | $.10_{\pm.05}$ | $.05_{\pm.04}$ | $.78_{\pm.07}$ | $.64_{\pm.07}$ | $.55_{\pm.08}$ | $.43_{\pm.08}$ |
| Kling | $.46_{\pm.08}$ | $.29_{\pm.07}$ | $.12_{\pm.05}$ | $.02_{\pm.02}$ | $.72_{\pm.05}$ | $.65_{\pm.07}$ | $.42_{\pm.08}$ | $.18_{\pm.08}$ |
| CogVideo | $.50_{\pm.08}$ | $.26_{\pm.07}$ | $.19_{\pm.06}$ | $.16_{\pm.06}$ | $.74_{\pm.06}$ | $.70_{\pm.07}$ | $.51_{\pm.07}$ | $.45_{\pm.08}$ |
| Veo3-Fast | $.53_{\pm.08}$ | $.34_{\pm.07}$ | $.19_{\pm.06}$ | $.11_{\pm.05}$ | $.65_{\pm.07}$ | $.60_{\pm.07}$ | $.40_{\pm.08}$ | $.30_{\pm.08}$ |
| Pika | $.50_{\pm.08}$ | $.42_{\pm.08}$ | $.24_{\pm.07}$ | $.12_{\pm.05}$ | $.75_{\pm.07}$ | $.73_{\pm.07}$ | $.55_{\pm.08}$ | $.25_{\pm.08}$ |

Table 6: Models collapse to *"common"* value. Model: Veo3-Fast.

| Scenario | Outcome | Some related factors | Expected T/F ratio | Generated T/F ratio |
|----------|---------|---------------------|--------------------|--------------------|
| The teapot pours hot tea into the cup. | Water line rise to rim | Teapot tilted Spout align with teacup | 10:10 | 19:1 |
| An air mattress floats on a pool | Waterline rises | Person boards during clip | 12:8 | 4:16 |
| A person strikes an ice block with a hammer | Glancing slide | Perpendicular strike | 10:10 | 2:18 |

## 6 CONCLUSION

In this paper, we propose an automated system for modeling causal relationships in scenarios and evaluating the causal behavior of VGMs. By combining LLM's commonsense understanding with intervention experiments, our automatic system can assess the causal learning in VGMs across diverse domains, scenarios, and rules without any human annotation. We validated its effectiveness through crowd experiments and manual checks. We introduced three progressive causal metrics to comprehensively analyze the models' causal behavior. Using this system, we created a benchmark and identified key causal flaws in existing models. As a long-term target, this work lays the foundation for large-scale detection of shortcut or biased learning, supplementing comprehensive training datasets, or reinforcement learning.

## 7 LIMITATION

We acknowledge several limitations in our current work. First, although the LLMs can already generate high-quality testbeds, occasional errors may still occur. For scenarios requiring extremely high quality assurance, human check and assistance is still recommended. Second, our evaluation assumes that model generate high-quality videos but some models still struggle with text understanding and coherent video generation, hindering the analysis of their causal behavior. We view our system as a forward-looking tool, believing that as video generation models rapidly improve, causal behavior analysis will become more critical.

## LLM USAGE

We use LLM to (1) help to polish writing and (2) help to search on related work. We always carefully read the response of LLM and use the response of LLM only as a reference and are responsible for the content.

## ETHICS STATEMENT

Our proposed benchmark and pipeline are designed solely for video evaluation tasks, and thus do not involve any ethically sensitive tasks or inference about social attributes. Any student-related data have been properly anonymized and de-identified to prevent identity disclosure or re-identification.

## REPRODUCIBILITY STATEMENT

Our paper focuses on providing a pipeline for evaluating video generation models and a reproducible benchmark. To ensure the robustness and reliability of our results, we report the variance for most of our experiments. We will release the majority of our training/testing data, training code, and testing code to guarantee reproducibility.

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

# Content List

## A  THE "STONE" AND "FEATHER" EXAMPLE

In Figure 1, we demonstrate that OpenAI's Sora (OpenAI, 2024) fails to distinguish between the different effects of a stone and a feather falling into water. This is not an isolated case of Sora. In figure 4, we show the generation of CogVideoX-2 (Hong et al., 2022) and Runway Gen-3 Alpha (Runway, 2024), showing that this spurious correlation is a common phenomenon that may exist in various models. These models seem to "directly" substitute the stone with the feather, without understanding the significant differences in the outcomes.

On the one hand, this spurious correlation, we believe, comes from the distribution of the data set. We found that videos of stones being thrown into water are abundant online, while videos of feathers being thrown into water are significantly less common. As supporting evidence, a search

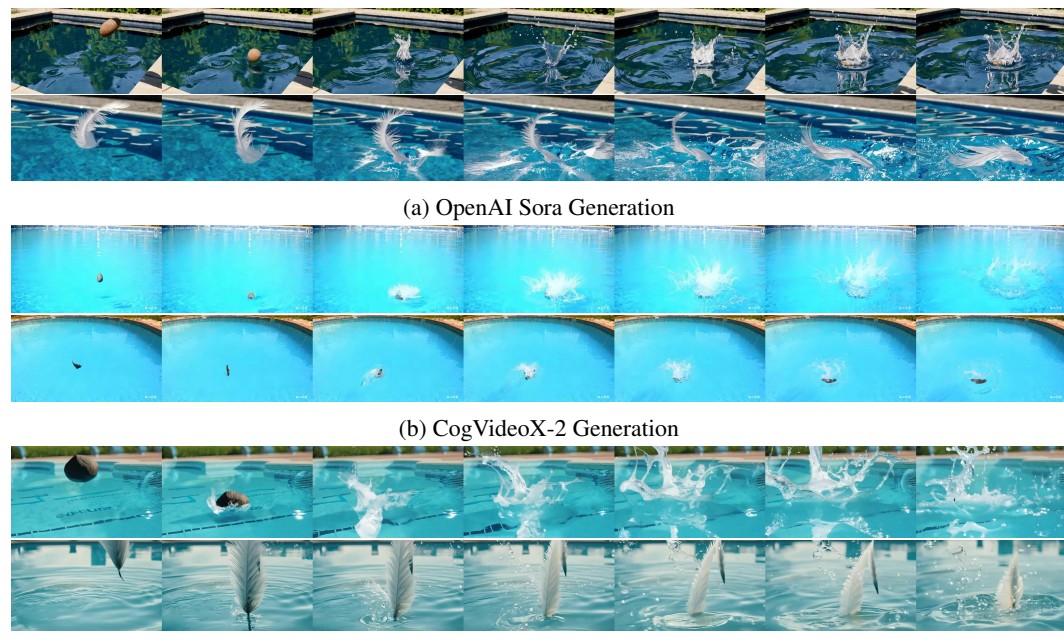

(a) OpenAI Sora Generation

(b) CogVideoX-2 Generation

(c) Gen-3 Alpha Generation

Figure 4: Videos generated by (a) OpenAI Sora, (b) CogVideoX-2 and (c) Gen-3 Alpha, shown as frames. For each model, the text prompt of the **Above** is: *a stone is thrown into a swimming pool*; **Below** is: *a feather is thrown into a swimming pool*. Both generation show *noticeable splashes*, which is correct for the above (stone) scene but **incorrect** for the **below** (feather) scene.

for "*thrown stone into water video*" returns approximately 180,000,000 results on google, while replacing "stone" with "feather" reduces the results to around 31,000,000. This data bias means that the model may have seen enough scenes of stones entering the water during training but not enough scenes of feathers doing the same. Additionally, this issue stems from the widespread overfitting of current VGM models, which causes them to rely heavily on common data in the dataset without fully understanding the underlying rules of the scene.

In contrast, the current *language* models like GPT-4o can more effectively grasp the different outcomes caused by various objects falling into the water. In this case, the language model can distinguish that feathers falling into the water will not cause splashes. See the following chat (where some excerpts have been made for ease of reading):

---

**Question**: Please detailed describe the scenario about "a stone is thrown into a swimming pool"

**Answer by GPT-4o**:

. . .

The stone experiences air resistance as it moves, but due to its density, air drag has little effect.

As the stone contacts the water, it displaces the liquid, creating an initial splash.

Water droplets are ejected in different directions, forming a dynamic spray pattern.

---

The surrounding water rushes in to fill the space left by the stone, leading to the formation of ripples.

The stone rapidly sinks due to its density being greater than water.

The initial impact generates concentric ripples that spread outward across the water's surface. If the throw was forceful, larger waves might be created, bouncing off the pool walls.

Over time, these waves dissipate due to friction and energy loss.

---

**Question**: Please detailed describe the scenario about "a feather is thrown into a swimming pool"

**Answer by GPT-4o**:

...

As it falls, it flutters down slowly, almost like it's floating through the air.

It lands gently on the surface of the water without much impact. The water ripples slightly where the feather touches, but there's no big splash.

The feather floats on the surface, moving slowly with the tiny waves or currents in the pool. It stays on the water, shifting around a little as the surface moves, but it doesn't sink or get pushed underwater.

---

It is obvious that current language models have clear and comprehensive understanding of the scenario and even the physical mechanism in the scenario. It is the reason why we believe the LLMs can be utilized to automatically generate the groundtruth causal rules of test cases.

## B   DETAILS OF AUTOMATIC GENERATION OF CAUSAL SYSTEMS

### B.1   DETAILS OF GENERATING PROCESS

We use the official API of OpenAI GPT-5 model (GPT-5-2025-08-07) (OpenAI, 2025) to generate the causal systems. The three tasks are divided and prompted sequentially, with the LLM completing them through multiple rounds of dialogue. Throughout this process, the entire dialogue history is retained within the context window. The model will proceed to the next task either once the maximum number of attempts is reached or when the external checks are passed and the LLM retains its answer after a self-check.

We require that the generated content for each step includes a file containing specific information, where:

- **Factor analysis**: a json file as a list of dictionary containing:
  - "`type`": choices from "factor" or "result".
  - "`name`": the name of the factor or result variable. They could be some words or a short sentence that can summarize the key meaning.
  - "`explanation`": A short explanation about how the factor or result can affect the scenario and why the variable is visible, binary and important. **Notice** that we find

that requiring the model not only generate the name list but also explain why the node is important and satisfies our requirement helps the output performance.

- **Causal Graph**: a dot file that constructs a digraph, which first declares each factor as a node, then declares some directed edges between nodes. We use the Python library "NetworkX" (Hagberg et al., 2008) to verify the format.

- **Causal System**: a json file as a list of dictionary containing:
  - "scenario": a string describing the event,
  - "roots": a list of strings, each of which is a name of cause variable,
  - "non_roots": a list of strings, each of which is a name of outcome variable,
  - "rules": a dictionary where each outcome variable corresponds to a Boolean function of its parents in the causal graph. The boolean function should be expressed as a disjunctive normal form (DNF), where each conjunctive clause are expressed as a dictionary ($A \land B \land \neg C$ expressed as {'A': True, 'B': True, 'C': False}. And the DNF is expressed as a list of the dictionary-expressed conjunctive clause.

The complete generation process consumes roughly 20k reading tokens (10k cached) and 10k prediction tokens, costing about \$0.74 per causal system. This is approximately one-third the cost of manual labeling, which is 15 CNY per annotation.

## B.2 REQUIREMENT: RULE-BASED & SELF CORRECTION

We have specific requirements for both the internal results and the final output causal systems. The detailed requirements can be found in the prompt in Appendix B.5. To ensure these requirements are met as thoroughly as possible, we have designed a check-and-correction loop.

### B.2.1 RULE-BASED CHECK

**Factor analysis:** In the step of factors analysis, we mainly use the self-check. We use the *"structured output"* function with *"pydantic"* to make sure the correctness of the output formulation.

**Causal graph:** First we also use *"structured output"* function with *"pydantic"*, where the generated graph is described as a set of edges. Then, we check the following requirements by a Python program:

- whether the graph is a DAG,
- whether the set of nodes is the same as the factors list proposed in the last step,
- whether there is an isolated node in the graph.
- If a "maximal number of nodes" is set, check the number of nodes.

**Causal system:** We first use a nested "pydantic" model to define our expected rules.

- "VariableValue" with two attributes: var and val,
- the Conjunction Clause: the conjunction clause [VariableValue(var='B', val=True), VariableValue(var='C', val=False)] represents $B \land \neg C$.
- a piece of Rule: a head factor and its disjunctive normal form as a list of conjunction clause,
- the Rules set (as a list of rules).

The "pydantic" library provides basic check for the output format. Then our additional rule-based check includes:

- Whether the *roots* and *non-roots* are consistent with the graph proposed in the last step,
- whether all the *non-roots* have exact one DNF and the *roots* do not have their DNF.
- whether the rules leads to the same causal graph generated in the "*causal graph*" step.

If any requirement has not been met, an error message will be the feedback to the LLM with the full history, and the LLM is required to regenerate its answer given the error message and the history information. Notice that if the LLM believes the error is caused by more previous steps, for example, it is the inappropriate graph leads to graph-rules inconsistency, the LLM can require to generate a new graph with all the history as input.

### B.2.2 SELF-CHECK

If the rule-based check has passed, we then prompt the LLM to further check its answer by itself. The self-check prompts repeat the requirement in a more concrete way. These prompts are shown in Appendix B.5.

Although the current reasoning models like OpenAI GPT-5 has learned to self-check during its thinking steps, we find the explicit self-check prompt can further help to improve the performance. We provide an example in Appendix B.3. We believe that it could be because in this step, an LLM can think in more detail about whether the answer satisfies the condition without having to take into account the generation task at the same time.

Considering that we have adopted a step-by-step strategy, we also allow the model to regret the previous answer in the subsequent steps. For example, when generating causal rules, if the model finds that the previous causal graph is unreasonable during the process, we allow the model to generate `<regenerate_graph>` to go back to the previous step. While this situation is rare, we have found that it effectively reduces the likelihood of the model producing low-quality answers.

We allow the model to generate `<keep_answer>` after self-checking. If this occurs, we skip the subsequent checking steps. We found that after a total of *three* checks, most requirements are satisfied, and the model is also typically satisfied with its answer, generating `<keep_answer>`. See the Table 7 showing the satisfying ratio after several check steps. The statistics shows that our checks did indeed provide additional thinking space to LLM, and that LLM did indeed modify past outputs. This demonstrates the importance of our process design. In the next section B.3, we provide some case study to further support this claim.

Table 7: The ratio of modification numbers caused by rule-based check and self-check.

| #Modifying | Factors | Graph | Rules | Regen Factors | Regen Graph |
|:---:|:---:|:---:|:---:|:---:|:---:|
| 0 | 39.3% | 67.9% | 96.4% | 39.3% | 92.9% |
| 1 | 32.1% | 21.4% | 3.6% | 28.6% | 3.6% |
| 2 | 10.7% | 3.6% | 0% | 32.1% | 3.6% |
| 3 | 17.9% | 7.1% | 0% | - | - |

### B.3 CASE STUDY OF RULE-BASED CHECK AND SELF-CHECK

In this section, we provide some case study to show that our rule-based or self-check provide useful modification and performance improvement. The Table 8 provides an example where our self-check improve the performance of LLM generated *factors*. LLM finds that the two factors *"end pinned down"* and *"seesaw not level"* are almost identity and so remove one. It is also interesting that LLM thinks about whether *"unequal weights"* is visible and suggest that it can be shown as visible size difference in the corresponding explanation. Notice that these explanations are also used in the following steps (to hint T2V prompt generation and VLLM answer extraction), so these self-check suggestions are useful for the latter steps.

Table 8: An example of self-check modification in factor analysis.

| Scenario | Two children of (different weights) are sitting on a seesaw. |
|:---|:---|
| Before self-check | [Unequal weights, feet contact present, end pinned down, seesaw not level] |
| After self-check | [Unequal weights, feet contact present, seesaw not level] |

Another example is show the graph self-check to remove the abundant links. As shown in Table 9, the LLM rethinks the graph and believe whether the stirring occurs does not affect the melts of chocolate but affect whether it is disperses, so it refine the graph to remove the link.

Table 9: An example of self-check modification in graph construct

| Scenario | Before | After |
|---|---|---|
| Chocolate melds into milk. | | |

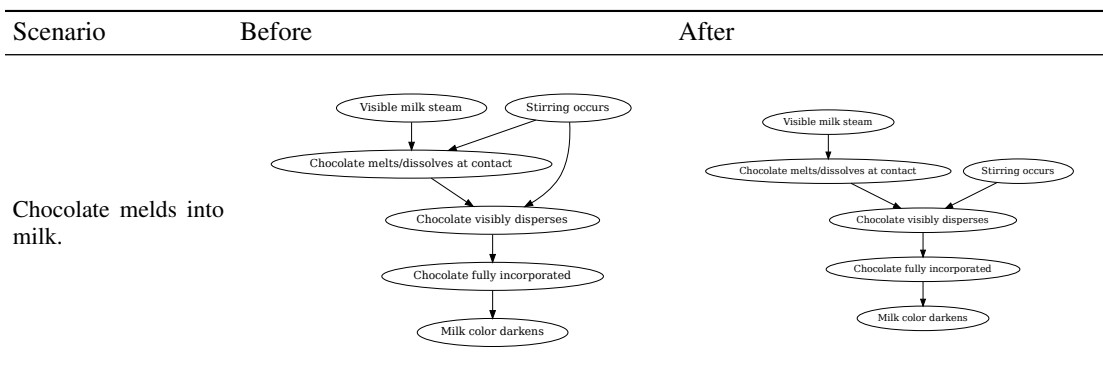

The "regenerating factors" and "regenerating graph" provides LLMs a choice to make some feedback to more previous steps. It is important that LLMs can reflect on and revise the results of previous steps based on problems encountered in subsequent steps. For example, in the scenario *"Chocolate melds into milk"*, the LLM firstly provide a factor *"chocolate finely divided"* in the first step but then find that this factor cannot deterministically define a relation, so that decide to remove it. The generated explanation of the refinement says:

> While particle size affects rates, it does not deterministically cause visible melting/dissolution at the interface across typical conditions without heat or agitation. Keeping only primary, deterministically influential, and clearly visible variables improves independence of roots and clarity of causal relations.

This is an example of how reflecting on subsequent steps can help modify preceding steps.

### B.4 CASE STUDY OF EXPLANATION

During data generation, LLMs are required not only to produce an answer but also to explain their answer piece by piece. The Table 10 is an example of proposed factors and their explanations. These explanations clearly discribe why LLMs think these factors satisfy the requirement (i.e., visibility, independence, importance ...). These explanations are further utilized in the subsequent steps. The probe questions (i.e., T/F questions which are used in step: *"answer retrieval"*) are heavily dependent on these explanation. For example, the question of factor *"Hard impact surface"* is that:

> Please determine whether the ground the cup is about to touch appears rigid and non-deformable based only on its visible surface characteristics before contact. Look for cues such as exposed tile/stone/concrete/wood/laminate, straight grout lines or board seams, sharp edges, and a smooth hard sheen, and the absence of any visible pile or cushioning like carpet or foam. Answer only based on what is directly visible in the video. If there is no clear evidence to answer this question, or if the view is unclear or incomplete, answer 'NaN'.

The orange parts are obviously similar. The detailed input questions can help VLM to read the video and retrieve more accurate answers for the video, considering now the reasoning ability of VLMs are generally weaker than reasoning language models.

### B.5 PROMPTS

In this section, we provide all of the prompts we use to facilitate the LLM to generate causal systems.

Prompt for Identifying Key Factors in a Scenario:

Table 10: Some examples of the explanation of the factors.

| Factor | Explanation |
|---|---|
| Hard impact surface | True if the cup's first contact is a visibly rigid, non-deformable floor (tile, stone, concrete, hardwood) with no soft layer; false if first contact is a clearly soft/deformable surface (carpet, thick rug, foam mat, soft grass/soil). Determine from first-contact frames by the surface appearance and the absence/presence of visible give or indentation. Visible, binary, and independent environmental property. Deterministic effect: if true, the outcome "Surface compresses at impact" is deterministically false; if false (soft), compression can be visibly present. |
| Cup is glass | True if the vessel clearly appears to be glass (transparent/clear, rigid, glossy specular reflections, refractive edges, no visible flex); false if it is clearly non-glass (opaque ceramic/metal or visibly flexible/seamed plastic). Determine from pre-impact visuals of the material and rim/body appearance. Satisfies the bracketed variable (glass). Visible, binary, and independent object property. Deterministic effect: if false (not glass), the outcome "Cup shards" is deterministically false. |
| Surface compresses at impact | True if the contacted surface visibly indents or compresses under the cup upon impact; false if no deformation is visible. Look for local squish/indentation of carpet/foam/grass in the contact area at the moment of impact. Directly observable outcome of the surface's compliance. Deterministically tied to "Hard impact surface": if the surface is hard, compression is not seen. |
| Cup shards | True if multiple separate small transparent/reflective fragments are visible on the ground after impact; false otherwise. Clear, visible fragmentation outcome. Deterministically impossible if "Cup is glass" is false. |

You will be provided with a brief description of a scenario. There could be some physical phenonmenon in this scenario. Please identify some **important** and **common** potential factors whose changes could significantly influence some important outcome of the scenario. These factors can fall into one of the following categories:

1. The objects or their properties in the scenario.

2. The object in the environment or the properties of the environment.

3. The actions or some properties of the action.

For each factor, ensure that it meets the following criteria:

1. It should be **visible** and easily recognizable in a video.

2. It should be **binary**, meaning it can be clearly labeled as either "yes" or "no", rather than a continuous value.

3. It should be **independent**, not dependent on other factors.

4. Its effect on the outcome should be **deterministic** (i.e., it directly leads to a certain result, rather than just increasing or decreasing the probability).

5. The resulting effect should also be **visible** in a video.

If there is a pair bracket in the description, it means the content in the bracket is expected to be a variable (factors or outcome). For example, "A (large) stone is thrown into a swimming pool (and splash water)." means we expect "does the water splash" as one of the outcome and whether the stone is large enough is expected as one of "factors". But notice that it does not mean that other factors or outcomes are not allowed, you can also propose other factors or outcomes.

Please organize your answer as a **json** file as a list of dict, where each dict is like { "type": "factor_or_result", "name": "factor_or_result_name", "explanation": "how it affects the scenario and why you believe it is important and common"}. Start your answer with a ⟨json⟩tag and end with a ⟨/json⟩tag.

Prompt for Causal Graph Construction:

Based on the factors you proposed and their expected results, generate a causal graph that summarizes the physical relationships between them. In the graph, include only the most important and common factors or results; omit any overly detailed or trivial ones.

The graph should be a **directed acyclic graph**, where:

- Each **node** represents a factor or a result.

- Each **edge** represents a direct causal relationship between two nodes.

The graph should be formatted in **DOT** format. Begin the DOT file with a ⟨dot⟩tag and end it with a ⟨/dot⟩tag.

Prompt for Causal Rule Generation:

Given the causal graph you generated, please create a Boolean expression for each **non-root** factor (factors with incoming edges) that represents the conditions under which that factor is **true**. The Boolean expression for each non-root factor should involve only the **parent factors** (i.e., the factors directly connected to it in the causal graph). The condition should be expressed as a **disjunctive normal form** (DNF), which is a disjunction (OR) of conjunctions (AND) of literals.

Your response should include a set of boolean expressions, formatted as a 'dict[str, list[dict[str, bool]]]', where the key is the name of this non-root factor and the value is a list of conditions (disjunctions), where each condition is a conjunction clauses (AND). Each condition is represented as a dictionary, where the key is the name of the parent factor and the value is a boolean value (True or False).

For example, if a factor A is true when B is true or (C is true and D is false), the boolean expression should be '{"A": [{"B": True}, {"C": True, "D": False}]}'.

Your final answer should be a JSON file with the following keys

- "roots": a list of root factors.

- "non_roots": a list of non-root factors.

- "rules": a dictionary where each non-root factor is associated with its corresponding Boolean expression.

Please begin your response with a ⟨json⟩tag and end with a ⟨/json⟩tag.

For self-check prompt for factors:

Please review the factors you have proposed. Ensure that each factor satisfies the following 5 requirements:

1. It should be **visible** and easily recognizable in a video.

2. It should be **binary**, meaning it can be clearly labeled as either "yes" or "no", rather than a continuous value.

3. It should be **independent**, not dependent on other factors.

4. Its effect on the outcome should be **deterministic** (i.e., it directly leads to a certain result, rather than just increasing or decreasing the probability).

5. The resulting effect should also be **visible** in a video.

Please ensure that the content in the bracket has been correctly identified as a variable (factor or outcome) in your answer.

Additionally, filter out any factors that are:

- **Too detailed**, **corner-case**, or **uncommon** in the scenario.

- Have an effect that is **too indirect** or difficult to understand.

If necessary, you may regenerate the factors to meet the criteria. It's OK to keep your previous answer by just generate ⟨keep_factor⟩ without any other words but you should carefully check every requirement for every factor and result.

For self-check prompt for graph:

Please review your causal graph. Ensure that it meets the following criteria:

1. All nodes are **visible** and **binary**.

2. All root nodes are **independent** of each other, which means the choice of one root node should not influence the choice of another root node.

3. All edges in the graph is a **direct** and **deterministic** causal relation

4. Include all **important** causes and results, while omitting trivial or overly detailed nodes.

Please ensure that the content in the bracket has been correctly identified as a variable (factor or outcome) in your answer.

If necessary, regenerate the causal graph to meet these requirements. It's OK to keep your previous graph if it already meets the criteria by just generate ⟨keep_graph⟩ without any other words but you should carefully check every requirement for every node and edge.

For self-check prompt for rules:

Please review your answer. Ensure your answer meets the following criteria:

1. The "roots" and "non_roots" list must be consistent with the causal graph.

2. For the bool expressions:

- All the nonroot factors are included in the rules dict, and no other factors are mistakenly included as keys.

- All variables in the Boolean expressions are exactly the parents of the corresponding non-root factors in the causal graph.

- The boolean expressions should correctly represent the physical rules in the real world.

If necessary, regenerate the json file to meet the requirements. It's OK to keep your previous rules if they already meet the criteria by just generate ⟨keep_rule_json⟩ without any other words but you should carefully check every requirement for every variable and rule.

If you find that you need to modify your generated causal graph, please generate ⟨regenerate_graph⟩⟨dot⟩... ⟨/dot⟩where the content between ⟨dot⟩and ⟨/dot⟩is the new causal graph.

## C    SCENARIOS IN CROWD EXPERIMENTS AND BENCHMARK

The scenarios used in our crowd experiments and benchmark evaluations are listed below. Each scenario contains approximately 4–7 variables. These scenarios differ in the types of relationships they involve, their complexity, and the extent to which they include variables. To simulate situations where users may already have specific variables of interest, we also designed a "bracket" representation to prompt the LLM, indicating that the content within the brackets must be treated as a variable.

1. A stone is thrown into a pool, (creating a splash).
2. A cat jumped onto the table.
3. A person strikes an ice block with a hammer.
4. Smoke spreads into the air from the chimney.
5. The billiard balls collide with each other on the table.
6. Drop dye into the water.
7. Two children of (different weights) are sitting on a seesaw.
8. Pour one liquid into another.
9. Flip the hourglass.
10. A person is walking on the snow.
11. Chocolate melds into milk.
12. The teapot pours hot tea into the cup.
13. A brush dips into watercolor on a palette.
14. A glass bowl rolls on the table.
15. Car jolting as it hits a pothole.
16. An air mattress floats on a pool.
17. The jack crank raises a car.
18. A snowball falls to the ground.
19. Pour the salt into the water.

We also show some LLM-generated examples of various relationships between variables on the above 19 scenarios. These examples illustrate the diversity and effectiveness of automatic generation.

In the scenario "A stone is thrown into a pool (creating a splash)," the LLM identifies key factors such as Steep entry angle, Impact adjacent to wall, and Calm water surface, along with outcomes Splash visible, Water hits wall/deck, Concentric ripples visible, and No surface skipping. It produces diverse relationships by considering entry geometry and environmental state: state-based links (a calm surface tends to produce concentric ripples), causal links (a steep entry angle leads to a visible splash; an impact near the wall leads to water hitting the wall or deck), and interaction effects (a steep entry angle combined with a calm surface yields a clear splash while avoiding surface skipping).

In the scenario "Car jolting as it hits a pothole," the LLM identifies key factors like "Left wheel enters," "Right wheel enters," and "Braking at impact," with outcomes "Left side drops," "Right side drops," "Upward rebound observed," and "Nose dive observed." It captures diverse behaviors: a left-wheel entry tilts the car to the left; a right-wheel entry tilts it to the right; one or both wheels entering compresses the suspension and then produces an upward rebound; braking at the moment of impact produces a noticeable nose dive. When both wheels enter, the rebound is stronger; when entry is asymmetric, the body rolls before recovering. These patterns illustrate the method's diversity and effectiveness in modeling vehicle dynamics.

# D  DETAILS OF CROWD EXPERIMENT

We conducted a crowd experiment to validate our automatic annotation of causal systems based on scenario descriptions. We first invite three undergraduates (2 from physics school and 1 from computer science school) to annotate the same 19 text scenarios. We provide them with the same requirements as we provided to LLM. We first check their annotation with first 5 attempts and then feedback some obvious misalignment with our requirement. We also instructed the annotators to avoid (1) referencing textbooks, as we wanted them to rely on commonsense rather than professional background knowledge, (2) using LLMs or other automatic annotation tools, to ensure their annotations reflected human intuition, and (3) communicating with each other to prevent bias. For human annotators, we prompted them to think in three steps similar to LLM; but we only collected the final rules. In order to ensure the seriousness of the annotators, we took a small number of samples and asked the annotators to explain their annotation reasons, which were checked by the authors. For the purpose of real comparison, we allowed a small number of non-systematic errors or deviations in the annotations — because this reflects the true level of human annotators.

These 57 annotations collected for the 19 scenario will be randomly shuffled together with the 57 annotations generated by LLM and given to five other annotators for scoring. The five annotators were also undergraduates (3 from computer science, 1 from mathematics, and 1 from economics).

The scoring standard we provide is:

- **Requirement**: whether the annotation meets all of our requirements including visibility, binary, and root node independence.
- **Rationality**: whether all the nodes in the causal system are consistent with public knowledge and common; and whether the most important factors and causal relations are included in the annotation.
- **Soundness**: whether all the rules in the causal graph are correct and definitive (from both physics and commonsense).

Each criterion is scored on a scale of 1-4, where

- 4: the annotation is completely correct (or meet the requirement),
- 3: there are minor errors,
- 2: there are obvious errors,
- 1: there are essential errors and the annotation needs to be rewritten.

The average scores have shown in Table 1 in the main paper. Here, we provide the detailed distribution of each scorer in Figure 5.

For "requirement" and "soundness", the LLM achieve excellent performance with a larger proportion of scores clustering around the top rating of 4 and the average score is significantly higher than human annotations. For rationality, the LLM- and human-annotation can not be clearly distinguished. The overall tendency of the five raters was consistent. Surprisingly, scorer 2 and 4 gave full marks of 4 points to all 57 items of LLM in requirement and soundness respectively.

Several examples highlight the reasons for the superior performance of the LLM in certain areas. Regarding the Requirement scores, the explicit guidelines provided in the prompt ensured that the LLM annotations generally met the requirements, resulting in consistently high scores. In contrast, human annotators occasionally failed to adhere to these requirements, either due to imprecise expressions or inadvertent oversights. For instance, in the scenario "Pour one liquid into another", one human annotator included the nodes "the densities of the liquids differ greatly" and "the chemical structures of the liquids are similar", both of which are unobservable factors. The LLM, however, avoided such missteps.

In terms of Soundness, where we require that the rules in the causal graph be both correct and definitive, human annotations displayed considerable variability across different scenarios. Some annotations included many nodes and rules, while others were sparse. In cases where a larger number of rules were included, human annotators sometimes overcomplicated their annotations, which led to errors. For example, in the scenario "A bullet is shot towards an object", a human annotator

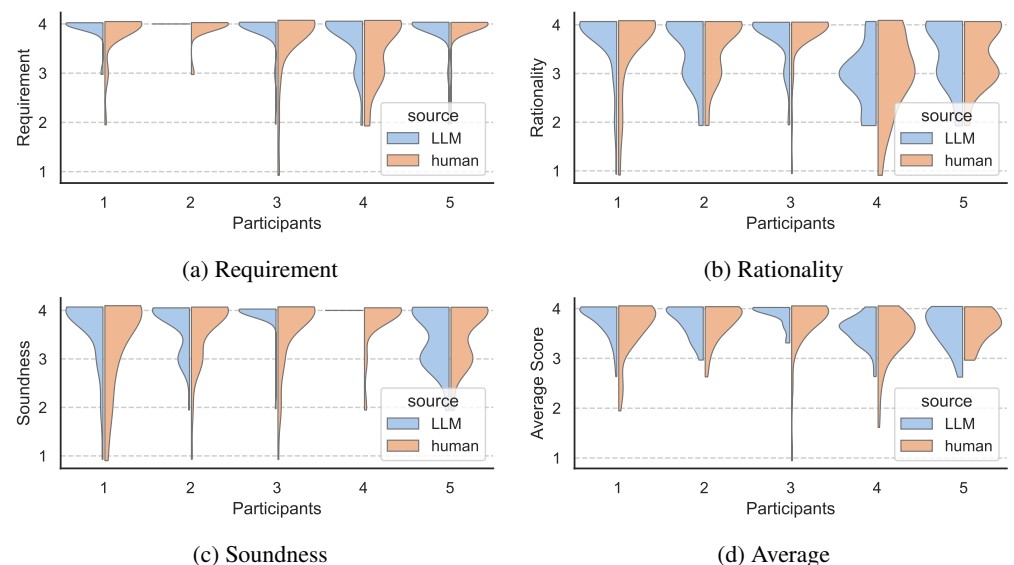

(a) Requirement

(b) Rationality

(c) Soundness

(d) Average

Figure 5: The violin plot as detailed distribution of 5 scorers. The width shows the number of the samples. The x-axis represents the 5 annotators.

included the rule (A bullet hole appeared on the back of the object)= (The bullet moves quickly) $\wedge$ (The object is hard). The increased complexity of the rule, while addressing multiple factors, led to inaccuracies. The LLM, by contrast, considered fewer factors and produced simpler, more accurate rules.

For the Rationality criterion, which required the inclusion of the most important factors and causal relationships, human annotators excelled in some scenarios but failed to fully account for relevant factors in others. This variability resulted in a broader distribution of scores, with a greater number of high and low ratings for human annotations. Overall, the performance of both human and LLM annotations in this category was similar.

We took great care to ensure that all annotations generated in this experiment adhered to ethical guidelines, ensuring that no violent, pornographic, discriminatory, or offensive content was included in the annotated scenarios. We ensure that the collected data does not contain any personal privacy information. To safeguard against potential ethical violations, we closely monitored the content throughout the annotation process and implemented a strict review mechanism. Additionally, all annotators were explicitly instructed on the importance of maintaining a respectful and non-harmful approach in their work.

In recognition of the effort and time invested by the annotators, they were compensated at a rate of 100 CNY per hour, which is in line with standard industry practices for similar tasks. This compensation not only reflects the value of their contributions but also ensures that the annotators were fairly incentivized for their participation in the study. Furthermore, we provided a feedback loop for annotators, encouraging them to express any concerns or challenges they faced during the annotation process, fostering an open and transparent working environment.

# E  DETAILS OF TEST PIPELINE

Here we introduce the details of the test pipeline. For step "prompt generation", see Appendix E.1. For step answer retrieval, see Appendix E.2 and Appendix E.3.

### E.1 DETAILS OF TEXT PROMPT GENERATION

Given a causal system as a test case, we need to generate some text prompts, which constrain the variable values in the scenario and are used to prompt the VGMs to generate corresponding videos. (In other words, they are used as the input of the tested T2V models.)

The step can be automated by an LLM. In this paper, we utilize the OpenAI GPT-5 (GPT-5-2025-08-07) to finish it. To reduce communication overhead, we adopt the strategy of generating first and then sampling from the generated sentences, which is slightly different from the one described in the pipeline. Specifically, we provide the LLM with the original sentence description of the scenario and the list of variables (roots and non-roots separately). We require the model to generate $m$ sentences for every $2^N$ possible combinations of $\mathbf{X}$ where $N = |\mathbf{X}|$. We find for most situation $N < 5$, the strategy of generating all value combinations at once is effective and works better than generating one value at a time. We observed that the former allows the model to consciously distinguish different values of $\mathbf{X}$. For cases where $N$ is too large, we take the approach of generating one value of $\mathbf{X}$ at a time. In our experiments, we set $m = 10$. In this setting, for each causal system, About 500 tokens are reading and 500 - 1000 tokens are generated by gpt-4o, costing about \$0.005.

The prompt we use in the step is shown as follows:

Prompt for sentence generation without results:

> You are a helpful assistant to generate corresponding short description about a scenario given some conditions. You will be provided with a short sentence to describe a scenario as well as some factors (variables) in the scenario. You should generate some short sentences which are slightly different from the originial sentence and describe the situation where the scenario is the same but the corresponding variables take given different value from original situation.
>
> The scenario is: {scenario}
>
> In this scenario, there are some factors are considered as (binary) variables and you should generate new description to change the original scenario to meet the corresponding value.
>
> Factors: {str(factors)}.
>
> There are also some results variables which are the outcome of the above factors: {non_roots}. The values of these variables should not be mentioned in the generated sentences.
>
> Each variable can take value as "yes" or "no" independently so that there are 2**{num_factors} = {num_comb} compositions. You should generate {num_sent} sentences for each yes/no composition for these variables.
>
> Please make sure (1) each sentence meet and explicitly express the corresponding value of variables and (2) the generated sentences as diverse as possible. Notice that you can add, delete or modify some words in original description to get the new sentence.
>
> Your answer should be following the schema provided. Here,
>
> - factors: The names of provides variables.
>
> - compositions: Samples for all compositions. It is a list (len = 2**{num_factors} = {num_comb}) where each element has two parameters:
>
> value: a list of bool. One-to-one correspondence with the values or the variables in the factors list.
>
> samples: a list contains the given number of generated sentences.

Prompt for sentence generation with results:

You are a helpful assistant to generate corresponding short description about a scenario given some conditions. You will be provided with a short sentence to describe a scenario as well as some factors (variables) in the scenario. You should generate some short sentences which are slightly different from the originial sentence and describe the situation where the scenario is the same but the corresponding variables take given different value from original situation.

The scenario is: {scenario}

In this scenario, there are some factors are considered as (binary) variables and you should generate new description to change the original scenario to meet the corresponding value.

Factors: {str(factors)}.

There are also some results variables which are the outcome of the above factors with their expected value: {non_roots}. In each possible composition of factor values, you should first induce the corresponding value of the results variables and then generate the sentences.

In these sentences, please explicitly and clearly express the corresponding value of both the factors and the results variables in the generated sentences. The rules of the results: {"\n".join(rules)}

Each variable can take value as "yes" or "no" independently so that there are 2**{num_factors} = {num_comb} compositions. You should generate {num_sent} sentences for each yes/no composition for these variables.

Please make sure (1) each sentence meet and explicitly express the corresponding value of variables and (2) the generated sentences as diverse as possible. Notice that you can add, delete or modify some words in original description to get the new sentence.

Your answer should be following the schema provided. Here,

- factors: The names of provided factor variables.

- results: The names of provided results variables.

- compositions: Samples for all compositions. It is a list (len = 2**{num_factors} = {num_comb}) where each element has three parameters:

value: a list of bool. One-to-one correspondence with the values or the variables in the factors list.

results: a list of bool. One-to-one correspondence with the values or the variables in the results list. Calculated by the given rules.

samples: a list contains the given number of generated sentences.

## E.2 DETAILS OF PROBE QUESTION GENERATION

We utilize GPT-5-2025-08-07 to generate questions for each variable. In a single conversation, we provide a short description of the scenario along with the factors that should be focused on. We instruct the model to generate questions for all root and non-root factors simultaneously. The prompt we design requires the model to generate a yes-no question for each factor in the scenario, ensuring that the questions are directly focused on the specific factor without incorporating any assumptions or conditions related to other factors.

To improve Answer Retrieval accuracy, we instruct GPT-5 to jointly produce, for each variable, the yes/no question and, simultaneously, a concise probe specifying the visual cues to attend to and the decision criteria for labeling "yes," "no," or "NaN."

For example, for the factor Ball is fragile, the probe may say: "Please determine whether there are clear visual cues that the ball is fragile. For example, you may judge fragility from: (1) the apparent material of the ball — if it looks like glass or thin ceramic it is fragile, while plastic, wood, or metal usually indicates it is not fragile; (2) the presence of visible cracks, chips, or surface lines that suggest breakability; (3) a glossy, brittle, or semi-transparent texture often associated with fragile

materials. Answer only based on what is directly visible in the video. If there is no clear evidence to answer this question, or if the view is unclear or incomplete, answer 'NaN'. Answer criteria: - Answer **TRUE** if the ball visibly shows fragile materials (e.g., glass, porcelain, crystal) or has cracks, chips, fracture lines, or a brittle/semi-transparent texture. - Answer **FALSE** if the ball visibly shows non-fragile materials (e.g., rubber, plastic, wood, metal) and there are no visible signs of cracks or brittleness. - If no relevant visual evidence is visible, the correct answer is **'NaN'**.

The prompt we use in this step is shown as follows:

Prompts for Probe Question Generation:

You are a helpful assistant to generate yes-no questions about some factors in a scenario. You will be provided with a short description of a scenario and some factors that should be focused on. You should generate **ONE** yes-no question for **EACH** of the factors in the scenario.

Detailed requirements for each question:

1. The question must ask the model to directly OBSERVE the factor in the video, not to infer or guess it from actions, effects, or indirect evidence.

2. Use a **clear imperative statement** starting with "Please determine whether...".

3. Each question must provide **explicit visual cues** that guide the judgment of the factor (e.g., material type, surface texture, cracks, transparency, color, shape).

- Focus only on the object's own visible characteristics.

- Do not use indirect evidence such as how the object behaves after impact, how other objects respond, or sound effects.

4. The question must explicitly remind: if the video is unclear, incomplete, or the factor cannot be observed, the correct answer is 'NaN'.

5. Do not mention or assume other factors. Only focus on the target factor itself.

6. Questions should be simple, clear, and concrete, but also detailed enough so the model knows it must rely only on visual evidence.

7. Do NOT include any specific time durations (e.g., 'within 3 seconds', 'for 5 frames').

8. If the factor's description explicitly states when to answer TRUE or FALSE, you must append a detailed statement at the end of the generated question. This statement should specify: (1) the visual cues or conditions to be checked; (2) the directly visible situations where the answer is TRUE; (3) the directly visible situations where the answer is FALSE.

If the factor involves prior actions conditions, you must explicitly identify and list them at the end of the question. Clearly separate them into two categories:

- Conditions that belong to the factor itself: if such a condition is not satisfied, the correct answer is FALSE.

- Conditions that do not belong to the factor but are assumed as prior or default requirements: if such a condition is not satisfied, the correct answer is 'NaN'.

All numeric time or frame references (e.g., "within 3 seconds", "for 2 seconds", "5 frames") must be rewritten as non-numeric temporal expressions such as "briefly", "quickly after contact", "continuously", or "sustained".

Examples:

Factor: "Ball is fragile" Question: "Please determine whether there are clear visual cues that the ball is fragile. For example, you may judge fragility from: (1) the apparent material of the ball — if it looks like glass or thin ceramic it is fragile, while plastic, wood, or metal usually indicates it is not fragile; (2) the presence of visible cracks, chips, or surface lines

that suggest breakability; (3) a glossy, brittle, or semi-transparent texture often associated with fragile materials. Answer only based on what is directly visible in the video. If there is no clear evidence to answer this question, or if the view is unclear or incomplete, answer 'NaN'. Answer criteria: - Answer **TRUE** if the ball visibly shows fragile materials (e.g., glass, porcelain, crystal) or has cracks, chips, fracture lines, or a brittle/semi-transparent texture. - Answer **FALSE** if the ball visibly shows non-fragile materials (e.g., rubber, plastic, wood, metal) and there are no visible signs of cracks or brittleness. - If no relevant visual evidence is visible, the correct answer is **'NaN'**. Prior action conditions: - This factor does not require any prerequisite action. "

Factor: "Ground is hard" Question: "Please determine whether the ground appears hard based only on its directly visible characteristics. For example, you may look for: (1) the material type, such as concrete, stone, or tile; (2) a rigid and non-flexible surface texture; (3) sharp edges, straight lines, or rough finishes typical of solid ground. Answer only based on what is directly visible in the video. If there is no clear evidence to answer this question, or if the view is unclear or incomplete, answer 'NaN'. Answer criteria: - Answer TRUE if the ground visibly appears to be hard material (e.g., stone, concrete, tile, brick) or shows surface features consistent with rigidity and solidity. - Answer FALSE if the ground visibly appears to be soft material (e.g., sand, soil, grass, fabric, rubber mat) or shows surface features consistent with flexibility or softness. - If the ground surface is not clearly visible, the correct answer is 'NaN'. Prior action conditions: - This factor does not require any prerequisite action; judgment depends only on the visible surface features of the ground."

Follow this style for all factors.

The scenario is {scenario}.

The factors are as follows, where the description tells the detailed definition of the variables and how to determine the Boolean value from a piece of video and the explanation tells how the variable affects the scenario.

The factors are: {factors}.

### E.3 DETAILS OF ANSWER RETRIEVAL

We tested two models to answer questions based on video content: Gemini and OpenAI GPT-5. Gemini has built-in video reading capabilities, extracting one frame per second for processing. In contrast, OpenAI GPT-5 can process multiple images, so we extract one frame every 10 frames from the video and provide these key frames to the model for question answering. Ultimately, we adopted OpenAI GPT-5 as the primary model for our experiments due to its superior performance.

For each video, we need to ask multiple questions. To ensure that the model relies strictly on the video content rather than commonsense or context, we explored two distinct questioning strategies. The first strategy involves asking one question at a time, ensuring the independence of each answer, though this approach incurs higher costs. The second strategy involves asking all the questions in a single round, within a single prompt. To avoid the model inferring subsequent questions based on prior answers or external commonsense, we topologically sort the nodes in the causal graph, ensuring that result variables are queried before cause variables. This method prevents the model from reasoning through previous answers when addressing subsequent questions. Additionally, we specify in the prompt that the model should answer based solely on the video.

For each question, we allow the model to respond with True, False, or N/A. Some videos suffer from lower generation quality, or fail to align with the textual descriptions, causing critical factors to be unobservable. In these cases, when the video does not provide enough evidence to answer the question, we allow the model to respond with N/A.

The prompt we use in this step is shown as follows:

Prompt for Video Analysis and Question Answering:

You are a professional video analysis expert, specialized in answering questions based on video content. Please answer the following questions based **strictly** on the video provided. Ensure that your response is based only on the video itself, and not on your own guesses or general knowledge.

Your answer must be one of: 'true', 'false', or 'N/A'. In addition, provide a brief explanation or evidence for your answer.

General Rules for Answering 'N/A':

1. The video quality is too low, or the content is too unclear to make any meaningful inference.

2. The video is not continuous or complete (temporal or spatial discontinuities prevent reasonable judgment).

3. The question asks about something that cannot be observed or recognized in the video.

4. The video does not provide enough context or evidence to form a conclusion.

5. The answer is unclear or could be interpreted in multiple ways, leading to ambiguity.

General Rules for Answering each question:

A. When the question is about an action

- Answer 'true' if the action clearly happens in the video (confirmed by continuous frames, not just one frame).

- Answer 'false' if the subject is present, the necess

- Answer 'N/A' if: - The subject of the action is not present or is too unclear to identify. *Example*: "Does the ball roll away after impact?" → If no ball exists, or it is unclear whether the object is a ball, answer 'N/A'.

you should refer to some continuous frames to make sure the action is happening, instead of just one frame.

B. When the question is about an object's existence

- Answer 'true' if the object is clearly observed in the video.

- Answer 'false' if the object is not observed in the video.

- Answer 'N/A' only if the video quality is too low or unclear to determine whether the object is present.

C. When the question is about an object's feature

- Judge based only on the object itself (its shape, material, appearance, etc.), not on indirect evidence such as collisions, rebounds, or interactions.

- Answer 'true' if there is clear evidence the object has the feature.

- Answer 'false' if there is clear evidence the object does not have the feature, **or if the feature is clearly absent from the object**.

- Answer 'N/A' only if the video quality is too low, the object is too unclear, or there is no sufficient visual evidence to make a judgment.

- If you believe you can reasonably conclude 'true' or 'false', you should not answer 'N/A'. Default to 'false' if the feature is visibly absent rather than uncertain.

Independence of Judgments:

Each question must be judged independently. Do not use the answer or content of other questions as evidence for the current one.

Return the question field as an exact verbatim copy of the input question.

Please pay attention to the instructions in each question, refer to the answer criteria, and if prior action conditions are provided, follow them: for conditions that belong to the factor, if not satisfied answer 'fasle'; for prior/default requirements, if not satisfied answer 'N/A'. Based on the above guidelines, please answer the following questions:

"\n".join({questions})

## F  DETAILED DEFINITION FOR METRICS

In this subsection, we give a detailed definition for our proposed metrics in Section 4.

First we review the definitions and symbols. Let $\mathbf{V}$ be a set of variables representing all factors of interest in a causal system. Let $G$ a directed acyclic graph with node set $\mathbf{V}$ and edge set $\mathbf{E}$. For every $V_j \in \mathbf{V}$, let $pa(V_j) = \{V_k \in \mathbf{V} : V_k \to V_j \in \mathbf{E}\}$ be the set of nodes that has a directed edge pointing to $V_j$. Suppose there is a deterministic structural equation model over $\mathbf{V}$. That is, for every $V_j \in \mathbf{V}$ such that $pa(V_j) \neq \emptyset$, there exists a function $f_j$ such that $V_j = f_j(pa(V_j))$. Denote $\mathbf{X} = \{V_j \in \mathbf{V} : pa(V_j) = \emptyset\}$ and $\mathbf{Y} = \mathbf{V} \setminus \mathbf{X}$. We also write $\mathbf{X} = (X_1, X_2, \ldots, X_{m_1})$ and $\mathbf{Y} = (Y_1, Y_2, \ldots, Y_{m_2})$ as random vectors. Then $\mathbf{X}$ is called the set of root (or cause) variables, and $\mathbf{Y}$ is called the set of non-root (or outcome) variables. In structural equation $Y_j = f_j(pa(Y_j))$ for every $Y_j \in \mathbf{Y}$, the function $f_j$ is called the rule of $Y_j$. The structural equations can be equivalently represented as $\mathbf{Y} = f(\mathbf{X})$. Since the value of non-root variables is determined by root variables, we also write $Y_j = f'_j(\mathbf{X})$ for every $Y_j \in \mathbf{Y}$. Let $D(\mathbf{X}) = \{1, 0\}^{|\mathbf{X}|}$ denote the domain of $\mathbf{X}$, that is, the set of all possible values of $\mathbf{X}$.

In our pipeline, we use a large language model for generating prompt from the given causal system and specified variables, a video generation model for generating video from the prompt, and an multi-modale LLM for retrieving the value of variables from the video. For specified $\mathbf{X}, \mathbf{Y}$, let $f_P(\mathbf{X}, \mathbf{Y})$ denote the generated prompt under the given causal system, with specifying both $\mathbf{X}$ and $\mathbf{Y}$. Let $f_P(\mathbf{X})$ denote the generated prompt under the given causal system with only specifying only $\mathbf{X}$. Note that $f_P$ includes an independent error $\varepsilon_P$ implicitly, so it is not a deterministic function of $\mathbf{X}$ and $\mathbf{Y}$. For a prompt $P$, let $f_V(P)$ denote the video generated by video generation model with prompt $P$. Finally, let $\hat{\mathbf{X}}, \hat{\mathbf{Y}} = f_A(f_V(P))$ denote the **observation** of all variables from the generated video. For simplicity, we also write $\hat{\mathbf{X}}, \hat{\mathbf{Y}} = f_V(P)$. In this situation, we also call $\mathbf{X}, \mathbf{Y}$ the **ground truth**. For the $i$-th sample, let $\mathbf{X}^{(i)}, \mathbf{Y}^{(i)}$ denote the ground truth and $\hat{\mathbf{X}}^{(i)}, \hat{\mathbf{Y}}^{(i)}$ denote the observation. For any $V \in \mathbf{V}, X \in \mathbf{X}$ and $Y \in \mathbf{Y}$, we use $V^{(i)}, X^{(i)}, Y^{(i)}$ or $\hat{V}^{(i)}, \hat{X}^{(i)}, \hat{Y}^{(i)}$ to denote the corresponding component of $\mathbf{X}^{(i)}, \mathbf{Y}^{(i)}$ or $\hat{\mathbf{X}}^{(i)}, \hat{\mathbf{Y}}^{(i)}$, just as we use $V, X, Y$ to denote the corresponding component of $\mathbf{X}, \mathbf{Y}$. We also use $V_j^{(i)}$ to denote the component $V_j$ in vector $\mathbf{V}^{(i)}$. For variable $Y_j \in \mathbf{Y}$, we use $\hat{pa}(Y_j)$ to denote the observed value of $pa(Y_j)$.

### F.1  TEXT CONSISTENCY

For text consistency, let $\mathbf{X}^{(1)}, \mathbf{X}^{(2)}, \ldots, \mathbf{X}^{(n_1)}$ be $n_1$ samples that are i.i.d. are uniform distributed over $D(\mathbf{X}) = \{1, 0\}^{|\mathbf{X}|}$. Let $\mathbf{Y}^{(i)} = f(\mathbf{X}^{(i)})$ for $i = 1, 2, \ldots, n_1$.

Since we have specified the value of every variable in the prompt, we expect that the value of every observed variable matches with its ground truth. However, due to the internal causal mechanism in the video generation model, the value of outcome variables in the video may be influenced by the value of root variables in the video. Therefore, we propose two versions of metric: $s_1^{\text{all}}$ by comparing the observed value of all variables with their ground truth, and $s_1^{\text{roots}}$ by comparing the observed value of only root variables with their ground truth. For $s_1^{\text{roots}}$, we generate prompt $P^{(i)} = f_P(\mathbf{X}^{(i)})$ by specifying only root variables, and for $s_1^{\text{all}}$, we generate prompt $P^{(i)} = f_P(\mathbf{X}^{(i)}, \mathbf{Y}^{(i)})$ by specifying both $\mathbf{X}^{(i)}$ and $\mathbf{Y}^{(i)}$. Finally, we get observation $\hat{\mathbf{X}}^{(i)}, \hat{\mathbf{Y}}^{(i)} = f_{VA}(P^{(i)})$ by generating video from prompts and asking questions from videos.

The metrics for text consistency is defined as:

$$s_1^{\text{all}} = \frac{1}{n_1 |\mathbf{V}|} \sum_{i=1}^{n_1} \sum_{V \in \mathbf{V}} \mathbb{1}(V^{(i)} = \hat{V}^{(i)}), \tag{1}$$

and

$$s_1^{\text{roots}} = \frac{1}{n_1|\mathbf{X}|} \sum_{i=1}^{n_1} \sum_{X \in \mathbf{X}} \mathbb{1}(X^{(i)} = \hat{X}^{(i)}), \tag{2}$$

where $\mathbb{1}(\cdot)$ denotes the indicator function.

## F.2   GENERATION CONSISTENCY

For generation consistency, we construct some groups of samples. Samples within the same group should have the same ground truth. Therefore, by comparing observations within the same group, we can test whether generations for the same ground truth are consistent.

Formally, let $\mathbf{x}^{(1)}, \mathbf{x}^{(2)}, \ldots, \mathbf{x}^{(n_2)}$ be $n_2$ different values that are randomly selected from $D(\mathbf{X}) = \{1, 0\}^{|\mathbf{X}|}$. We construct $n_2$ groups, with $r$ samples in each group, that is, letting

$$\begin{aligned} \mathbf{X}^{(1)} = \cdots = \mathbf{X}^{(r)} = \mathbf{x}^{(1)}, \\ \cdots \\ \mathbf{X}^{((n_2-1)r+1)} = \cdots = \mathbf{X}^{(n_2 r)} = \mathbf{x}^{(n_2)}. \end{aligned} \tag{3}$$

For $i = 1, 2, \ldots, n_2 r$, let $P^{(i)} = f_P(\mathbf{X}^{(i)})$ be the generated prompt and $\hat{\mathbf{X}}^{(i)}, \hat{\mathbf{Y}}^{(i)} = f_{VA}(P^{(i)})$ be the observation.

To measure the inconsistency of observations within a group, we propose two versions of metric: $s_2^{\text{truth}}$ and $s_2^{\text{observe}}$. For $s_2^{\text{truth}}$, we assume that text consistency holds, that is, observation of root variables should remains the same within each group. Therefore, we compare all variables for each group. For $s_2^{\text{observe}}$, we allow for observation of root variables to be different within each group. Relatively, we see the observed root variables as the truth understood by the video generation model. So we reconstruct the groups by partitioning the samples by $\hat{\mathbf{X}}^{(i)}$, and compare the observed outcome variables within each group.

Formally, for an index set $\mathbf{S} \subseteq \{1, 2, \ldots, n_2 r\}$ and variable $V \in \mathbf{V}$, denote $\bar{V}_\mathbf{S} = \frac{1}{|\mathbf{S}|} \sum_{i \in \mathbf{S}} \hat{V}^{(i)}$ be the mean, and $d(V, \mathbf{S}) = \frac{1}{|\mathbf{S}|} \sum_{i \in \mathbf{S}} \left( \hat{V}^{(i)} - \bar{V}_\mathbf{S} \right)^2$ be the sample variance of $V$ in subgroup $\mathbf{S}$. For group index $k = 1, 2, \ldots, n_2$, let $\mathbf{S}_k = \{(k-1)r+1, (k-1)r+2, \ldots, kr\}$ be the index of samples within group $k$. Then we have

$$s_2^{\text{truth}} = \frac{1}{n_2|\mathbf{Y}|} \sum_{k=1}^{n_2} \sum_{Y \in \mathbf{Y}} d(Y, \mathbf{S}_k). \tag{4}$$

For definition of $s_2^{\text{observe}}$, for each $\mathbf{x} \in D(\mathbf{X})$, let $\mathbf{S}_\mathbf{x} = \{i : \hat{\mathbf{X}}^{(i)} = \mathbf{x}\}$, and let $\mathcal{S} = \{\mathbf{S}_\mathbf{x} \neq \emptyset : \mathbf{x} \in D(\mathbf{X})\}$. Then we have

$$s_2^{\text{observe}} = \frac{1}{|\mathbf{Y}||\mathcal{S}|} \sum_{Y \in \mathbf{Y}} \sum_{\mathbf{S}_\mathbf{x} \in \mathcal{S}} d(Y, \mathbf{S}_\mathbf{x}). \tag{5}$$

## F.3   RULE CONSISTENCY

For rule consistency, we generate samples for each outcome variable independently. For each $Y_j \in \mathbf{Y}$, let $\mathbf{S}_j^T = \{\mathbf{x} \in D(\mathbf{X}) : f_j'(\mathbf{x}) = 1\}$ be the set of values of $\mathbf{X}$ that making $Y_j = f_j'(\mathbf{X}) = 1$, and let $\mathbf{S}_j^F = D(\mathbf{X}) \setminus \mathbf{S}_j^T$. Then for ground truth $\mathbf{X}$ and $\mathbf{Y} = f(\mathbf{X})$, we have $Y_j = 1$ if and only if $\mathbf{X} \in \mathbf{S}_j^T$.

To test whether the video generation model has learned this rule, we draw $n_3$ samples $\mathbf{X}^{(1)}, \mathbf{X}^{(2)}, \ldots, \mathbf{X}^{(n_3)}$ uniformly from $\mathbf{S}_j^T$, and $n_3$ samples $\mathbf{X}^{(n_3+1)}, \mathbf{X}^{(n_3+2)}, \ldots, \mathbf{X}^{(2n_3)}$ uniformly from $\mathbf{S}_j^F$. Comparing to drawing sample uniformly from $D(\mathbf{X})$, this sampling method avoids the bias that may arise when $|\mathbf{S}_j^T|/|\mathbf{S}_j^F|$ is near 0 or 1. For $i = 1, 2, \ldots, 2n_3$, let $P^{(i)} = f_P(\mathbf{X}^{(i)})$ be the generated prompt and $\hat{\mathbf{X}}^{(i)}, \hat{\mathbf{Y}}^{(i)} = f_V(P^{(i)})$ be the observation.

We also propose two versions of metrics for rule consistency, $s_3^{\text{truth}}$ and $s_3^{\text{observe}}$. For $s_3^{\text{truth}}$, we assume that text consistency holds, and check whether the value of observed outcome variables

matches its ground truth. For $s_3^{\text{observe}}$, we see the observed parents of each outcome variabe as the truth understood by the video generation model. Therefore, we calculate the value of outcome variables from the rules and its observed parents, and compare them with observed outcome variables. Remind that $Y_j^{(i)}$ denotes the component $Y_j$ in vector $Y^{(i)}$. Formally, we have

$$s_3^{\text{truth}}(Y_j) = \frac{1}{2n_3} \sum_{i=1}^{2n_3} \mathbb{1}\left(Y_j^{(i)} = \hat{Y}_j^{(i)}\right), \tag{6}$$

and then

$$s_3^{\text{truth}} = \frac{1}{|\mathbf{Y}|} \sum_{Y_j \in \mathbf{Y}} s_3^{\text{truth}}(Y_j). \tag{7}$$

For $s_3^{\text{observe}}$, we propose a strategy to rebalance samples such that the expected value of $Y_j$, $f_j(\hat{pa}(Y_j))$, has equal weights over $\{0, 1\}$. Let $\hat{pa}^{(i)}(Y_j)$ denote the observed value of parents of $Y_j$ in the $i$-th sample $\mathbf{Y}^{(i)}$. Then, denote $g_j = \sum_{i=1}^{2n_3} f_j(\hat{pa}^{(i)}(Y_j))$ as the total number of samples such that the expected value of $Y_j$ is 1, then we reweight each sample and define $s_3^{\text{observe}}(Y_j)$ as

$$s_3^{\text{observe}}(Y_j) = \frac{1}{2} \sum_{i=1}^{2n_3} \mathbb{1}\left(\hat{Y}_j^{(i)} = f_j(\hat{pa}^{(i)}(Y_j))\right) \cdot$$
$$\left(\frac{f_j(\hat{pa}^{(i)}(Y_j))}{g_j} + \frac{1 - f_j(\hat{pa}^{(i)}(Y_j))}{2n_3 - g_j}\right), \tag{8}$$

and then

$$s_3^{\text{observe}} = \frac{1}{|\mathbf{Y}|} \sum_{Y_j \in \mathbf{Y}} s_3^{\text{observe}}(Y_j) \tag{9}$$

For revealing more intuition under the evaluation of rule consistency, we also define the threshold-based metrics for rule consistency. Let $t \in (0, 1)$ denote the threshold, the threshold-based metrics are defined as

$$s_3^{\text{truth,threshold}}(t) = \frac{1}{|\mathbf{Y}|} \sum_{Y_j \in \mathbf{Y}} \mathbb{1}\left(s_3^{\text{truth}}(Y_j) \geq t\right), \tag{10}$$

$$s_3^{\text{observe,threshold}}(t) = \frac{1}{|\mathbf{Y}|} \sum_{Y_j \in \mathbf{Y}} \mathbb{1}\left(s_3^{\text{observe}}(Y_j) \geq t\right). \tag{11}$$

These metrics measures the probability that for a given causal rule, the model gives a correct value for the outcome variable corresponding to this rule with an accuracy over the threshold.

### F.4 SAMPLE STRATEGY FOR THREE-LEVEL METRICS

We propose a unified sampling framework designed to optimize sample efficiency across different evaluation metrics. First, we perform sampling for each metric. Specifically, for Metric 1: text consistency, we collect $n_1$ samples, where the $\mathbf{X}$ values are uniformly random from the set $D(\mathbf{X}) = \{1, 0\}^{|\mathbf{X}|}$. For Metric 2: generation consistency, we collect $n_2$ groups, each containing $r$ samples with the same $\mathbf{X}$ value. For Metric 3: rule consistency, for each $Y_j \in \mathbf{Y}$, we collect $n_3$ samples from the positive set $\mathbf{S}_j^T$ and the negative set $\mathbf{S}_j^F$, respectively. During each sampling step, we record the number of samples corresponding to different $\mathbf{X}$.

With the separate sampling results, we construct a total sample set, where for each possible $\mathbf{X}$ value, the sample count is the **maximum** across the three metrics. While each sample may be used multiple times to compute different metrics or different rule accuracies for $Y_j$, within the same metrics (or within metric 3 for the same $Y_j$), each sample is used only once. The framework ensures that no sample is reused within the calculation of any single metric. By doing so, we maintain the independent and identically distributed (IID) conditions for sampling, while preserving the integrity of each metric's evaluation criteria. The architecture also achieves significant storage efficiency, reducing redundancy compared to traditional independent sampling approaches, without compromising the statistical validity of the results. Finally, we use the total sample set to select the corresponding text prompts and generate videos.

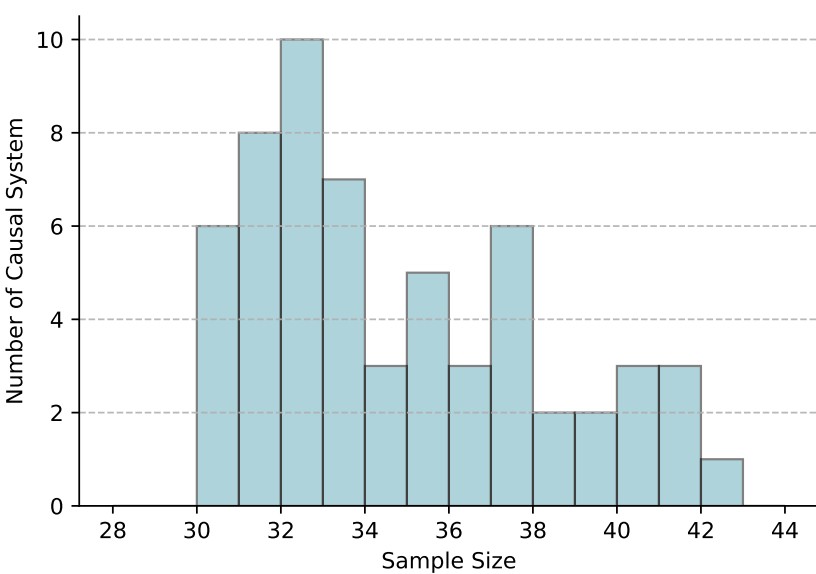

Figure 6: Distribution of sample sizes over causal systems.

In our benchmark, we set the parameters as follows: $n_1 = 10$, $n_2 = 5$, $n_3 = 10$, and $r = 3$. Using these values, we apply our strategy to draw samples. Appendix H.4 demonstrate that this sample size is sufficient for distinguishing between metrics across different models. Specifically, we draw $n_1$ samples for the evaluation of text consistency, $n_2 r$ samples for the evaluation of generation consistency, and $2n_3|\mathbf{Y}|$ samples for the evaluation of rule consistency. In contrast, without this strategy, a total of $N = 25 + 20|\mathbf{Y}|$ samples would be required for each causal system, which could significantly increase computational costs. The distribution of sample sizes for each causal system is depicted in Figure 6, which illustrates a considerable reduction in the number of samples needed by our approach.

### F.5 Sample-based Scores

Our metrics can also be applied to each sample, showing how each sample contributes to the evaluation. The definitions are as follows.

For text consistency, the metrics for a sample $i$ are defined as

$$s_{1,i}^{\text{all}} = \frac{1}{|\mathbf{V}|} \sum_{V \in \mathbf{V}} \mathbb{1}(V^{(i)} = \hat{V}^{(i)}), \tag{12}$$

and

$$s_{1,i}^{\text{roots}} = \frac{1}{|\mathbf{X}|} \sum_{X \in \mathbf{X}} \mathbb{1}(X^{(i)} = \hat{X}^{(i)}). \tag{13}$$

For the sake of sample efficiency, samples with same ground truth $\mathbf{X}$ are reused in testing generation consistency and rule consistency. Let $i$ be the index of a sample in a group $\mathbf{S}$. Similarly, let $\bar{V}_{\mathbf{S}} = \frac{1}{|\mathbf{S}|} \sum_{i \in \mathbf{S}} \hat{V}^{(i)}$ be the mean of observed values for variable $V \in \mathbf{V}$. Then the metrics for generation consistency on the $i$-th sample are defined as

$$s_{2,i}^{\text{truth}} = \frac{1}{|\mathbf{Y}|} \sum_{Y \in \mathbf{Y}} (\hat{Y}^{(i)} - \bar{Y}_{\mathbf{S}_k})^2, \tag{14}$$

and

$$s_{2,i}^{\text{observe}} = \frac{1}{|\mathbf{Y}|} \sum_{Y \in \mathbf{Y}} (\hat{Y}^{(i)} - \bar{Y}_{\mathbf{S}_\mathbf{x}})^2, \tag{15}$$

where $\mathbf{S}_k$ and $\mathbf{S}_\mathbf{x}$, as defined in Appendix F.2, are groups which contain the $i$-th sample.

For rule consistency, samples are reused so that some samples are contained in the test samples for multiple outcome variables. Let $i$ be the index of a sample. Write $\mathbf{Y} = (Y_1, Y_2, \ldots, Y_{m_2})$, and let $\mathbf{Z}^{(i)} \subseteq \{1, 2, \ldots, m_2\}$ be the index of all outcome variables whose test samples contains the $i$-th sample. Then the metric $s_{3,i}^{\text{truth}}$ for the $i$-th sample is

$$s_{3,i}^{\text{truth}} = \frac{1}{|\mathbf{Z}^{(i)}|} \sum_{j \in \mathbf{Z}_i} \mathbb{1}\left(Y_j^{(i)} = \hat{Y}_j^{(i)}\right). \tag{16}$$

For the definition of $s_{3,i}^{\text{observe}}$, we also rebalance the samples such that the expected value of $Y_j$, $f_j(\hat{pa}(Y_j))$, has equal weights over $\{0, 1\}$. Recall that $g_j = \sum_{i=1}^{2n_3} f_j(\hat{pa}^{(i)}(Y_j))$ is the total number of samples such that the expected value of $Y_j$ is 1. We define the weight of $Y_j$ in calculating the metric as

$$w_j = n_3 \left( \frac{f_j(\hat{pa}^{(i)}(Y_j))}{g_j} + \frac{1 - f_j(\hat{pa}^{(i)}(Y_j))}{2n_3 - g_j} \right). \tag{17}$$

Then the metric $s_{3,i}^{\text{observe}}$ for sample $i$ is defined as

$$s_{3,i}^{\text{observe}} = \frac{\sum_{j \in \mathbf{Z}_i} w_j \mathbb{1}\left(\hat{Y}_j^{(i)} = f_j(\hat{pa}^{(i)}(Y_j))\right)}{\sum_{j \in \mathbf{Z}_i} w_j}. \tag{18}$$

## G  MANUAL VERIFICATION OF AUTOMATIC RESULTS

For automatic annotation of causal systems, we have verified the effectiveness through crowd experiments. Here we verify other automatic steps, including:

- Section G.2: generating text prompts from value combinations,
- Section G.3: generation probe questions from factors,
- Section G.4: retrieve observed value from videos.

For each step, we randomly choice some automatic generation in our test cases from our VACT benchmark and manually check whether the automatic annotation is correct.

### G.1  METRICS NOISE BOUND CAUSED BY ERROR FROM LLM/VLLM COMPONENTS

In Table 3, we report the sampled accuracy of LLM/VLLM components of our pipeline. To give a sense of how the audited accuracies estimate the noise in our metrics, consider a simple symmetric-noise model. Let us assume a most strict error accumulation situation, where the incorrect in each component will leads to incorrect factor value in the final retrieved answer and at least one error of factor value leads to the error of the rule.

First, let us estimate the error of the level one score *text consistency* and level three score *rule consistency*. Notice that these two scores are both factor or rule based average accuracy. Let $s_{\text{true}}$ denote the (unobserved) true text or rule consistency of a video generator, and used the sampled accuracy $p_{\text{prompt}} = 0.98$, $p_{\text{probe}} = 1$ and $p_{\text{retrieve}} = 0.96$ as estimation, the absolute error can be directly bound by add the error ratio of each component up, where the upper bound is $6\%$, and has similar scaling of the sample variance. If we estimate the error by using an independent setting, where we assume the error caused by parsing are independent of the actual errors in the model itself, we have another estimation:

If the VLLM flips the binary label with probability $1 - p_{\text{(V)LLM}}$ independently of $s_{\text{true}}$, then the observed score satisfies

$s_{\text{obs}} = \Pr(\hat{Y} = 1) = p_{\text{(V)LLM}} s_{\text{true}} + (1 - p_{\text{(V)LLM}})(1 - s_{\text{true}}) = (1 - p_{\text{(V)LLM}}) + (2p_{\text{(V)LLM}} - 1) s_{\text{true}}.$

Plugging in $p_{\text{(V)LLM}} \approx 0.94$ gives

$$|s_{\text{obs}} - s_{\text{true}}| \approx 0.06(1 - 2s_{true}),$$

which will further smaller than $6\%$.

## G.2 MANUAL VERIFICATION OF PROMPT CORRECTNESS

Below are randomly selected scenarios and corresponding prompts from our dataset. We manually verify the correctness by checking whether the two types of prompts (with and without non-root nodes ) match the given values for the variables.

We examined a total of four scenarios and their corresponding 106 prompts. Nearly all of the prompts passed inspection, with the exception of **two**. The issue with the first prompt arises from our setting the variable "*sponge is wet*" to false, while the prompt only specifies that the hand is dry and fails to clarify the condition of the sponge. The second issue pertains to a prompt that was expected to contain only root nodes; however, it includes the word "*slide*", which introduces a non-root value.

We show these 106 samples as follows, where we marked the correct ones with ✓ and the 2 incorrect results with ✗.

Scenario:

---

Rules for each non-root node:

Pile Catches Fire = (Ball Actively Burning ∧ Ball Contact Pile)

$X_1$ = Ball Actively Burning     $X_2$ = Ball Contact Pile     $Y$ = Pile Catches Fire

---

Prompts:

---

Without non-root nodes:

$(X_1 = \text{False}, X_2 = \text{True})$

    A smoldering paper ball, now unlit, was tossed but missed the stack of old newspapers.✓

    A ball of paper that had stopped burning eventually settled into a stack of paper.✓

    A cooling ball of paper was placed carelessly into a mound of papers.✓

    An extinguished paper sphere was accidentally dropped into a heap of documents. ✓

    A barely expired paper ball softly landed in a collection of scraps. ✓

$(X_1 = \text{True}, X_2 = \text{True})$

    A flaming ball of paper crashed into a stack of old newspapers. ✓

    A lit paper ball was hurled into a heap of documents.✓

    A burning paper sphere landed directly in a pile of loose-leaf papers. ✓

    A fireball of paper was tossed straight into a mound of papers. ✓

    An ignited ball of paper rolled into a collection of scraps. ✓

$(X_1 = \text{True}, X_2 = \text{False})$

    A burning ball of paper was thrown close to but missed hitting a pile of paper. ✓

    A flaming ball of paper flew past a stack of old newspapers without making contact. ✓

    A lit paper ball was launched near a heap of documents, but it didn't touch them. ✓

$(X_1 = \text{False}, X_2 = \text{False})$

    A ball of paper, which had extinguished, was thrown away from a pile of paper.✓

    A smoldering paper ball, now unlit, was tossed but missed the stack of old newspapers.

✓

---

A once aflame ball of paper, now out, was hurled and did not touch the paper heap. ✓

With non-root nodes:

($X_1$ = False , $X_2$ = False, $Y$ = False)

An unlit ball of paper passed by the paper pile without touching it, leaving the pile unburned.✓

($X_1$ = False , $X_2$ = True, $Y$ = False)

Since the ball wasn't on fire upon contact, the paper pile stayed unharmed.✓

A non-burning, thrown ball of paper landed on the pile but didn't ignite it. ✓

Though the ball reached the pile, it was not burning, and thus the pile remained safe. ✓

A ball that wasn't actively burning was thrown onto the pile, and the pile stayed unignited. ✓

The paper ball made contact with the pile, but without being on fire, the pile did not catch alight. ✓

($X_1$ = True , $X_2$ = True, $Y$ = True)

The flaming paper sphere, still ablaze, was thrown and hit the paper pile, which then caught fire.✓

A blazing ball of paper made contact with a stack of paper, causing the pile to ignite, since the ball was burning and it struck the pile.✓

($X_1$ = True , $X_2$ = False, $Y$ = False)

Despite being actively on fire, the paper ball missed the pile, and as a result, the pile did not catch fire. ✓

Even though the ball was burning, it did not make contact with the pile of paper, so the pile remained unburned. ✓

Scenario:

Rules for each non-root node:

Butter Sliced = (Butter Solid ∧ Downward Slicing Motion Applied)

$X_1$ = Butter Solid    $X_2$ = Downward Slicing Motion Applied    $Y$ = Butter Sliced

Prompts:

Without non-root nodes:

($X_1$ = False , $X_2$ = True)

A knife pierces the soft butter with effortless downward motion.✓

The knife sweeps downward, slicing perfectly through softened butter.✓

Swiftly moving downwards, the knife glides through the creamy butter easily. ✓

($X_1$ = True , $X_2$ = True)

A knife slides effortlessly downward through a solid block of butter. ✓

The solid butter yields smoothly as a knife slices through it with a downward motion.✓

With a straight down slice, the knife cuts cleanly through the solid butter. ✓

Cutting a solid piece of butter with a knife moving downward feels like slicing through soft clay.✓

A sturdy push downward sends the knife through the solidified butter seamlessly. ✓

$(X_1$ = True , $X_2$ = False)

Simply pressing a knife against the solid butter won't cut it.✓

The knife doesn't glide through the solid butter without a downward push. ✓

A knife pressed horizontally against the solid butter fails to cut through.✓

$(X_1$ = False , $X_2$ = False)

A knife resting on soft butter is ineffective without downward force. ✓

No downward motion makes the knife linger atop the soft butter. ✓

Simply resting a knife on soft butter won't achieve a cut. ✓

Without cutting downward, a knife barely breaks the soft butter surface. ✓

The knife sits idle against the soft butter, lacking downward pressure. ✓

With non-root nodes:

$(X_1$ = False , $X_2$ = False, $Y$ = False)

There is no slicing of the butter, as it is neither solid nor subjected to a downward motion. ✓

With the butter not solid and without a downward motion, no slicing occurs. ✓

Neither solid state nor downward motion is present, leaving the butter unsliced.✓

The butter is not solid, and no downward slicing motion is applied, so the butter is not sliced. ✓

$(X_1$ = True , $X_2$ = True, $Y$ = True)

Since the butter is solid and a downward force is used, the knife slices the butter.✓

Solid butter is easily sliced through as a downward slicing motion is applied.✓

The butter, being solid, is sliced through as a downward slicing motion is applied. ✓

The butter is solid, and a downward slicing motion is applied, resulting in the butter being sliced. ✓

With the butter in a solid state and a downward slicing motion in action, the butter gets sliced. ✓

$(X_1$ = True , $X_2$ = False, $Y$ = False)

The butter is solid but no downward slicing motion is applied, so the butter is not sliced.✓

Scenario:

Rules for each non-root node:

Water Emerges from Sponge = (Sponge is Wet ∧ Hand Fully Compresses Sponge)

Sponge Shape Visibly Changes = (Hand Fully Compresses Sponge)

$X_1$ = Sponge is Wet    $X_2$ = Hand Fully Compresses Sponge

$Y_1$ = Water Emerges from Sponge    $Y_2$ = Sponge Shape Visibly Changes

Prompts:

Without non-root nodes:

$(X_1 = \text{False} , X_2 = \text{True})$

A dry sponge is entirely compressed by a hand squeezing it.✓

The hand fully compresses a dry sponge with its grip.✓

A hand squeezes a dry sponge until it's fully compressed. ✓

Fully closing, a hand compresses a dry sponge.✓

The hand squeezes a dry sponge as much as it will go. ✓

$(X_1 = \text{True} , X_2 = \text{True})$

The hand squeezes a wet sponge, fully compressing it. ✓

A hand grips a wet sponge and fully squeezes it.✓

The hand exerts force on a wet sponge, squeezing it flat.✓

A wet sponge is completely compressed by a hand. ✓

A wet sponge is gripped and fully squeezed by a hand. ✓

$(X_1 = \text{True} , X_2 = \text{False})$

Squeezing a wet sponge, the hand stops before fully compressing it.✓

A hand grips a wet sponge, compressing it only slightly. ✓

The hand applies pressure but doesn't fully squeeze the wet sponge.✓

A hand gently squeezes a wet sponge without fully compressing it. ✓

A wet sponge is partially squeezed by a hand. ✓

$(X_1 = \text{False} , X_2 = \text{False})$

The hand applies some pressure to a dry sponge but doesn't compress it completely.✓

A hand holds and gently squeezes a dry sponge without full compression. ✓

The hand grips and squeezes a dry sponge lightly, without full compression. ✓

A hand partially squeezes a dry sponge without complete compression. ✓

With non-root nodes:

$(X_1 = \text{False} , X_2 = \text{False}, Y_1 = \text{False}, Y_2 = \text{False})$

The dry sponge remains unchanged when the hand gives it a gentle squeeze both in terms of shape and water release. ✓

$(X_1 = \text{True} , X_2 = \text{True}, Y = \text{True}, Y_2 = \text{True})$

When the hand squeezes the wet sponge completely, the sponge visibly deforms and water emerges.✓

With a wet sponge being fully pressed by the hand, water seeps out and the sponge's form changes.✓

The wet sponge is fully compressed by the hand, resulting in a change in its shape and water being squeezed out. ✓

$(X_1 = \text{False} , X_2 = \text{True}, Y_1 = \text{False} , Y_2 = \text{True})$

A dry hand compresses the sponge completely, causing its shape to change, but no water releases.✗

$(X_1 = \text{True}, X_2 = \text{False}, Y_1 = \text{False}, Y_2 = \text{False})$

The damp sponge is only partially squeezed by the hand, meaning no water is released and the shape remains consistent.✓

A hand lightly squeezes the wet sponge, leaving its shape and water content unchanged.✓

Although the sponge is wet, the hand does not fully compress it, so no water comes out, and its shape stays the same. ✓

Scenario:

Rules for each non-root node:

Ice Block Moves = (¬ Ice Block On Stable Surface)

Ice Block Cracks = (Hammer Head Metal)

$X_1$ = Ice Block On Stable Surface     $X_2$ = Hammer Head Metal

$Y_1$ = Ice Block Moves     $Y_2$ = Ice Block Cracks

Prompts:

Without non-root nodes:

$(X_1 = \text{False}, X_2 = \text{True})$

A person strikes an ice block with a metal hammer, causing it to slide on the surface.✗

A metal-headed hammer is wielded by a person to hit an ice block that's not stably placed.✓

Someone hits a sliding ice block with a metal hammer. ✓

An individual uses a metal hammer to strike an ice block that isn't on stable footing. ✓

$(X_1 = \text{True}, X_2 = \text{True})$

A person uses a metal-headed hammer to hit an ice block resting on a stable base. ✓

An individual strikes a stable ice block with a metallic hammer.✓

A hammer with a metal head is used by a person to hit a stable ice block.✓

An ice block on a stable platform is struck by someone wielding a metal hammer. ✓

$(X_1 = \text{True}, X_2 = \text{False})$

An individual hits a secure ice block with a hammer that lacks a metal head. ✓

Someone uses a non-metallic hammer to hit an ice block resting stably.✓

A person uses a hammer with a non-metal head to hit an ice block on a stable surface.✓

Striking a solidly placed ice block with a hammer that doesn't have a metal head. ✓

$(X_1 = \text{False}, X_2 = \text{False})$

A person hits an ice block with a non-metal hammer, and the block is not stable.✓

Striking a shifting ice block with a hammer that has a non-metal head. ✓

The hand fully compresses a dry sponge with its grip.Using a non-metal headed hammer, a person hits an unsteady block of ice.✓

The ice block, not secure, is struck by a person with a non-metal hammer. ✓

> Someone uses a hammer without a metal head to hit a loosely sitting ice block. ✓
>
> With non-root nodes:
>
> ($X_1$ = True , $X_2$ = True, $Y_1$ = False, $Y_2$ = True)
>
> The ice block, resting securely on a stable surface, is struck by a hammer with a metal head, which causes it to crack. ✓
>
> ($X_1$ = False , $X_2$ = False, $Y$ = True, $Y_2$ = False)
>
> The ice block on an unsteady surface moves but does not crack when struck with a non-metal hammer. ✓
>
> Even on an unsteady surface, the ice block only shifts without cracking when hit by a non-metal hammer. ✓
>
> A non-metal hammer causes the ice block on an unstable surface to move but avoids cracking. ✓
>
> An ice block shifts on its unstable foundation, though uncracked, under a non-metal hammer blow. ✓
>
> ($X_1$ = True , $X_2$ = False, $Y_1$ = False , $Y_2$ = False)
>
> The ice block, placed securely on a stable surface, does not move or crack when struck by a non-metal hammer. ✓
>
> Striking the ice block on a stable foundation with a non-metal hammer results in no movement or cracking. ✓
>
> A hammer with a non-metal head hits an ice block on stable ground, neither moves nor cracks it. ✓
>
> The ice block, secured by its stable surface, withstands the non-metal hammer blow without cracking or shifting. ✓
>
> A non-metal hammer strikes the ice block on stable ground, leaving it neither cracked nor moved. ✓

## G.3 MANUAL VERIFICATION OF FACTOR-QUESTION ALIGNMENT

To evaluate whether the generated videos comply with causal rules, we utilize a VLLM to extract the values of both root and non-root nodes. When posing "yes-no" questions about the video, it is essential to ensure that the questions are appropriately aligned with the relevant factors in each specific scenario.

In this section, We randomly selected 18 scenarios, comprising a total of 85 factor-question pairs, all of which were found to be correct.

Scenario: A small ball impacts the ground.

| Factor | Question | Correct |
|---|---|---|
| ball is deflated | Is the ball deflated? | ✓ |
| the ground is soft | Is the ground soft? | ✓ |
| ball bounces | Does the ball bounce? | ✓ |

Scenario: Sunlight shines on the water surface, creating sparkling reflections.

| Factor | Question | Correct |
|---|---|---|
| direct sunlight present | Is direct sunlight present on the water surface? | ✓ |
| water ripples visible | Are water ripples visible on the surface? | ✓ |
| unobstructed water surface | Is the water surface unobstructed? | ✓ |

Scenario: A person strikes an ice block with a hammer.

| Factor | Question | Correct |
|---|---|---|
| block is small | Is the ice block small? | ✓ |
| direct hammer strike | Is the hammer striking the ice block directly? | ✓ |
| block breaks | Does the ice block break when struck? | ✓ |

Scenario: Flag waving in the wind at the top of pole.

| Factor | Question | Correct |
|---|---|---|
| is flag hoisted | Is the flag hoisted at the top of the pole? | ✓ |
| is there wind | Is there wind present in the environment? | ✓ |
| flag waving | Is the flag waving? | ✓ |

Scenario: A broom drags across the dirty ceramic floor.

| Factor | Question | Correct |
|---|---|---|
| broom bristles contact floor | Are the broom bristles making contact with the floor? | ✓ |
| floor is wet | Is the floor wet? | ✓ |
| obstruction on floor | Is there an obstruction on the floor? | ✓ |
| floor becomes clean | Does the floor become clean after using the broom? | ✓ |

Scenario: Drop dye into the water.

| Factor | Question | Correct |
|---|---|---|
| dye is water soluble | Is the dye water soluble? | ✓ |
| water is stirred | Is the water stirred? | ✓ |
| water becomes colored | Does the water become colored? | ✓ |
| water becomes uniformly colored | Does the water become uniformly colored? | ✓ |

Scenario: A stone is thrown into a pool, creating a splash.

| Factor | Question | Correct |
|---|---|---|
| Steep entry angle | Please determine whether the stone approaches the water on a clearly steep downward path. | ✓ |
| Impact adjacent to wall | Please determine whether the stone's first contact point with the water is immediately next to a vertical pool wall/edge. | ✓ |
| Calm water surface | Please determine whether the water surface around the impending impact area appears calm just before contact. | ✓ |
| Splash visible | Please determine whether water is visibly ejected above the undisturbed surface at the moment of impact. | ✓ |
| Water hits wall/deck | Please determine whether splash water visibly strikes the pool's wall or deck. | ✓ |
| Concentric ripples visible | Please determine whether near-circular ripples are clearly visible expanding outward on the water surface from the impact point. | ✓ |
| No surface skipping | Please determine whether the stone does not skip on the surface after first contact. | ✓ |

Scenario: Smoke spreads into the air from the chimney.

| Factor | Question | Correct |
|---|---|---|
| Emission visible | Please determine whether visible smoke is exiting the chimney outlet. | ✓ |
| Chimney cap present | Please determine whether a cap, hood, or mesh cover is mounted over the chimney outlet. | ✓ |
| Visible dispersion | Please determine whether the visible smoke plume broadens and becomes less opaque with distance from the chimney. | ✓ |
| Edge-emission around cap | Please determine whether smoke emerges around the perimeter of a visible horizontal hood or mesh at the top of the chimney. | ✓ |

Scenario: The billiard balls collide with each other on the table.

| Factor | Question | Correct |
|---|---|---|
| Head-on into stationary | Please determine whether, at the instant of first contact, the target ball is visibly motionless and the striker's incoming path is aligned with the line connecting their centers. | ✓ |
| Frozen pushed into cushion | Please determine whether, at ball–ball contact, the target ball is visibly touching the rail cushion with no gap and the push direction points into the cushion. | ✓ |
| Third ball inline touching | Please determine whether, at the instant of the two-ball collision, the target ball is in visible contact with a third ball and all three ball centers lie nearly on a straight line. | ✓ |
| Target follows striker line | Please determine whether, immediately after impact, the target ball's initial motion is along the striker's pre-impact path. | ✓ |
| Immediate cushion bounce | Please determine whether the target ball clearly contacts the rail cushion and then reverses direction right after the ball–ball impact. | ✓ |
| Third ball moves immediately | Please determine whether a third ball begins to translate right after the initial contact between the striker and target balls. | ✓ |

Scenario: Flip the hourglass.

| Factor | Question | Correct |
|---|---|---|
| Inverted Hourglass | Please determine whether the hourglass is visibly inverted relative to gravity. | ✓ |
| Sand In Upper Bulb | Please determine whether there is visible sand inside the bulb that is uppermost. | ✓ |
| Neck Unblocked | Please determine whether the narrow neck of the hourglass appears unblocked. | ✓ |
| Sand Starts Flowing | Please determine whether grains of sand visibly begin to pass through the neck. | ✓ |
| Steady Sand Stream | Please determine whether a steady, unbroken-looking stream of sand is visible through the neck. | ✓ |

Scenario: A cat jumped onto the table.

| Factor | Question | Correct |
|---|---|---|
| Clear landing spot | Please determine whether the exact first-contact footprint on the tabletop under the cat's first paws and chest is free of discrete items. | ✓ |
| Draped cloth under paws | Please determine whether a loose, draped fabric (tablecloth or runner) with visible slack lies directly beneath the cat's first-contact paw footprint. | ✓ |
| Object contact at landing | Please determine whether any discrete tabletop object (excluding any cloth/covering) is contacted at the instant of first paw touchdown. | ✓ |
| Cloth visibly shifts | Please determine whether the cloth or runner located under the landing footprint moves or deforms at or immediately after the cat's touchdown. | ✓ |

Scenario: Pour one liquid into another.

| Factor | Question | Correct |
|---|---|---|
| Pour after brim | Please determine whether pouring into the receiving container continues after its liquid surface visibly reaches the rim. | ✓ |
| Stream outside mouth | Please determine whether any portion of the falling liquid stream or droplets misses the receiving container's opening and lands outside. | ✓ |
| Overflow occurs | Please determine whether overflow occurs, meaning liquid passes over the receiving container's rim from the inside to the outside during or immediately after the pour. | ✓ |
| Spillage outside | Please determine whether any spillage outside is present, meaning some liquid is located outside the receiving container's mouth on its exterior surface or surrounding area during the pour. | ✓ |

Scenario: Chocolate melds into milk

| Factor | Question | Correct |
|---|---|---|
| Visible milk steam | Please determine whether translucent vapor wisps are clearly visible rising from the milk. | ✓ |
| Stirring occurs | Please determine whether a utensil or device is visibly moving through the milk with repeated motions. | ✓ |
| Chocolate melts/dissolves at contact | Please determine whether, at the chocolate-milk boundary, the chocolate visibly melts or dissolves. | ✓ |
| Chocolate visibly disperses | Please determine whether chocolate visually disperses into the milk. | ✓ |
| Chocolate fully incorporated | Please determine whether, by the end of the clip, the mixture appears fully incorporated. | ✓ |
| Milk color darkens | Please determine whether the milk's overall color visibly darkens during the clip. | ✓ |

Scenario: A brush dips into watercolor on a palette.

| Factor | Question | Correct |
|---|---|---|
| Brush contacts liquid | Please determine whether the brush bristles visibly touch or enter the puddle. | ✓ |
| Colored liquid present | Please determine whether the puddle on the palette is visibly colored. | ✓ |
| Liquid surface disturbed | Please determine whether the surface of the puddle shows visible disturbance. | ✓ |
| Bristles change color | Please determine whether the brush's bristle bundle shows a new visible hue or darkening compared with its appearance earlier in the video. | ✓ |

Scenario: Car jolting as it hits a pothole.

| Factor | Question | Correct |
|---|---|---|
| Left wheel enters | Please determine whether the left front wheel visibly drops into a depression in the road surface. | ✓ |
| Right wheel enters | Please determine whether the right front wheel visibly drops into a depression in the road surface. | ✓ |
| Braking at impact | Please determine whether the vehicle's rear brake lamps are illuminated at the moment the front wheel(s) reach the pothole edge. | ✓ |
| Upward rebound observed | Please determine whether the vehicle body shows a rapid upward motion after a preceding downward movement. | ✓ |
| Nose dive observed | Please determine whether the front of the car pitches downward relative to the rear at or just before the impact area. | ✓ |
| Left side drops | Please determine whether the left side of the car lowers at the moment of impact. | ✓ |
| Right side drops | Please determine whether the right side of the car lowers at the moment of impact. | ✓ |

Scenario: An air mattress floats on a pool.

| Factor | Question | Correct |
|---|---|---|
| Adequately inflated | Please determine whether the air mattress appears adequately inflated based only on what is directly visible. | ✓ |
| Person boards during clip | Please determine whether a person is visibly getting onto the air mattress during the clip by directly observing a human body moving from the water or poolside onto the mattress. | ✓ |
| External push or pull | Please determine whether the air mattress is being directly pushed or pulled by a visible hand, foot, pole, or other object. | ✓ |
| Deck above waterline initially | Please determine whether, at the first stable moment shown, the top sleeping surface (deck) of the air mattress is above the waterline. | ✓ |
| Waterline rises after boarding | Please determine whether the visible waterline on the air mattress rises after a person boards the mattress. | ✓ |
| Moves due to push | Please determine whether the air mattress's motion changes after a visible push or pull on it. | ✓ |

Scenario: A snowball falls to the ground.

| Factor | Question | Correct |
|---|---|---|
| Snow at landing spot | Please determine whether the exact area the snowball is about to touch is visibly covered by snow. | ✓ |
| Pre-impact fragmented | Please determine whether the snowball is already in multiple separate pieces immediately before touching the ground. | ✓ |
| Impact severity high | Please determine whether the snowball's approach right before impact appears very fast based only on immediate pre-contact visuals. | ✓ |
| First contact on snow | Please determine whether the very first touch between the snowball and the surface is on snow. | ✓ |
| Arrives intact at contact | Please determine whether the snowball is a single continuous piece at the instant just before first contact. | ✓ |
| Breaks on impact | Please determine whether the snowball breaks as a direct consequence of contacting the ground. | ✓ |

Scenario: Pour the salt into the water.

| Factor | Question | Correct |
|---|---|---|
| Salt contacts water | Please determine whether any white granular salt particles are visibly touching the water surface inside the container. | ✓ |
| Rim impact occurs | Please determine whether any falling salt grains make direct visual contact with the container's rim or outer wall. | ✓ |
| Rotational stirring occurs | Please determine whether there is a visible circular stirring motion applied by a utensil or by rotating the container. | ✓ |
| Surface ripples at impact | Please determine whether concentric ripples or localized wave patterns are visible on the water surface directly under where the falling salt arrives. | ✓ |
| Stream deflects at rim | Please determine whether the falling salt stream visibly changes direction or splits exactly at the container rim/edge line. | ✓ |
| Visible vortex motion | Please determine whether a visible vortex (rotational swirl) appears in the water during or shortly after the pour. | ✓ |

## G.4 MANUAL VERIFICATION OF VLLM ANSWER RETRIEVAL CORRECTNESS

The answers provided by VLLM serve as the foundation for calculating the final score of the generated videos. Therefore, it is essential to manually verify the accuracy of these responses. In this section, we select four models and examine three distinct scenarios, each accompanied by three corresponding prompts.

The sampled scenarios encompass both challenging and easy prompts, with and without non-root nodes, and feature answers classified as True, False, or N/A. A comprehensive explanation of the conditions under which VLLM provides an N/A response is available in H.3. For example, the explanation provided by VLLM for the N/A response regarding a video generated by Pika, as presented in Table 16, is: "*The images do not provide a clear view of the top of the boot. It is not possible to determine if it is sealed or not from the given angles.*" for the factor "*boot top sealed*", which is consistent with our observations.

Regarding the accuracy of model responses, we find that VLLM demonstrates sufficient capability to handle simple scenarios and prompts (such as those in Table 13, Table 15,Table 16,Table 17,Table 18, and Table 19). However, its performance declines when addressing more complex questions (such as those in Table 11, Table 12, and Table 14). Currently, the accuracy of this approach hovers around 95%, which is acceptable but still leaves room for improvement. The shortcomings in correctness primarily stem from two factors. First, the VGMs often generate videos with low quality and ambiguity, which increases the difficulty for VLLM to provide accurate answers. Additionally, VLLMs still lack the ability to clearly understand intricate details in images or videos, particularly when dealing with more complex questions. Nevertheless, we are optimistic that as the foundational capabilities of VLLMs continue to improve, the performance of this video description system will experience significant enhancement.

The checked question-answer pairs are shown below, accompanied by the generated videos. Our verification results are presented in a table that closely follows each prompt.

**Scenario: A ray of light is shining on a wooden block.**

Prompt-1: A beam of light grazes the polished surface of a wooden block in dust. (Videos: Figure 7; Results: Tabel 11)

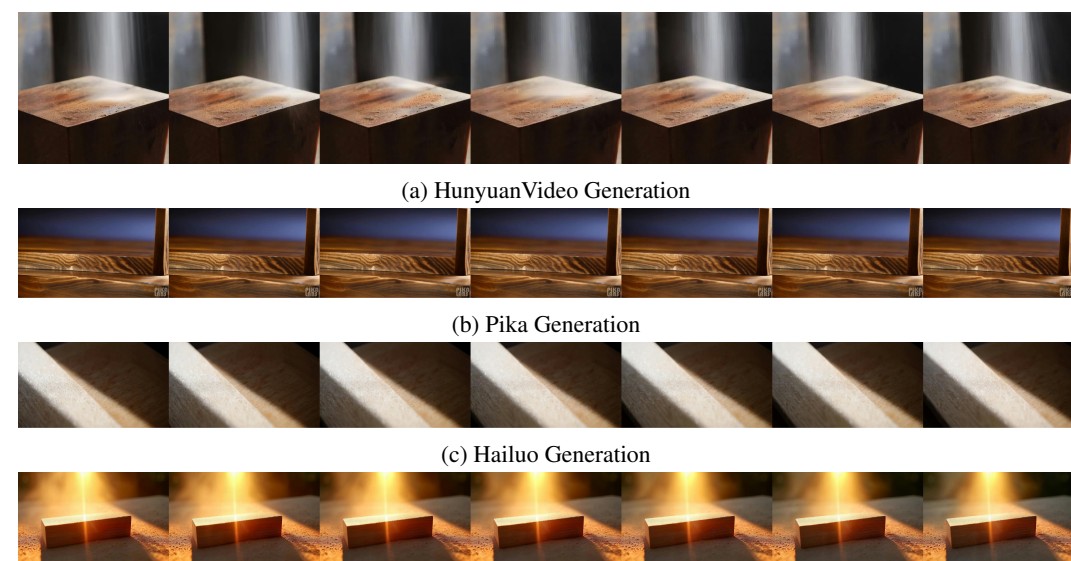

(a) HunyuanVideo Generation

(b) Pika Generation

(c) Hailuo Generation

(d) Pyramid Generation

Figure 7: Model Generation

| Model | Factor | Model Answer | Correct |
|-------|--------|--------------|---------|
| Hunyuan | in direct path | True | ✓ |
| | surface polished | False | ✗ |
| | environment dusty | False | ✓ |
| | block illuminated | True | ✓ |
| | reflection visible | False | ✓ |
| | beam visible in air | True | ✓ |
| Pika | in direct path | False | ✓ |
| | surface polished | True | ✓ |
| | environment dusty | False | ✓ |
| | block illuminated | False | ✓ |
| | reflection visible | False | ✓ |
| | beam visible in air | False | ✓ |
| Hailuo | in direct path | N/A | ✓ |
| | surface polished | False | ✓ |
| | environment dusty | False | ✓ |
| | block illuminated | True | ✓ |
| | reflection visible | True | ✗ |
| | beam visible in air | False | ✓ |
| Pyramid | in direct path | True | ✓ |
| | surface polished | False | ✓ |
| | environment dusty | True | ✓ |
| | block illuminated | True | ✓ |
| | reflection visible | False | ✓ |
| | beam visible in air | True | ✓ |

Table 11: Verification of VLLM Answer Correctness

Prompt-2:The polished surface of a wooden block directly catches the light amid dust. (Videos:Figure 8;Results: Tabel 12)

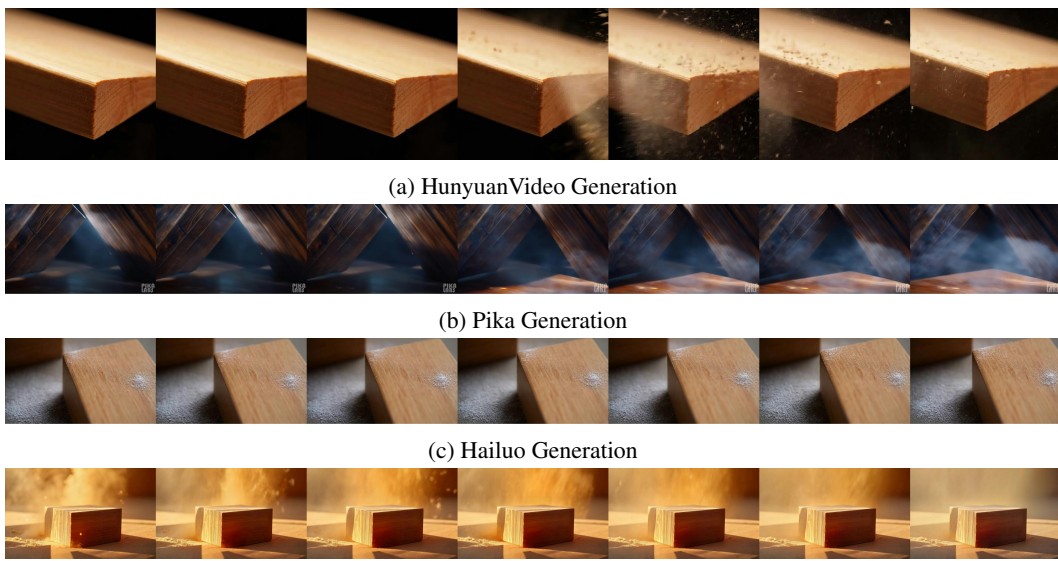

(a) HunyuanVideo Generation

(b) Pika Generation

(c) Hailuo Generation

(d) Pyramid Generation

Figure 8: Model Generation

| Model | Factor | Model Answer | Correct |
|-------|--------|--------------|---------|
| Hunyuan | in direct path | False | ✓ |
| | surface polished | False | ✓ |
| | environment dusty | True | ✓ |
| | block illuminated | False | ✓ |
| | reflection visible | False | ✓ |
| | beam visible in air | False | ✓ |
| Pika | in direct path | False | ✓ |
| | surface polished | False | ✗ |
| | environment dusty | True | ✓ |
| | block illuminated | False | ✓ |
| | reflection visible | False | ✓ |
| | beam visible in air | True | ✓ |
| Hailuo | in direct path | False | ✓ |
| | surface polished | False | ✓ |
| | environment dusty | True | ✓ |
| | block illuminated | False | ✓ |
| | reflection visible | False | ✓ |
| | beam visible in air | False | ✓ |
| Pyramid | in direct path | True | ✓ |
| | surface polished | False | ✓ |
| | environment dusty | True | ✓ |
| | block illuminated | True | ✓ |
| | reflection visible | True | ✗ |
| | beam visible in air | False | ✓ |

Table 12: Verification of VLLM Answer Correctness

Prompt-3:A ray of light directly illuminates a polished wooden block and the environment is dusty, causing both the block to be lit and reflections to be visible, with the beam clearly seen in the air. (Videos:Figure 9;Results: Tabel 13)

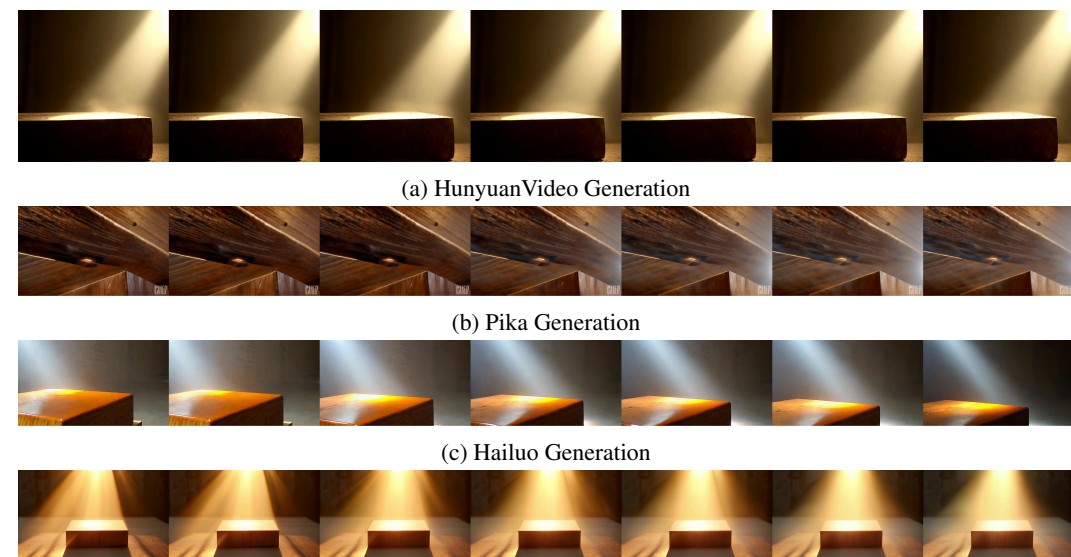

(a) HunyuanVideo Generation

(b) Pika Generation

(c) Hailuo Generation

(d) Pyramid Generation

Figure 9: Model Generation

| Model | Factor | Model Answer | Correct |
|---|---|---|---|
| Hunyuan | in direct path | True | ✓ |
| | surface polished | False | ✓ |
| | environment dusty | True | ✓ |
| | block illuminated | True | ✓ |
| | reflection visible | False | ✓ |
| | beam visible in air | True | ✓ |
| Pika | in direct path | True | ✓ |
| | surface polished | False | ✓ |
| | environment dusty | True | ✓ |
| | block illuminated | False | ✓ |
| | reflection visible | False | ✓ |
| | beam visible in air | True | ✓ |
| Hailuo | in direct path | True | ✓ |
| | surface polished | True | ✓ |
| | environment dusty | False | ✓ |
| | block illuminated | True | ✓ |
| | reflection visible | False | ✓ |
| | beam visible in air | True | ✓ |
| Pyramid | in direct path | True | ✓ |
| | surface polished | False | ✓ |
| | environment dusty | True | ✓ |
| | block illuminated | True | ✓ |
| | reflection visible | False | ✓ |
| | beam visible in air | True | ✓ |

Table 13: Verification of VLLM Answer Correctness

**Scenario: A boot stomps into a puddle of mud.**

Prompt-1:An intense stomp by an open-topped boot into a puddle of watery mud occurs. (Videos:Figure 10;Results: Tabel 14)

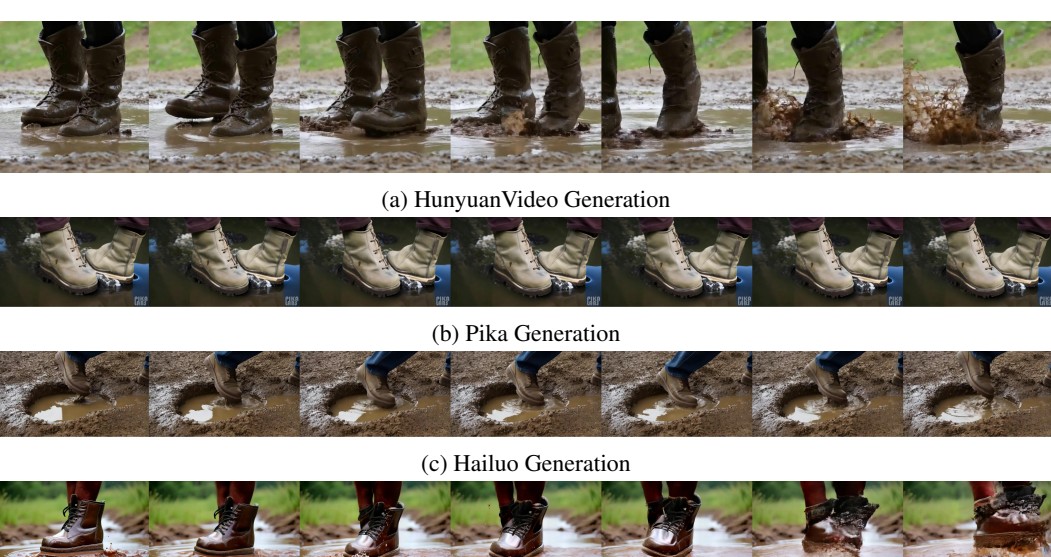

(a) HunyuanVideo Generation

(b) Pika Generation

(c) Hailuo Generation

(d) Pyramid Generation

Figure 10: Model Generation

| Model | Factor | Model Answer | Correct |
|---|---|---|---|
| Hunyuan | watery mud | True | ✓ |
| | big downward stomp | True | ✓ |
| | boot top sealed | False | ✓ |
| | mud splashes out of puddle | True | ✓ |
| | mud enters the boot | False | ✓ |
| Pika | watery mud | True | ✓ |
| | big downward stomp | False | ✓ |
| | boot top sealed | True | ✗ |
| | mud splashes out of puddle | False | ✓ |
| | mud enters the boot | False | ✓ |
| Hailuo | watery mud | True | ✓ |
| | big downward stomp | True | ✓ |
| | boot top sealed | N/A | ✓ |
| | mud splashes out of puddle | True | ✗ |
| | mud enters the boot | False | ✓ |
| Pyramid | watery mud | True | ✓ |
| | big downward stomp | True | ✓ |
| | boot top sealed | False | ✓ |
| | mud splashes out of puddle | True | ✓ |
| | mud enters the boot | False | ✓ |

Table 14: Verification of VLLM Answer Correctness

Prompt-2:In non-watery mud, no splashes occur, but mud enters an unsealed boot during light stepping. (Videos:Figure 11;Results: Tabel 15)

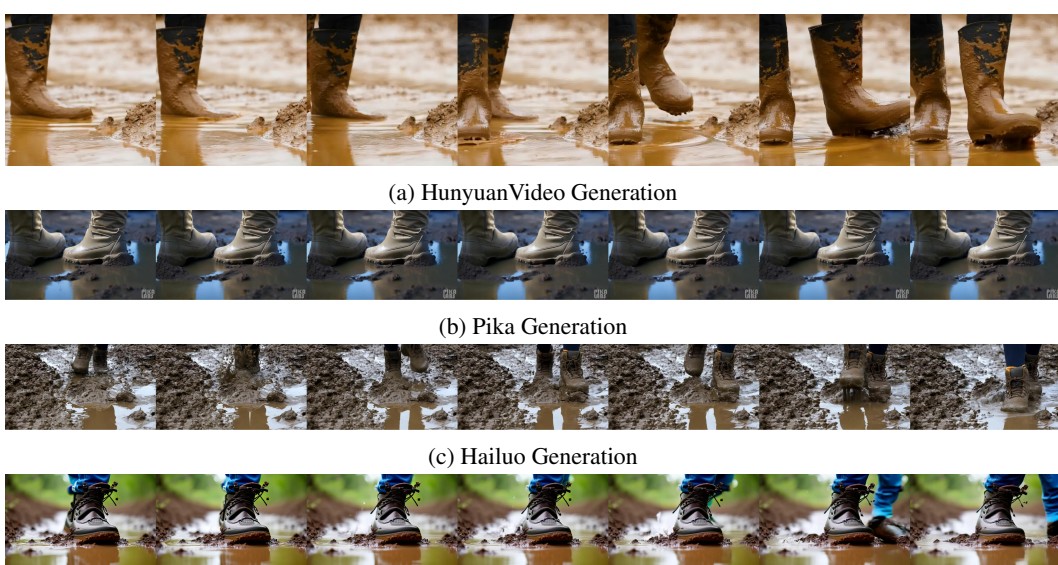

(a) HunyuanVideo Generation

(b) Pika Generation

(c) Hailuo Generation

(d) Pyramid Generation

Figure 11: Model Generation

| Model | Factor | Model Answer | Correct |
|---|---|---|---|
| Hunyuan | watery mud | True | ✓ |
| | big downward stomp | False | ✓ |
| | boot top sealed | False | ✓ |
| | mud splashes out of puddle | False | ✓ |
| | mud enters the boot | False | ✓ |
| Pika | watery mud | True | ✓ |
| | big downward stomp | False | ✓ |
| | boot top sealed | True | ✓ |
| | mud splashes out of puddle | False | ✓ |
| | mud enters the boot | False | ✓ |
| Hailuo | watery mud | True | ✓ |
| | big downward stomp | True | ✓ |
| | boot top sealed | True | ✓ |
| | mud splashes out of puddle | True | ✓ |
| | mud enters the boot | False | ✓ |
| Pyramid | watery mud | True | ✓ |
| | big downward stomp | True | ✓ |
| | boot top sealed | False | ✓ |
| | mud splashes out of puddle | True | ✓ |
| | mud enters the boot | N/A | ✓ |

Table 15: Verification of VLLM Answer Correctness

Prompt-3:A boot with a sealed top makes a big downward stomp into watery mud, causing mud to splash out of the puddle but none enters the boot. (Videos:Figure 12;Results: Tabel 16)

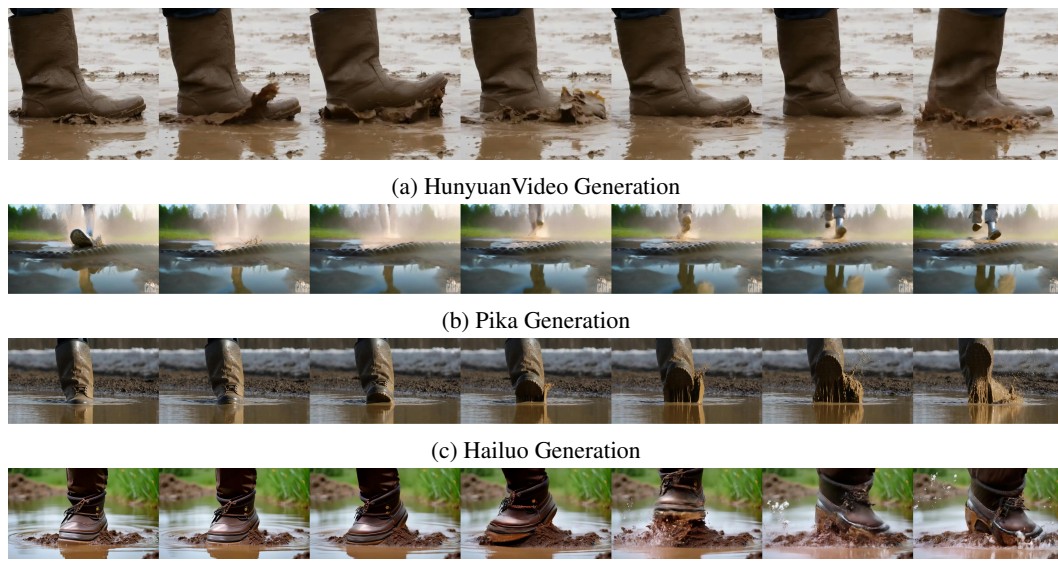

(a) HunyuanVideo Generation

(b) Pika Generation

(c) Hailuo Generation

(d) Pyramid Generation

Figure 12: Model Generation

| Model | Factor | Model Answer | Correct |
|---|---|---|---|
| Hunyuan | watery mud | True | ✓ |
| | big downward stomp | True | ✓ |
| | boot top sealed | True | ✓ |
| | mud splashes out of puddle | True | ✓ |
| | mud enters the boot | False | ✓ |
| Pika | watery mud | True | ✓ |
| | big downward stomp | True | ✓ |
| | boot top sealed | N/A | ✓ |
| | mud splashes out of puddle | True | ✓ |
| | mud enters the boot | N/A | ✓ |
| Hailuo | watery mud | True | ✓ |
| | big downward stomp | True | ✓ |
| | boot top sealed | N/A | ✓ |
| | mud splashes out of puddle | True | ✓ |
| | mud enters the boot | False | ✓ |
| Pyramid | watery mud | True | ✓ |
| | big downward stomp | True | ✓ |
| | boot top sealed | True | ✓ |
| | mud splashes out of puddle | True | ✓ |
| | mud enters the boot | False | ✓ |

Table 16: Verification of VLLM Answer Correctness

**Scenario: Knife slicing through butter.**

Prompt-1:The knife meets little opposition as it slices through the butter. (Videos:Figure 13;Results: Tabel 17)

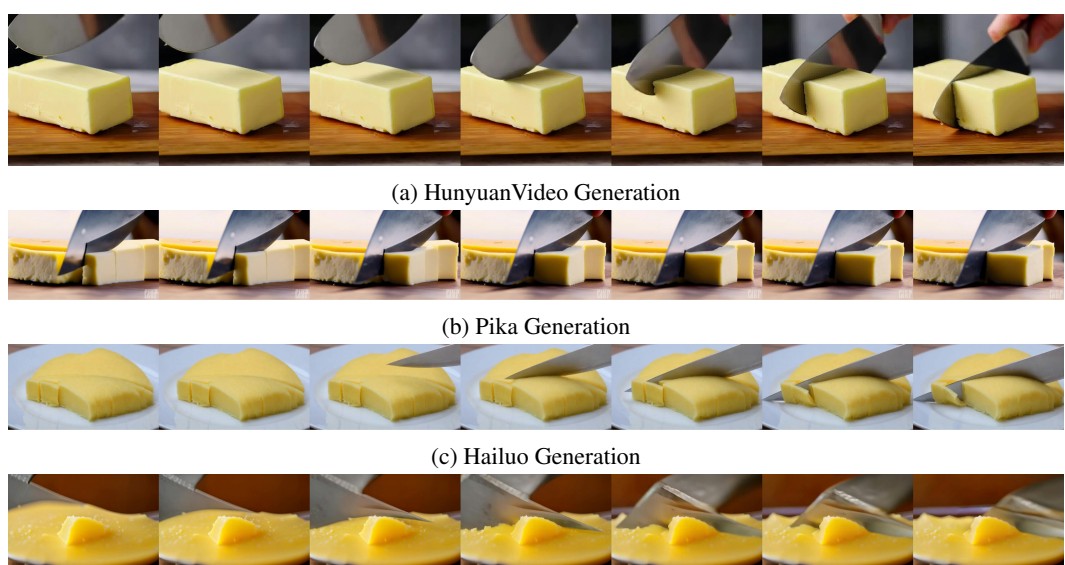

(a) HunyuanVideo Generation

(b) Pika Generation

(c) Hailuo Generation

(d) Pyramid Generation

Figure 13: Model Generation

| Model | Factor | Model Answer | Correct |
|---|---|---|---|
| Hunyuan | blade in contact with butter | True | ✓ |
| | Knife is moving against butter | True | ✓ |
| | Butter is sliced | True | ✓ |
| Pika | blade in contact with butter | True | ✓ |
| | Knife is moving against butter | True | ✓ |
| | Butter is sliced | True | ✓ |
| Hailuo | blade in contact with butter | True | ✓ |
| | Knife is moving against butter | True | ✓ |
| | Butter is sliced | True | ✓ |
| Pyramid | blade in contact with butter | True | ✓ |
| | Knife is moving against butter | True | ✓ |
| | Butter is sliced | False | ✓ |

Table 17: Verification of VLLM Answer Correctness

Prompt-2:With no movement or contact, the butter sits undisturbed. (Videos:Figure 14;Results: Tabel 18)

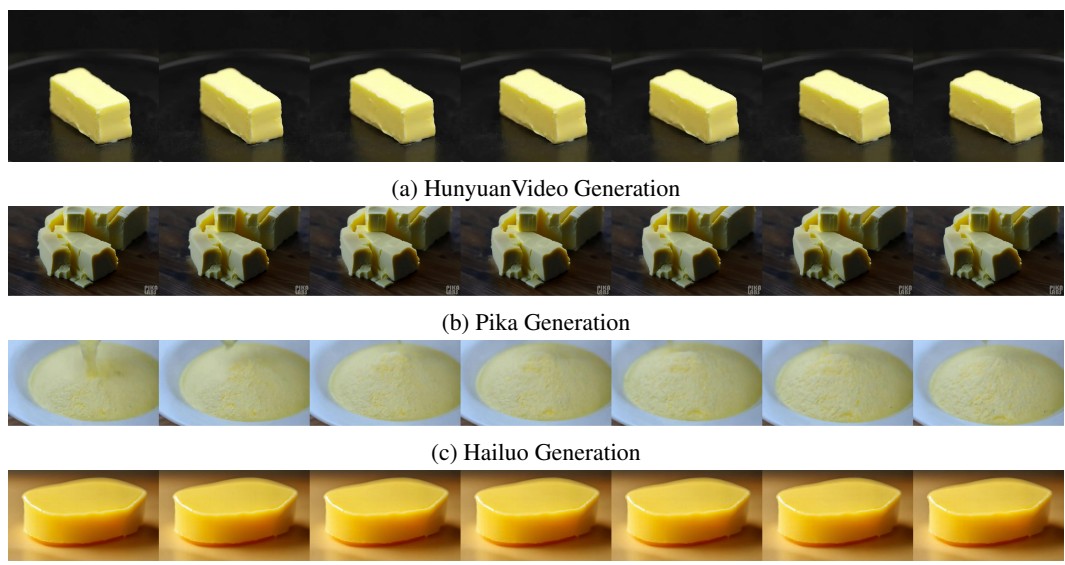

(a) HunyuanVideo Generation

(b) Pika Generation

(c) Hailuo Generation

(d) Pyramid Generation

Figure 14: Model Generation

| Model | Factor | Model Answer | Correct |
|---|---|---|---|
| Hunyuan | blade in contact with butter | False | ✓ |
| | Knife is moving against butter | False | ✓ |
| | Butter is sliced | False | ✓ |
| Pika | blade in contact with butter | False | ✓ |
| | Knife is moving against butter | False | ✓ |
| | Butter is sliced | False | ✓ |
| Hailuo | blade in contact with butter | False | ✓ |
| | Knife is moving against butter | False | ✓ |
| | Butter is sliced | False | ✓ |
| Pyramid | blade in contact with butter | False | ✓ |
| | Knife is moving against butter | False | ✓ |
| | Butter is sliced | False | ✓ |

Table 18: Verification of VLLM Answer Correctness

Prompt-3:Contact with the butter is established, but without motion, the butter remains unsliced. (Videos:Figure 15;Results: Tabel 19)

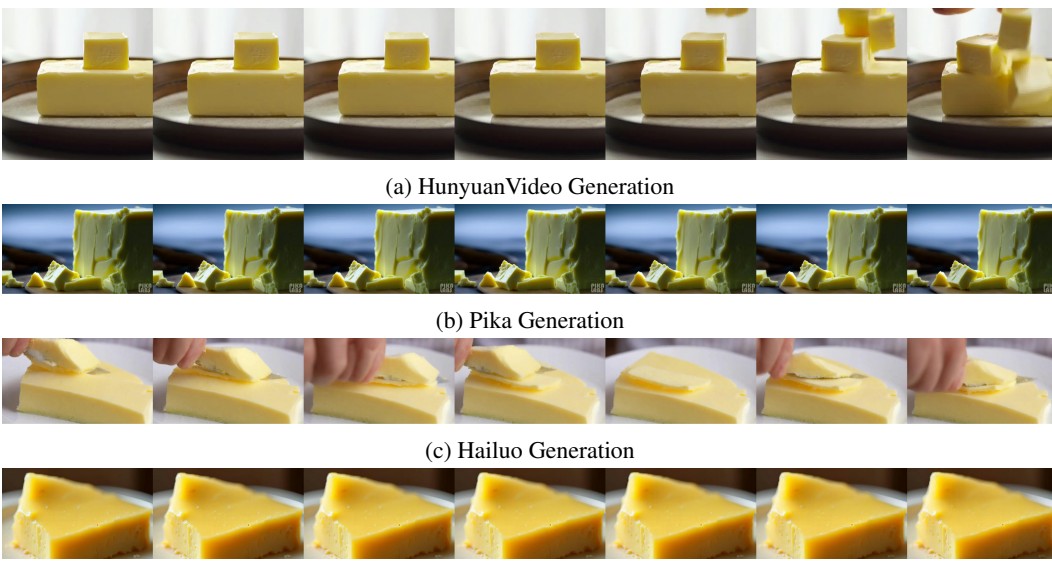

(a) HunyuanVideo Generation

(b) Pika Generation

(c) Hailuo Generation

(d) Pyramid Generation

Figure 15: Model Generation

| Model | Factor | Model Answer | Correct |
|-------|--------|--------------|---------|
| Hunyuan | blade in contact with butter | False | ✓ |
| | Knife is moving against butter | False | ✓ |
| | Butter is sliced | False | ✓ |
| Pika | blade in contact with butter | False | ✓ |
| | Knife is moving against butter | False | ✓ |
| | Butter is sliced | False | ✓ |
| Hailuo | blade in contact with butter | True | ✓ |
| | Knife is moving against butter | True | ✓ |
| | Butter is sliced | True | ✓ |
| Pyramid | blade in contact with butter | False | ✓ |
| | Knife is moving against butter | False | ✓ |
| | Butter is sliced | False | ✓ |

Table 19: Verification of VLLM Answer Correctness

# H  DETAILS AND MORE DISCUSSION ABOUT BENCHMARKS

## H.1  EVALUATED MODELS

To conduct a comprehensive benchmark, we evaluate a total of 5 closed-source models. Detailed information about the models included in our evaluation is provided in this section.

**Close-Source Models:**

For the close-source models, we benchmark

- Pika2.2 (pika, 2024), developed by Pika Labs;

- Hailuo2.0 (MiniMax, 2024), developed by MiniMax, is used in its T2V-02 version;

- Kling 2.1-master (Kuaishou, 2024), a closed VGM released by Kuaishou.

- CogVideoX-3 (Hong et al., 2022), ZhipuAI's production text-to-video model, distinct from the open-source CogVideoX 2B/5B releases that target research and local use.

- Veo3 Fast (Google DeepMind, 2025), Veo 3 is Google's state-of-the-art text-to-video model that produces high-fidelity 8-second clips at 720p or 1080p with native audio across a wide range of cinematic styles; Veo 3 Fast offers a faster, more cost-effective path to similar capabilities for rapid iteration.

We access all the closed-source models by calling their APIs, either through their official websites or third-party interfaces. Detailed information can be found in H.2. Some of the models provide an additional prompt enhancement trick but for fair comparison, we do not turn it on if there is an option. See discussion about this trick in Appendix J.

## H.2 COST OF BENCHMARKING

We report the money cost of benchmarking each model here.

Close-Source Models: see Table 20.

Table 20: The money cost for close-source models.

| Name | API Source | Cost / Video | Total Cost (above 2000 videos) |
|---|---|---|---|
| Pika2.2 | pollo.ai | $ 0.21 | $ 320 |
| Kling | PiAPI | $ 0.96 | $ 730 |
| Hailuo | Official | $ 0.28 | $ 430 |
| CogVideo | Official | $ 0.14 | $ 215 |
| Veo3 Fast | pollo.ai | $0.8 | $ 1216 |

## H.3 ABOUT N/A RESULTS

When we retrieve the observed values in a video by a VLLM, we allow the model to answer "N/A" besides yes or no. We prompt the model the conditions of answering N/A as follows:

General Rules for Answering "N/A":

1. The video quality is too low, or the content is too unclear to make any meaningful inference.

2. The video is not continuous or complete (temporal or spatial discontinuities prevent reasonable judgment).

3. The question asks about something that cannot be observed or recognized in the video.

4. The video does not provide enough context or evidence to form a conclusion.

5. The answer is unclear or could be interpreted in multiple ways, leading to ambiguity.

General Rules for Answering each question:

A. When the question is about an action

- Answer 'true' if the action clearly happens in the video (confirmed by continuous frames, not just one frame).

- Answer 'false' if the subject is present, the necessary prior action occurs, but the action clearly does not happen.

- Answer 'N/A' if:

- The subject of the action is not present or is too unclear to identify. *Example*: "Does the ball roll away after impact?" → If no ball exists, or it is unclear whether the object is a ball, answer 'N/A'.

B. When the question is about an object's existence

- Answer 'true' if the object is clearly observed in the video.

- Answer 'false' if the object is not observed in the video.

- Answer 'N/A' only if the video quality is too low or unclear to determine whether the object is present.

C. When the question is about an object's feature

- Judge based only on the object itself (its shape, material, appearance, etc.), not on indirect evidence such as collisions, rebounds, or interactions.

- Answer 'true' if there is clear evidence the object has the feature.

- Answer 'false' if there is clear evidence the object does not have the feature, **or if the feature is clearly absent from the object**.

- Answer 'N/A' only if the video quality is too low, the object is too unclear, or there is no sufficient visual evidence to make a judgment.

- If you believe you can reasonably conclude 'true' or 'false', you should not answer 'N/A'. Default to 'false' if the feature is visibly absent rather than uncertain.

We report the N/A ratio in all observation in Table 4

We acknowledge that the appearance of N/A may introduce some bias to subsequent metrics. For example, if the model generates N/A in scenarios where it performs poorly, removing these N/A responses could lead to inflated scores. This would make the model appear better than it actually is, or falsely narrow the performance gap between different models. But as we mentioned in the introduction (Section 1), as a longer-term goal, our evaluation focuses more on the evaluation of the "world simulator", and the guarantee of general video generation quality should be taken as a prerequisite rather than the focus of this article. At the same time, we observe that better (newer, larger) models tend to have a lower N/A ratio, which is in line with our expectations and shows that as the model generation capability continues to improve, the probability of obvious serious errors will gradually decrease.

## H.4 EXPERIMENT FOR SAMPLE SIZE

We conduct an empirical study to determine the minimum sample size required for statistically distinguishing performance metrics between two video generation models (VGMs). The experiment compares CogVideoX-2B (representing open-source models) and Pika (representing closed-source models) under a specific causal system where both models exhibited competent video generation quality. We vary sample sizes from 2 to 100 for text consistency, group sizes from 2 to 16 for generation consistency, and sample sizes for each outcome variable from 2 to 50 samples for rule consistency. To ensure statistical validity, we employ bootstrap resampling (1,000 iterations) with finite-population correction to estimate standard deviations of metric estimators. Standard deviations are adjusted for matching our scenario pool (57 causal systems). For text consistency metrics, we implement two evaluation protocols: 1) excluding missing (N/A) observations, and 2) treating N/A values as incorrect responses. Confidence intervals (95% coverage) are constructed using bias-corrected accelerated bootstrap methods centered on the minimum-variance unbiased estimator.

The results, visualized in Figure 16, reveal distinct sample size requirements across metrics. As a efficiency-accuracy trade-off, we established an operational criterion where the minimal sufficient sample size occurs when the confidence interval of one model's metric no longer overlaps with the point estimate of the competitor model. From the figure we can see that:

- For text consistency, drawing $n_1 = 10$ samples is enough to distinguish metrics between two models in most cases. When N/A observed variables are seen as incorrect, $s_1^{\text{all}}$ between two models cannot be distinguished for any number of samples.

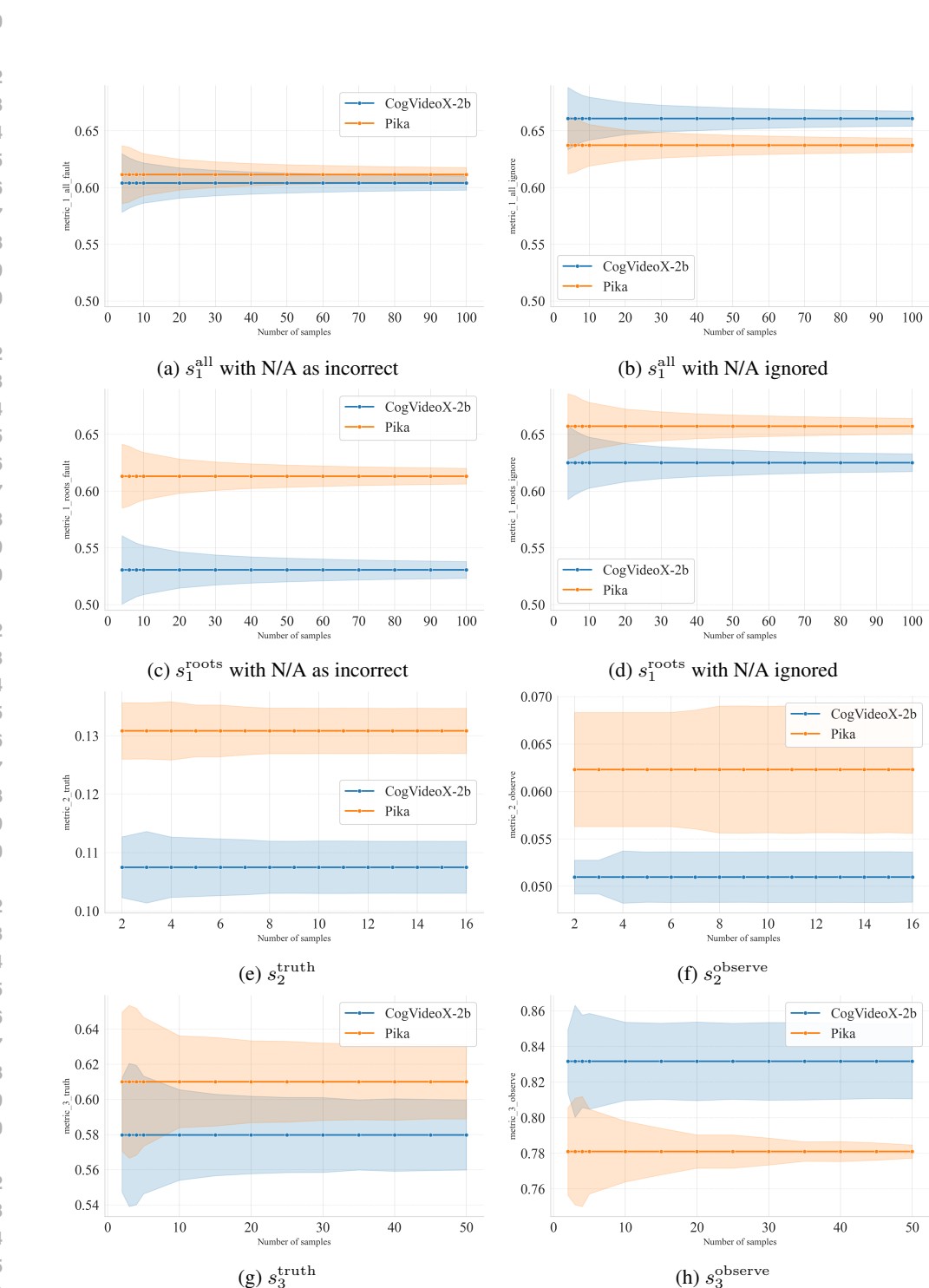

Figure 16: Estimated confidence interval for each metric as the sample size increases.

- For generation consistency, drawing $n_2 = 5$ groups can distinguish metrics between two models.

- For rule consistency, drawing $n_3 = 10$ samples for each outcome variable can distinguish metrics between two models.

Based on these findings, our benchmark protocol adopts $n_1 = 10$, $n_2 = 5$, and $n_3 = 10$ as optimal parameters balancing statistical power and evaluation efficiency, leading to total 2079 video samples. The sample numbers of these 57 causal systems are shown in Figure 17.

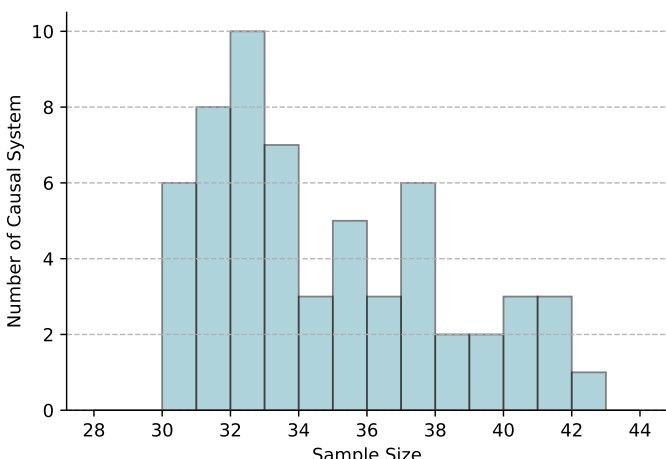

Figure 17: Sample numbers of the 57 causal systems in VACT benchmark.

## H.5 MODEL PERFORMANCE VS CAUSAL GRAPH COMPLEXITY

To study the relationship between model performance and causal-graph complexity, we group scenarios by the total number of variables in their causal graphs and average each metric over all scenarios with the same number of variables and across all five VGMs. We then compute the Pearson correlation coefficient between the average metric value and the number of variables. The results are summarized in Table 21. Overall, most accuracy-style metrics (text and rule consistency) exhibit a moderate negative correlation with graph size, indicating that scenarios with more complex causal graphs are generally harder for current VGMs, while generation-consistency metrics show much weaker trends. It is because our generation-consistency is not an accuracy but a variance score. The variance cannot be directly compared with different numbers of factors.

Table 21: Average model performance grouped by the total number of variables in the causal graph, and the Pearson correlation between each metric and graph size. Values are averaged over all scenarios with the same total number of nodes and over all VGMs.

| #Nodes | Text Consistency ↑ | | Generation Consistency ↓ | | Rule Consistency ↑ | |
|---|---|---|---|---|---|---|
| | all | root | truth | observe | truth | observe |
| 3 | .58 | .69 | .10 | .11 | .61 | .61 |
| 4 | .69 | .72 | .06 | .07 | .69 | .90 |
| 5 | .62 | .74 | .04 | .08 | .63 | .69 |
| 6 | .60 | .67 | .06 | .04 | .62 | .77 |
| 7 | .53 | .58 | .05 | .12 | .61 | .62 |
| Pearson $r$ | -.51 | -.69 | -.69 | -.05 | -.33 | -.14 |

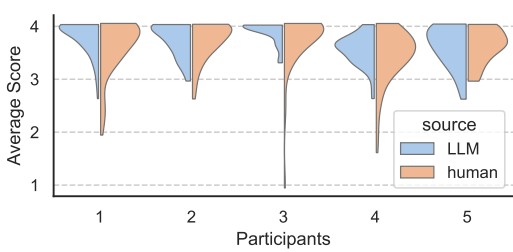

Figure 18: Model *"collapse"* to *"common"* values.

## H.6 CONFUSION ANALYSIS

As we have discussed in the introduction (Section 1) and Appendix A, the common model collapse behavior is to always generate the "common" value of a factor, ignoring the causal relationships in the scenario. For example, we observe that: 1) when a knife appears around a piece of butter, the butter is always cut up ignoring whether the knife is move against the butter. When tea is poured into a teacup, the water level always rises to the top, regardless of whether water is continuously poured or whether water has actually been poured into the cup. It can be roughly illustrated by the Figure 18.

We provide per-scenario confusion analysis for five scenarios in the Table 22 to support our findings. For each scenario, we list the rules most frequently violated and the typical "collapsed" behaviors that VGMs tend to default to, likely reflecting patterns learned from their training data.

Table 22: Per-scenario confusion analysis.

| Scenario | Causal rules for outcomes | Rule violated | Common collapsed outcome |
|---|---|---|---|
| A stone is thrown into a pool, creating a splash. | Splash visible = Steep entry angle
Water hits wall/deck = Impact adjacent to wall
Concentric ripples visible = Calm water surface
No surface skipping = Steep entry angle | **No surface skipping = Steep entry angle**: even for small entry angles, the object often still does not skip. | The object usually just enters the water and sinks without skipping, regardless of entry angle. |
| A person strikes an ice block with a hammer. | Contact imprint = Sharp striker
Support yields = (Rigid support = false)
Glancing slide = (Perpendicular strike = false) | **Glancing slide = (Perpendicular strike = false)**: non-perpendicular strikes often still fail to slide or glance off. | The striker tends to stick or stop on impact, with little glancing or sliding even when the strike is oblique. |
| A brush dips into watercolor on a palette. | Liquid surface disturbed = Brush contacts liquid
Bristles change color =
(Brush contacts liquid ∧ Colored liquid present) | **Bristles change color = Brush contacts liquid ∧ Colored liquid present**: the brush touches colored liquid but the bristles often remain unchanged. | The brush frequently stays essentially the same color, despite contact with the colored liquid. |
| An air mattress floats on a pool. | Deck above waterline initially = Adequately inflated
Waterline rises after boarding = Person boards
Moves due to push = External push or pull | **Waterline rises after boarding = Person boards during clip**: the waterline often does not rise when a person boards the mattress. | The mattress and surrounding waterline remain at nearly the same height, independent of whether someone boards it. |
| Pour the salt into the water | Surface ripples at impact = Salt contacts water
Stream deflects at rim = Rim impact occurs
Visible vortex motion = Rotational stirring occurs | **Visible vortex motion = Rotational stirring occurs**: rotational stirring rarely produces a clear vortex. | The liquid typically shows only mild local mixing without a coherent vortex structure, regardless of stirring. |

## I CASE STUDY ON BENCHMARK RESULTS

### I.1 ABOUT THE "DEGENERATIVE" RULES

Since our metric 2 only focus on the stability but not the correctness, we are worried that the lower (better, stabler) metric 2 combined with the poorer metric 1 and metric 3 (low accuracy) actually implies that the model learns shortcut on common scenario. In many cases, models ignore the changes in $\mathbf{X}$ but directly generate the most common results. We support our concern through some case studies.

In the scenario about "A burning candle is placed with (wind and rain).", a key outcome is whether the candle remains lit or is extinguished by these environmental factors. However, we found that most of the VGMs consistently generate a candle that continues to burn, without accounting for

these influences. For Gen-3 Alpha, in three test cases of this scenario, the expected outcome—an extinguished candle—occurred 11, 10, and 10 times, respectively. However, the actual results were only 2, 0, and 3 instances where the candle was extinguished. This makes the "candle extinguished" result appear almost as a constant "False". Similar phenomenon can be found about the outcome "whether the pencil mark has been removed" in the scenario "Rubber eraser rubs off (pencil) marks on paper.". Similarly, the statement "the water color is uniform" is always false after "Dropping dye into the water" regardless of "whether the water is stirred sufficiently".

### I.2 ABOUT SAMPLE-BASED SCORE

Here we demonstrate how the sample-based scores provide a more detailed analysis of model behavior by an example. Taking the model CogVideoX1.5-5B and the scenario "*A hand squeezes a sponge.*" as the example, one of the generated causal system is:

"hand squeeze sponge ∧ sponge is wet → water is squeezed out".

By checking the scores of the generated videos, we observe that some videos have a metric 3 score (rule consistency) of 1.0 (full score), indicating that these videos comply with all rules. We show these videos are shown in Figure 19, corresponding to some successful generation. As comparison, some of generation have much lower metric 3 score and are shown in Figure 20. Intuitively, we can see the gap in generated causal content between them. In this way, we can select some better samples which could be used to further finetune the model to achieve better causal alignment in this scenario.

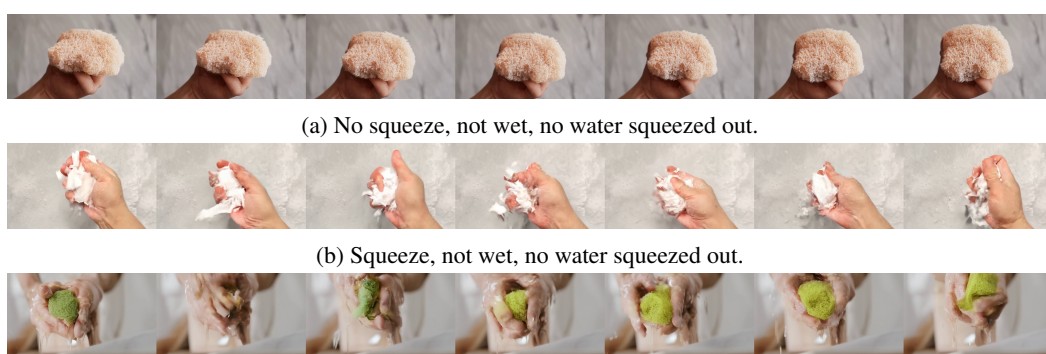

(a) No squeeze, not wet, no water squeezed out.

(b) Squeeze, not wet, no water squeezed out.

(c) Squeeze, wet (deeper color in first several frames), water squeezed out.

Figure 19: Good examples with rule consistency score 1.0.

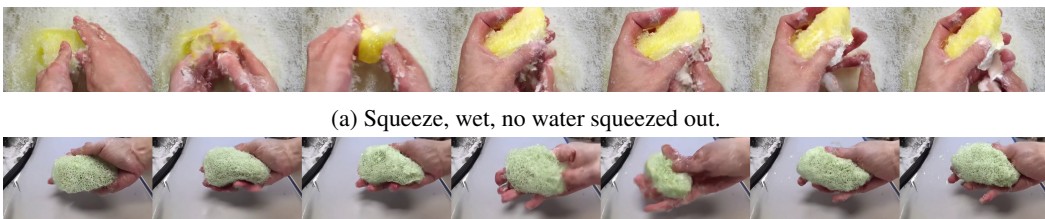

(a) Squeeze, wet, no water squeezed out.

(b) Squeeze, not wet, water squeezed out (water droplets appear in the last two images).

Figure 20: Bad examples with rule consistency score 0.0.

## J DISCUSSION ABOUT LLM PROMPT ENHANCEMENT TECHNIQUE

Sora (OpenAI, 2024) inherits a technique from Dall-E (Betker et al., 2023) called prompt enhancement, where the model doesn't directly rely on the provided text prompt for generation. Instead, it first uses a pre-trained LLM to expand the prompt, adding missing elements such as environmental

details and turning abstract concepts into more intuitive descriptions. Some models have already integrated this functionality into their latest VGM versions.

We indeed observed that this technique slightly improved the model's ability to correctly understand causal rules. However, when scenarios became slightly more complex, either the LLM's expansion did not address the relevant parts, or even if the LLM did provide an expansion, the VGM still failed to generate reasonable results. We believe that, this technique is not the ultimate solution to creating a world simulator. On one hand, it supplements the VGM's shortcomings by leveraging the LLM's capabilities, but it doesn't address the VGM's core strengths. On the other hand, prompt enhancement cannot capture every detail because vision is much more complicated and informative than text, and once a scenario goes beyond the scope of the prompt, the VGM will struggle to respond appropriately.

To faithfully reflect the performance of the VGMs themselves, we disabled the prompt enhancement option for all closed-source models (where possible). Specifically, for Hailuo, we turned off this feature. For Kling and Pika, however, we couldn't find any official description on whether this technique was used.

