# OpenReview forum: "VACT: A Video Automatic Causal Testing System and a Benchmark"
_ICLR.cc/2026/Conference — Submitted to ICLR 2026_

### Official Review · Reviewer_GRfX · 2025-10-28

**Soundness:** 4
**Presentation:** 3
**Contribution:** 4
**Rating:** 4
**Confidence:** 3

**Summary:**

This work introduces VACT(Video Automatic Causal Testing), a fully automated framework to evaluate and measure causal understanding in Text to Video Generation Models.
The work leverages a large language model pipeline to assess causal behavior under difference scenarios via intervention experiments.
Analysis of causal learning are defined under 3 levels; text consistency, generation consistency and rule consistency.
The paper benchmarks several state-of-the-art VGMs (e.g., CogVideo, Hailuo, Kling, Veo3-Fast, Pika) across 19 scenarios and 718 evaluation videos, identifying significant gaps between human-like causal understanding and current model capabilities.

**Strengths:**

Work is well-motivated and introduces the first automated causal benchmarking pipeline offering a scalable solution to evaluating VGMs.\
Multiple self-checking and self-correction in causal rule extraction using LLMs to mitigate errors in the generation process.\
Crowdsourced comparisons between LLM-generated and human-annotated causal systems show that LLM annotations can outperform humans on rationality and soundness.\
The causal hierarchy (text / generation / rule consistency) provides a clear and extensible structure for measuring causal learning at multiple levels.

**Weaknesses:**

While the work attempts to offer a comprehensive evaluation suite to benchmark VGMs under the 3 mentioned frameworks evaluation is done only using VLLM for answer retrieval without any kind of upper-bound it is difficult to draw any conclusion from the metrics provided by the benchmark. An alternative would be perhaps an average or consensus across several VLLMs
Furthermore, N/A ratio is quite significant. Are the scores adjusted in anyway to account for this?

Claims at line 395 "Thus, our benchmark specifically requires models to handle multiple variables at once, including those corresponding to less common scenarios. The results highlight the models’ difficulty in dealing with complex properties and rare situations. This suggests that current models are still constrained to common scenarios and lack the generalization capability needed to effectively combine independent variables in broader contexts—a necessary capacity for building world simulators." needs to be better supported either with ablations such as models performance vs number of variables or complexities.\
The benchmark’s “truth” and “observe” metrics are under explained, and it is not entirely clear how to interpret discrepancies between them in causal terms. If the score is separated by grouping samples with and without generation errors, how much samples exists within each group? Is there a high deviation in the samples sizes?\

Minor comment\
Authors have a significant bulk of details in the appendix, i think they would be better served in summarizing some these in more details in main body itself especially on the evaluation process in section 4 and 5 to provide a better picture on the numerical significance of the results.

**Questions:**

Have you compared LLM-based evaluations to human causal judgments on a subset of data?
How stable are causal scores across different evaluator models or prompt perturbations?

The reliance on a single evaluator model and absence of clear upper/lower bounds make the causal metrics difficult to interpret confidently.\
The conceptual novelty,motivation and potential value are strong, but the empirical validation and benchmarks are insufficient for me to confidently believe in the supported the claimed conclusions.\

---

> ### Author Response · Authors · 2025-11-26
> **Official Comment by Authors (1/2)**
>
> Thank you for your inspired suggestions. We are glad to try our best to solve your concerns.
>
> 1. **An average or consenses across several VLLMs**:
>      - We agree that using different VLM and repeat the answer retrieval is a good way to reduce the error rate and effect. We add related content in section 3 Table 2 (line 324-328). In conclusion, we find that 1) The consistency between different VLM is high ( $\geq 93\%$ ), and 2) using multiple VLMs can further improve the reliability of the answer retrieval. So we now report the results using the majority vote from three VLMs.
>
> 2. **Single-VLM vs. multi-VLM evaluation**:
>    - We compared the benchmark results obtained using GPT-5 alone as the VLLM judge with those obtained using a majority-vote scheme over three VLLMs (GPT-5, Qwen-VL-Plus, and Gemini 2.5 Flash).
>
>    - The first table reports the results using **GPT-5 only** as the VLLM judge:
>
>       | Model Names | N/A ratio | Text Consistency (all) | Text Consistency (root) | Generation Consistency (truth) | Generation Consistency (observe) | Rule Consistency (truth) | Rule Consistency (observe) |
>       |-------------|-----------|------------------------|-------------------------|---------------------------------|----------------------------------|--------------------------|----------------------------|
>       | Recorded\*  | 0.00      | 0.83                   | 0.88                    | 0.10                            | 0.07                             | 0.95                     | 0.93                       |
>       | CogVideo    | 0.24      | 0.63 ± 0.02            | 0.67 ± 0.02             | 0.22 ± 0.03                     | 0.22 ± 0.05                      | 0.56 ± 0.03              | 0.83 ± 0.03                |
>       | Hailuo      | 0.27      | 0.56 ± 0.02            | 0.67 ± 0.03             | 0.40 ± 0.03                     | 0.28 ± 0.04                      | 0.61 ± 0.02              | 0.77 ± 0.04                |
>       | Kling       | 0.21      | 0.60 ± 0.02            | 0.70 ± 0.03             | 0.23 ± 0.04                     | 0.21 ± 0.05                      | 0.61 ± 0.03              | 0.76 ± 0.04                |
>       | Veo3-Fast   | 0.15      | 0.71 ± 0.02            | 0.76 ± 0.02             | 0.17 ± 0.03                     | 0.28 ± 0.05                      | 0.70 ± 0.03              | 0.79 ± 0.04                |
>       | Pika        | 0.25      | 0.67 ± 0.02            | 0.72 ± 0.03             | 0.16 ± 0.03                     | 0.33 ± 0.05                      | 0.72 ± 0.03              | 0.76 ± 0.04                |
>
>    - The second table reports the results using **majority voting** over GPT-5, Qwen-VL-Plus, and Gemini 2.5 Flash as the VLLM judges:
>
>       | Model Names | N/A ratio | Text Consistency (all) | Text Consistency (root) | Generation Consistency (truth) | Generation Consistency (observe) | Rule Consistency (truth) | Rule Consistency (observe) |
>       |-------------|-----------|------------------------|-------------------------|---------------------------------|----------------------------------|--------------------------|----------------------------|
>       | Recorded\*  | 0.00      | 0.83                   | 0.88                    | 0.10                            | 0.07                             | 0.95                     | 0.93                       |
>       | CogVideo    | 0.21      | 0.61 ± 0.02            | 0.68 ± 0.02             | 0.33 ± 0.03                     | 0.25 ± 0.06                      | 0.62 ± 0.03              | 0.82 ± 0.04                |
>       | Hailuo      | 0.22      | 0.58 ± 0.03            | 0.71 ± 0.03             | 0.25 ± 0.03                     | 0.27 ± 0.04                      | 0.62 ± 0.03              | 0.82 ± 0.05                |
>       | Kling       | 0.19      | 0.62 ± 0.02            | 0.70 ± 0.03             | 0.20 ± 0.04                     | 0.24 ± 0.05                      | 0.65 ± 0.03              | 0.76 ± 0.04                |
>       | Veo3-Fast   | 0.13      | 0.70 ± 0.02            | 0.75 ± 0.02             | 0.19 ± 0.02                     | 0.25 ± 0.04                      | 0.67 ± 0.03              | 0.76 ± 0.04                |
>       | Pika        | 0.20      | 0.69 ± 0.02            | 0.74 ± 0.02             | 0.21 ± 0.03                     | 0.29 ± 0.04                      | 0.68 ± 0.03              | 0.79 ± 0.04                |
>
>    - As shown in the tables, we obtain slightly different numeric values but the main conclusions remain unchanged: Veo3-Fast and Pika consistently achieve the best text and rule consistency scores. Overall text consistency remains only moderate. Generation and rule consistency further confirm the same qualitative picture as before, with models often collapsing to “common” outcomes and only partially capturing real-world causal rules rather than robust, generalizable causal mechanisms.

---

> ### Author Response · Authors · 2025-12-02
> **Official Comment by Authors (2/2)**
>
> 3. **Upper bound of the metrics**:
>    - All of our three scores range from [0,1] theoretically.
>    - If you wonder some error from the inaccuracy from VLLM answer retrieve, we newly added a discussion in Section 3.3 and Appendix G.1. In conclusion, the effect of the inaccuracy of LLM/VLLM is bounded.
>    - We also recorded some real-world videos, which is a proxy of the videos with all correct causal relations, to estimate the empirical upper bound of each metric. The results are reported in Section 5 (line 427-431).
>
> 4. **Are the scores adjust to account for N/A ratio?**
>    - Yes, we have filtered out videos with "N/A" answers during benchmark curation as we described in line 416.
>
> 5. **Support for *"constrained to common scenarios and lack the generalization capability"***
>    - Yes, we have added a statistical analysis about the correlation between the number of variables in the causal graph and the model performance in Section H.5 (line 3163-3170), which provides a quantitative view of how causal complexity affects performance.
>    - We also show some per-scenario analysis to illustrate the behavior of "preferring common". Please refer to the Point 5 of Reviewer iTw7 response for more details. The results show that current models indeed tend to collapse to some common behaviors, rather than generate diverse outcomes according to different causal factors.
> 6. **"Truth" vs. "Observe"**:
>    - These two scores are **not separated** but **calculated in different ways**.
>    - Considering these scores measure how well the model has learned a functional relationship $y=f(x)$, the value of $x$ shown in the video may differ from the one requested in the text. For example, although our text emphasizes *"a stone is thrown quickly into the water"*, the video may incorrectly depict it as *"slowly"*.
>    - So for each sample we compute two scores: (1) using the text-specified $x$ to compute the "Truth" score, and (2) using the observed $x$ in the video to compute the "Observe" score.
>    - The "Truth" score is an overall causal evaluation, where errors in handling $x$ are also taken into account; The "Observe" score reflects only the model’s error in mapping from $x$ to $y$, isolating this from any mistakes in generating $x$ itself.
> 7. **Appendix to main text**:
>    - Thank you for your suggestion. We have added or move some content to the main text to support our numerical significance (line 320-330, 427-431 and our new Table 2).
>    - We also added a Table of contents before Appendix to help readers navigate the supplementary material. We hope that it can improve the readability.
> 8. **Comparison between LLM-based evaluations and human evaluations**:
>    - Yes, we have manually evaluated some videos, where the consistency between human and LLM-based evaluations is 96% (new Table 2). It means that less than 4% noise is introduced by our VLM-based answer retrieval.
> 9. **How stable across prompt perturbations?**
>    - We emphasize that the *prompt engineering* is also an important part of our pipeline. The prompt of our VLM has been carefully designed and tested, which is described in Appendix E.3. We iteratively refined this prompt based on early experiments, aiming to provide sufficiently clear definitions for all plausible cases and to avoid ambiguity. Our early version (without these instructions) shows about 60-70% consistency and the current version achieves over 95% consistency based on our human evaluation.
>    - Importantly, the instructions focus on describing the task itself rather than adding VLM-specific tweaks, so the current version works well across different model families (GPT or Gemini) and versions (GPT-5 or GPT-4o).
> 10. **As a conclusion**, we appreciate your valuable suggestions to improve the soundness of our conclusions.
>     - We have added a statistical analysis about the error effect of LLM and VLLM (Point 1, Section 3.3).
>     - We have added comparison between different VLMs (Point 2).
>     - Our recorded real-world videos shows that the better causal consistency videos can achieve higher scores (Point 1).
>
> Thank you for your constructive suggestions. In response to your concerns, we have added new results in our revised paper. We hope these additions address your questions, and we would greatly welcome any further feedback.

---

### Official Review · Reviewer_GYuy · 2025-10-28

**Soundness:** 1
**Presentation:** 1
**Contribution:** 1
**Rating:** 2
**Confidence:** 4

**Summary:**

This paper presents a pipeline to check the factual and physical accuracy of videos generated by AI video generators. Pipeline is claimed to automatically extract out important rules that must be followed by generated videos from prompts and VLMs are used to check if these rules are followed in the generated videos. Rule extraction process is claimed to make use of causal graphs and is totally self-verified. So, yes, the problem of factual and physical accuracy is completely solved almost, as per the paper.

**Strengths:**

- Problem targeted by the paper is important

**Weaknesses:**

- presentation is quite poor and the paper lacks clarity along with typos. Paper seems like a rush job.
- my main issue is that paper claims that everything---exhaustive rule extraction, causal graph generation, physical quantification/estimation, rules verification---just magically works---we just had to mention the tasks to LLMs and VLMs, which no prior work did, and this paper simply did it.
- In reality and my experience, all these steps are very imperfect and self-checks are not reliable, and requires human oversight, but the paper claims otherwise, which I am not able to buy.
- Exhaustive rule extraction itself is extremely challenging
- Quantities like speed, density, etc are relative, but somehow VLMs are able to accurately measure them absolutely
- Paper also takes anecdotal observation and extends to be statistical
- "LLM-generated annotations surprisingly outperformed those from human" when compared with human annotations---just how? How are apples going to look more like oranges than other oranges?!
- Experimental results and analysis is lacking. I did not find good insights in them. Qualitative results are also missing.
- I am really sorry, but I am not able to digest, that everything just magically works, and quite a few things in this paper are not adding up for me.

**Questions:**

Please see weaknesses, and consider them as questions.

---

> ### Author Response · Authors · 2025-11-26
> **Official Comment by Authors (1/3)**
>
> 1. W1: Thank you for your comments. We believe the value of peer-review lies in providing suggestions for further improvement for our work as well as the research field. An empty criticism without content is not constructive. We have revised the manuscript as follows:
>    - Line 70: "being utilized" -> "to be utilized"
>    - Line 72: "some work" -> "some works"
>    - Line 106: "is referred to" -> "refers to"
>    - Line 220: add whitespace after the period.
>    - Line 237: add a missing period.
>
>     We are also curious about which parts you think is *"poor presentation"*, and we look forward to your specific suggestions.
>
> 2. W2.1: We respectfully disagree that our pipeline ''*just magically works*''. The careful design of both the pipeline and each component are very important to make the system work. Some important designs include:
>    - The complex task (automatic test case proposal) is split into three steps: 1) factor analysis, 2) graph construction, and 3) rule formulation. The decomposition propose sub-targets for the model to focus on and make it possible for LLMs to handle the overall task. (Line 243-246)
>    - Our **self-check** is not a simple "do it again" trick but provide a more concrete prompt to guide the model to check our requirements. (Appendix B.2.2 Line 926-945) We have also added a detailed comparison between the output before and after self-check as Appendix B.3.
>    - We also design **rule-based** check where the structure output and some python codes are used to ensure some basic requirements. For example, we check whether the graph contains all the factors which proposed in the *factor analysis* step. (Appendix B.2.1 Line 890-922) During data generation, the rule-based feedback eliminates errors in roughly 10\% steps, demonstrating its role in catching basic mistakes.
>    - Our design of the output format is also important. For example, we find that requiring the model to explain *why* it considers a factor important improves the reliability of the output, so we add this into the requirements (Line 858-863, Appendix B.4 Line 1005-1020). These factor explanations are also used to guide subsequent steps such as text prompt generation and VLM detection.
>    - In princile, our pipeline tries to prompt the early stages to produce outputs that are more suitable for downstream stages. An example is that in our *factor analysis* step, we require the model to make sure the factors are visibly discriminative in the video and provide instructions on how to detect them. The instructions are both used 1) when generating the text prompt for T2V models to make sure the factors are appropriately described in the text input, and 2) when prompting VLMs to detect these factors in the video. (Line 220-222, Appendix B.5 Line 1092-1097)
>
>     We believe it is not fair to say *"magically works"* without recognizing our design and effort. We are glad to provide a paragraph to summarize some important insights in the revised draft to help readers better understand our design. (Newly added in Section 3.2 and Appendix. B)
>
> 3. W2.2: Regarding the remark that "we just had to mention the tasks to LLMs and VLMs, which no prior work did," we do not claim to be the first to apply LLMs/VLMs to video or physics tasks. As an example, we have mention related work PhyGenBench [1] has utilize LLM to generate questions based on human-designed physics laws, where our pipeline is a further automation of this idea. Using VLM to extract answer and find errors from videos is also tested in many benchmark like [2](Line 155-157). We extend these ideas to our causal rule setting.
>
> 4. W3: First, we indeed have human oversight in both our pipeline design and our benchmark curation. Most of our oversight which provide general guidance has been integrated into our pipeline and prompt, like our three-step splitting, rule-based check and requirements in prompt. Then, although our pipeline can generate test cases automatically, we do a (manual and rule-based) post-filtering to make sure all the scenario used in our **benchmark** are valid. Finally, we want to emphasize that self-check with carefully designed prompt and context could be reliable in many cases. For example, the current reasoning LLM and LLM agent highly relies on self-check ability and have been shown performance competitive with human experts. See technique report of [DeepSeek-R1](https://arxiv.org/pdf/2501.12948) or a recent work using self-verification to improve math reasoning [Self-verify](https://arxiv.org/pdf/2506.01369).

---

> ### Author Response · Authors · 2025-11-26
> **Official Comment by Authors (2/3)**
>
> 5. W4: We are not sure what "exhaustive" means here. We would like to clarify that our goal is not to extract all possible rules in a given scenario, which is indeed infeasible, but rather to recover the most important and practically relevant rules.
> 6. W5: We emphasize that our method does not rely on VLMs producing absolute calibrated values. Instead, we only require concept-level judgments which are sufficient for our tasks (like "high temperature", "low speed") (line 218–222). We also require the test case proposer to carefully check whether these qualitative factors can be clearly observable in the video and explain it. For example, the density cannot be directly observed but can be inferred from the visual appearance like material. These hints which are generated in the "explanation" by the factor analyzer are also provided to VLM to guide the detection (New added paragraph in section 3.2).
> 7. W6: We also do not understand what "anecdotal" means.
> 8. W7: Human is not an ''orange'' and LLM is also not an ''apple''.
>    1. Our task is inherently complex, and even human writers may disagree on requirements such as whether a factor is important, visible, or independent (for root factors), so there is often no absolute answer. For example, a human-proposed factor “spinning ball” was challenged by another scorer on the grounds that it is difficult to reliably observe in the videos. In contrast, with our repeated self-checking, rule-based checks, and pipeline design, the LLM often produces more reliable and less controversial answers, because it can systematically explore more possibilities and propose more diverse factors than humans in a short time.
>    2. At the same time, the task is inherently creative rather than a binary true/false judgment. As discussed in Lines 275–279, the model is required to discover important factors, which demands both an accurate understanding of the scenario and imaginative consideration of alternative possibilities. For example, in the scenario "something is thrown into the swimming pool", human annotators typically identify the object’s density as important, whereas the LLM can additionally propose that sunlight may indicate whether the water is sparkling. This more careful analysis of the scenario leads to higher soundness scores in our human evaluation, and some human-designed rules are judged by scorers to be insufficiently comprehensive.

---

> ### Author Response · Authors · 2025-11-26
> **Official Comment by Authors (3/3)**
>
> 9. W8: Our experiment about our system is in Table 1 and Appendix D. For other components, we have tested their accuracy and reported in Appendix G. We also provide a new table to summarize the performance of each component in the revised draft. (See Section 3, New Table 2)
>
> Overall, our results rely on a carefully designed pipeline and a number of deliberate choices for testing VGMs’ causal reasoning ability, which we believe are themselves useful contributions to this area, as also reflected in the positive assessment by Reviewer GRfX (Strength 4). As the LLM/VGM field rapidly advances, these models’ foundational capabilities can be surprisingly strong when they are properly guided and constrained.
>
> [1] Meng et. al.,  world simulator: Crafting physical commonsense-based benchmark for video generation, 2024.
>
> [2] Li et. al., Worldmodelbench: Judging cideo generation models as world models, 2025.

---

### Official Review · Reviewer_iTw7 · 2025-11-05

**Soundness:** 3
**Presentation:** 3
**Contribution:** 3
**Rating:** 6
**Confidence:** 3

**Summary:**

The paper introduces an automated framework to test whether text-to-video generative models exhibit causal understanding of physical scenarios. It uses an LLM to propose scenario-specific causal systems, performs interventional prompt design over root variables. It evaluates generated videos with three metrics: text consistency, generation consistency, and rule consistency. On the constructed comprehensive benchmark dataset, current models often fail to respect causal rules despite good visual fidelity.

**Strengths:**

- Task novelty: An automatic evaluation framework is critical for VGM development. The proposed LLM-driven construction of scenario-specific causal graphs and DNF rules, with self-check and rule-check loops, reduces manual effort and scales across domains.
- Metric quality: Clear separation of text consistency, generation consistency, and rule consistency, plus “truth/observe” variants that expose degenerate behavior.
- Significance: The benchmark spans 19 scenarios and shows consistent causal errors in current systems; results provide actionable targets for future VGM development.
- Human validation: Crowd experiments indicate the LLM’s causal system proposals meet or exceed human annotations on requirement, rationality, and soundness.

**Weaknesses:**

- Measurement dependency on VLLMs: Answer retrieval uses a VLLM with limited manual auditing. If the VLLM misreads subtle events, metric validity is affected. Authors note “random manual checks,” but a systematic calibration study is absent.
- Binary discretization and visibility constraints: Mapping continuous physics to binary, video-visible variables may oversimplify rules and bias towards easily observable effects.
- LLM both proposes and helps judge: The same family of models proposes rules and generates probes; risk of confirmation bias remains.

**Questions:**

- How sensitive are the metrics to the choice of VLLM and its prompt template?
- Can you provide per-scenario confusion analyses showing which outcomes most often collapse to “common” behaviors?
- Do results hold when variables are non-binary (e.g., 3-level speed)? If not, how would VACT generalize?

---

> ### Author Response · Authors · 2025-11-26
> **Official Comment by Authors (1/2)**
>
> 1. **W1**: about the noise from VLLM
>     - During manual calibration, we randomly sampled generations from four VGMs on three scenarios, each evaluated under three different prompts, yielding 168 VLLM question–answer pairs. The VLLM predictions were correct on 162/168 pairs (96%) overall. By scenario, we obtained 68/72 correct (94%) for Scenario 1, 58/60 correct (97%) for Scenario 2, and 36/36 correct (100%) for Scenario 3. Detailed results are reported in Section G.3 and summarized in the table below.
>
>       | Scenario   | Q–A pairs | Correct | Wrong | Accuracy |
>       |-----------|------------|----------|--------|----------|
>       | Scenario 1| 72         | 68       | 4      | 94%      |
>       | Scenario 2| 60         | 58       | 2      | 97%      |
>       | Scenario 3| 36         | 36       | 0      | 100%     |
>       | **Total** | **168**    | **162**  | **6**  | **96%**  |
>
> 2. **W2, Q3**: Binary discretization and visibility constraints
>     - The abstraction from absolute numeric values to concept-level judgments is sufficient for assessing whether a video is causally plausible in **everyday** situations. We explicitly design our scenarios so that models must clearly exhibit "fast vs. slow" rather than ambiguous intermediate states, because our goal is not to evaluate *exact numerical physical laws* but to focus on *conceptual, everyday physical understanding*.
>     - In principle, our framework can also accommodate multi-level factors (e.g., three speed levels) without changing the underlying causal graph, since the formalism supports arbitrary discrete values. Concretely, one can generalize the rules to a multi-valued form (e.g., an if-elif-else structure) and adjust the VLM prompts accordingly, while leaving the rest of the pipeline unchanged.
>     - However, moving beyond binary variables would (1) substantially increase the difficulty for T2V models, especially given the current binary version has been proved as challenge for current models and (2) require more accurate world understanding from the LLM and more precise perception from the VLM. We leave a careful multi-level extension of VACT as important future work.
>     - Benchmarks that target precise continuous values already exist, such as ContPhy[1], which requires inferring continuous properties like mass and density and predicting dynamics across diverse scenes, and Morpheus[2], which evaluates video generation models using physics-informed metrics tied to conservation laws. But their works focus on limited and controlled situations, and several basic physical laws, while our work aims to evaluate more diverse and daily scenarios, which is more close to real-world applications of T2V models.
>
> 3. **W3**: LLM both proposes and helps judge
>     - In our paper, the LLM is used to create the test cases for T2V models. The T2V model is the target model we want to evaluate. The VLM is a helper model to extract answers. The final score is purely mathematical calculation and induced from the causal inference theory. These models can be different models and from different families.
>     - There are some techniques to minimize the risk of the bias. For example, we added some different VLMs (like Gemini and Qwen) to retrieve answers.
>     - At the same time, for each component, we have tested their reliable and accuracy by human evaluation, which suggests that the risk of confirmation bias is low in practice. Like our crowd experiments in Section 3.2 (line 262-272) and our error analysis in line 322-341.
>
> 4. **Q1**: about the VLLM
>     - We have added experiments using multiple VLMs to reduce the error rate and effect. Please refer to Point 1 and Point 2 of Reviewer GRfX response for more details. We find the consistency of different VLLMs are high and the results are stable for the VLLM selection. Furthermore, we believe it is a good suggestion to improve the reliability of our benchmark so now we report the results using the majority vote from three VLMs (line 321-331).
>    - We emphasize that the *prompt engineering* is also an important part of our pipeline. The prompt of our VLM has been carefully designed and tested, which is described in Appendix E.3. We iteratively refined this prompt based on early experiments, aiming to provide sufficiently clear definitions for all plausible cases and to avoid ambiguity. Our early version (without these instructions) shows about 60-70% consistency and the current version achieves over 95% consistency based on our human evaluation.
>    - Importantly, the instructions focus on describing the task itself rather than adding VLM-specific tweaks, so the current version works well across different model families (GPT or Gemini) and versions (GPT-5 or GPT-4o).

---

> ### Author Response · Authors · 2025-11-26
> **Official Comment by Authors (2/2)**
>
> 5. **Q2**:  per-scenario confusion analyses
>     - Sure! We believe it is a good suggestion to help reader understand better of the VLM's causal inaccuracies. We provide per-scenario confusion analyses for five scenarios in the following table. For each scenario, we list the rules most frequently violated and the typical “collapsed” behaviors that VGMs tend to default to, likely reflecting patterns learned from their training data. See Table 6 in the revised version (line 494-496, line 510-518) and Appendix H.6. Table 22.
>
> We sincerely appreciate your constructive feedback, which helped strengthen the paper. We have included additional experiments and analyses to address your concerns and updated the manuscript accordingly. We hope these changes clarify the issues you raised, and we would be glad to answer any further questions.

---

### Author Response · Authors · 2025-11-26
**Summary of our revision**

We are pleased that the reviewers recognize the novelty and significance of our work. We also greatly appreciate the constructive suggestions from several reviewers, and we have made improvements to the paper accordingly. We summarize the main changes below. We ...

1. add new discussion and comparison about our **pipeline design** to show the importance of step splitting and other design (like self-correction). (Section 3.2; l.243-246; Reviewer GYuy W2,W3)
2. mention that the **explanation generated by LLM will guide VLM** to retrieve the answer. (Section 3.2; l.253-258; Reviewer GYuy W2,W3)
3. add a **quantitative analysis about the error effect** of LLM and VLLM. (Section 3.3, G.1 ; l.322-330,1773-1789; Reviewer iTw7 W1, Q1)
4. add an experiments where **multiple VLLM's answer** will use together to get more accurate factor value
5. **update some LLM prompt** (like using pydantic to prompt LLM generate more accurate format). (Section B.3.1; l.794-827; Reviewer GYuy W2,W3)
6. add further discussion of the crowd experiment results to **clarify why LLM-generated annotations can outperform** those produced by human annotators. （Section 3.2; l.276-280; Reviewer GYuy W7)
7. **record real-world videos** to serve as **ground-truth references** and to estimate the empirical upper bound of each metric (Section 5; l.429-433; Reviewer GRfX W1,Q2)
8. compute **statistical correlations between model performance and the number of variables** in the causal graph, providing a quantitative view of how causal complexity affects performance. (Section H.5; l.3012-3019; Reviewer GRfX W2)
9. conduct **per-scenario confusion analyses** for 5 sampled scenarios to illustrate which behaviors current models most often collapse to (Section H.7; l.3134-3136;  Reviewer iTw7, Q1)
10. explain how to **filter out scenarios** in which the relevant factors cannot be reliably perceived from the video (Section 3.1; l.224-228; )

Note that we have submitted a **revised version** of the paper. We encourage reviewers to consult the revised paper for any parts of interest. Important modifications are highlighted in **blue**. To facilitate cross-referencing with the concerns raised in the reviews, we also cite line numbers extensively in our responses; please note that *all referenced line numbers correspond to the revised version*. We again sincerely thank the reviewers for their recognition and constructive feedback.

---

> ### Author Response · Authors · 2025-12-02
> **Summary of Rebuttal and Revision**
>
> We thank the reviewers for their constructive feedback. We have revised the manuscript to include comprehensive validation experiments and deeper analyses to ensure the robustness of VACT.
>
> **1. Reliability of VLM-based Evaluation**
> * **Manual Calibration:** To address concerns about VLM noise, we performed a manual audit on 168 randomly sampled generations. The VLM predictions achieved **96% consistency** with human judgment. (Reviewer iTw7 W1)
> * **Prompt Stability:** We emphasized the role of rigorous **prompt engineering**. Through iterative refinement based on early experiments, we improved consistency from 60-70% to over **95%** based on our human evaluation. The prompts focus on clear task definitions rather than model-specific tweaks, ensuring stability across different model families.(Reviewer iTw7 Q1;Reviewer GRfX W8)
> * **Multi-VLM Consensus:** We introduced a **majority vote mechanism** using three different VLMs (GPT-5, Gemini 2.5 Flash, and Qwen3-VL-Plus). The results show high inter-model consistency ($\ge 93\%$), significantly reducing the risk of single-model bias.(Reviewer iTw7 W1,W3,Q1;Reviewer GRfX W2)
> * **Empirical Upper Bound:** We evaluated **real-world videos** (as a proxy for ground truth) to establish an empirical upper bound. This validates that videos with correct causal relations indeed achieve higher scores in our metrics.(Reviewer GRfX W1)
> * **Bounded Error Impact:** We added a **quantitative analysis** demonstrating that the error introduced by VLLM inaccuracy is mathematically **bounded**. This ensures that the final metrics remain valid and reliable even in the presence of minor retrieval noise.(Reviewer iTw7 W1,Q1; Reviewer GRfX W1)
>
> **2. New Statistical Analyses and Insights**
> * **Per-Scenario Confusion Analysis:** We provided per-scenario confusion matrices to reveal specific failure modes. The analysis highlights that VGMs frequently **collapse to "common" behaviors** seen in training data rather than generating diverse outcomes based on control signals.
> * **Correlation Analysis:** We added a statistical analysis on the correlation between the **number of variables** in the causal graph and **model performance**. The results confirm a negative correlation, quantitatively demonstrating that current VGMs struggle as causal complexity increases.(Reviewer GRfX W4)
>
> **3. Clarifications on Methodology & Metrics**
> * We have also clarified several conceptual misunderstandings, including the validity of **binary physics abstraction** for evaluating everyday conceptual understanding, the definition of **the upper bound of each metric** (theoretically ranging in [0,1]), the handling of **N/A ratios**, and the specific mathematical distinction between **"Truth" vs. "Observe"** scores. (Reviewer iTw7 W2,Q3; Reviewer GRfX W1,W3,W5)

---

### Meta-Review · Area_Chair_VpRi · 2026-01-05

**Summary:**

This paper introduces VACT, an automated framework and benchmark (VACT Bench) to evaluate causal/commonsense consistency in text-to-video generation by (i) constructing scenario-specific causal systems and intervention prompts, and (ii) using VLM-based variable extraction to score text, generation, rule consistency (including truth vs. observe variants).
I find the direction timely and potentially useful, but I cannot recommend acceptance due to the review feedback and remaining risks.

**Reviewer Concerns:**

(1) Evaluation reliability and potential bias: multiple reviewers raise concerns that the pipeline heavily depends on VLM components for both constructing and evaluating causal rules, which may introduce systematic bias and limits interpretability of the scores.
(2) Simplification and ambiguity: the framework discretizes causal factors and filters “N/A” cases, which appears necessary for automation but may oversimplify causal mechanisms and complicate score interpretation (including the truth/observe distinction and N/A ratio concerns).
(3) Presentation and evidence concerns: one reviewer is strongly unconvinced, describing the work as over-claiming what VLM automation can reliably do and lacking compelling qualitative analysis, and this lack of confidence is difficult to offset within the current submission.

**Reviewer Scores:**

iTw7 (initial 6): likely stays 6 (could be slightly higher given added calibration, but still cautious about judge dependence).
GRfX (initial 4): likely moves slightly upward to 6 (some concerns are addressed, but evaluation issues remain).
GYuy (initial 2): likely stays 2 (strong skepticism about claims and automation reliability).
Wa5Z: no substantive review is visible in the provided discussion log, so I cannot estimate a score change.

---

### Decision · Program_Chairs · 2026-01-26

Reject